# IN-CONTEXT LEARNING FOR PURE EXPLORATION

**Alessio Russo**[*]
Boston University
arusso2@bu.edu

**Ryan Welch**[*]
Stanford University
rcwelch@stanford.edu

**Aldo Pacchiano**
Boston University
Broad Institute of MIT and Harvard
pacchian@bu.edu

## ABSTRACT

We study the *active sequential hypothesis testing* problem, also known as *pure exploration*: given a new task, the learner *adaptively collects data* from the environment to efficiently determine an underlying correct hypothesis. A classical instance of this problem is the task of identifying the best arm in a multi-armed bandit problem (a.k.a. BAI, Best-Arm Identification), where actions index hypotheses. Another important case is generalized search, a problem of determining the correct label through a sequence of strategically selected queries that indirectly reveal information about the label. In this work, we introduce *In-Context Pure Explorer* (`ICPE`), which meta-trains Transformers to map *observation histories* to *query actions* and a *predicted hypothesis*, yielding a model that transfers in-context. At inference time, `ICPE` actively gathers evidence on new tasks and infers the true hypothesis without parameter updates. Across deterministic, stochastic, and structured benchmarks, including BAI and generalized search, `ICPE` is competitive with adaptive baselines while requiring no explicit modeling of information structure. Our results support Transformers as practical architectures for *general sequential testing*. Code repository https://github.com/rssalessio/icpe.

## 1 INTRODUCTION

Sequential architectures have shown striking in-context learning (ICL) abilities: given a short sequence of examples, they can infer task structure and act without parameter updates (Lee et al., 2023; Schaul & Schmidhuber, 2010; Bengio et al., 1990). While this behavior is well documented for supervised input–output tasks, as well as regret minimization problems, many real problems demand sequential experiment design: how do we allocate experiments to reliably infer an hypothesis? For instance, imagine a librarian trying to figure out which book you want by asking a series of questions. Similarly, in generalized search (Nowak, 2008), the learner adaptively chooses which tests to run, each partitioning the hypothesis class, to identify the true hypothesis as quickly as possible. This raises a natural question: can we leverage ICL for adaptive *data collection and hypothesis identification* across a family of problems?

We study this question through the lens of Active Sequential Hypothesis Testing (ASHT) (Chernoff, 1992; Cohn et al., 1996), a.k.a. *pure exloration* (Degenne & Koolen, 2019), where an agent adaptively performs measurements in an environment to identify a ground-truth hypothesis. In particular, we study a Bayesian formulation of ASHT, where each environment is drawn from a family of possible problems $\mathcal{M}$.

Classically, ASHT has been studied either (i) with a fixed confidence $\delta$ (i.e., stop as soon as the predicted hypothesis is correct with error probability at most $\delta$) (Jang et al., 2024) or (ii) a fixed sampling budget (use $N$ samples to predict the correct hypothesis) (Atsidakou et al., 2022). For example, in the fixed-confidence setting one can use ASHT to minimize the number of DNA-based tests performed to accurately detect cancer (Gan et al., 2021). Another canonical instantiation is Best-Arm Identification (BAI) in stochastic multi-armed bandits (Audibert & Bubeck, 2010). In this problem the agent sequentially selects an action (the query) and observes a noisy reward: the task is to identify the action with the highest mean reward[1]. Other applications include medical diagnostics

---

[*]Equal contribution (alphabetical order).

[1]Note that, *in this particular case*, the hypothesis space coincides with the query space of the agent.

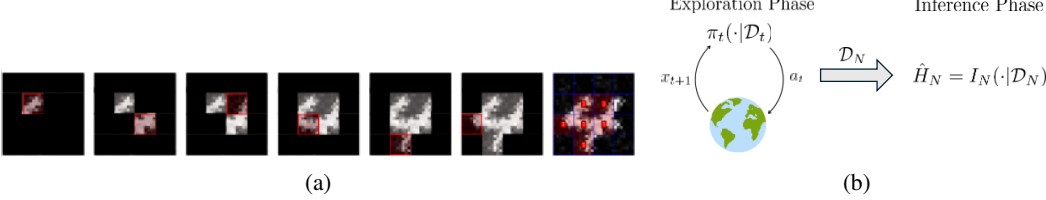

(a)                                                                    (b)

Figure 1: **(a)** Generalized search example: `ICPE` starts from a masked image (left), and sequentially reveals patches expected to reduce the posterior entropy over labels. It stops once the inferred label is $\delta$-correct (right). **(b)** After executing an action $a_t$, the agent observes $x_{t+1}$. At inference time, the data collected is used to infer an hypothesis.

(Berry et al., 2010), sensor management (Hero & Cochran, 2011) and recommender systems (Resnick & Varian, 1997).

Despite substantial progress (Ghosh, 1991; Naghshvar & Javidi, 2013; Naghshvar et al., 2012; Mukherjee et al., 2022), solving ASHT problems remains difficult. Even in simple tabular environments, computing optimal sampling policies often requires strong modeling assumptions (known observation models that do not depend on the history, and/or known inference rules) and solving challenging (often nonconvex) programs (Al Marjani et al., 2021). This leaves open whether one can *learn*, in a simple way, to both gather informative data and infer the correct hypothesis without such assumptions.

To answer this question, we introduce In-Context Pure Explorer (`ICPE`), a Transformer-based architecture meta-trained on a family of tasks to jointly learn a data-collection policy and an inference rule, in both *fixed-confidence* and *fixed-budget* regimes. `ICPE` is a model that transfers in-context: at inference time, `ICPE` gathers evidence on new tasks and infers the true hypothesis without parameter updates (Schaul & Schmidhuber, 2010; Bengio et al., 1990).

The practical implementation of `ICPE` emerges naturally from the theory alone, showing how a principled information-theoretic reward function can be used to train, using Reinforcement Learning (RL), an optimal data-collection policy. Additionally, `ICPE` relaxes classical assumptions: the data-generation mechanism $P$ is unknown and may be history-dependent, and the mapping from data to hypotheses is also unknown (we do not assume a known likelihood or a hand-designed test). These facts, combined with the simple and practical implementation of `ICPE`, offer a new way to design efficient ASHT methods in more general environments.

On BAI and generalized search tasks (deterministic, stochastic, structured), `ICPE` efficiently explores and achieves performance comparable to instance-dependent algorithms, while requiring only a forward pass at test time, and without requiring solving any complex optimization problem.

## 2    PROBLEM SETTING

The problem we consider is as follows: on an environment instance $M \sim \mathcal{P}$, sampled from a prior $\mathcal{P}$ over an environment class $\mathcal{M}$, the learner chooses actions (queries[2]) $a_t$ in rounds $t = 1, 2, \ldots$ and observes outcomes $x_{t+1}$. The aim is to gather a trajectory $\mathcal{D}_t = (x_1, a_1, \ldots, a_{t-1}, x_t)$ that is informative enough to identify an environment-specific ground-truth hypothesis $H^\star$ with high probability.

Informally, we seek to answer the following question:

> *Given an environment $M$ drawn from a prior $\mathcal{P}$, how can we learn (i) a sampling policy $\pi$ that collects data $\mathcal{D}$ from $M$ and (ii) an inference rule $I$ such that $I(\mathcal{D})$ reliably predicts $H^\star$?*

**Environments, sampling policy and hypotheses.** We consider environments $(M = (\mathcal{X}, \mathcal{A}, \rho, P, H^\star)$ with observation space $\mathcal{X}$, action set $\mathcal{A}$, initial observation law $\rho \in \Delta(\mathcal{X})$, and a (possibly history-dependent) generative mechanism $P = (P_t)_{t \geq 1}$ such that $x_{t+1} \sim P_t(\cdot | \mathcal{D}_t, a_t)$. All $M \in \mathcal{M}$ share the same $\mathcal{X}$ and $\mathcal{A}$. The learner uses a (possibly randomized) policy $\pi = (\pi_t)_{t \geq 1}$

---

[2]The reason why denote "queries" as "actions" stems from the fact that the problem can be modeled similarly to a Markov Decision Process (MDP)(Puterman, 2014), and queries correspond to actions in an MDP.

with $a_t \sim \pi_t(\cdot|\mathcal{D}_t)$, and a sequence of inference rules $I = (I_t)_{t \geq 1}$ with $I_t : \mathcal{D}_t \to \mathcal{H}$ for a finite hypothesis set $\mathcal{H}$. We assume throughout that $H^\star$ is induced by the environment via a measurable functional $h^\star$, i.e., $H^\star := h^\star(\rho, P)$, and is almost surely unique under $\mathcal{P}$.

**Example 2.1** (Best Arm Identification). *In BAI an agent seeks to identify the best arm among $K$ arms. Upon selecting an action $a$ at time $t$, it observes a random reward $x_t$ distributed according to a distribution $P(\cdot|a_t)$. The goal is to identify $a^\star = \arg\max_a \mathbb{E}_{x \sim P(\cdot|a)}[x]$ (so $H^\star = a^\star$). Many algorithms exist for specific assumptions ([Garivier & Kaufmann, 2016](); [Jedra & Proutiere, 2020]()), but designs change drastically with the model, and extensions to richer settings can be difficult and often non-convex ([Marjani & Proutiere, 2021]()).*

**Fixed confidence and fixed budget regimes.** Two regimes are usually considered in pure exploration:

- **Fixed confidence:** Given a target error level $\delta \in (0, 1)$, the learner chooses: (i) a stopping time $\tau \in \mathbb{N}$ that denotes the total number of queries and *marks* the *random* moment data collection stops, (ii) a data-collection policy $\pi$, (iii) and an inference rule $I$ that minimize the expected total number of queries $\tau$ while meeting a correctness guarantee:

$$\inf_{\tau, \pi, I} \mathbb{E}^\pi[\tau] \quad \text{s.t.} \quad \mathbb{P}^\pi(I_\tau(\mathcal{D}_\tau) = H^\star) \geq 1 - \delta. \tag{1}$$

  where $\mathbb{P}^\pi(\cdot)$ denotes the probability of the underlying data collection process when $\pi$ gathers data from $M$, and $M$ is sampled from a prior $\mathcal{P}$.

- **Fixed budget:** For a given horizon $N \in \mathbb{N}$, the learner chooses $\pi$ and $I$ to maximize the chance of predicting the correct hypothesis after exactly $N$ queries:

$$\sup_{\pi, I} \mathbb{P}^\pi(I_N(\mathcal{D}_N) = H^\star). \tag{2}$$

These two objectives capture the main operational modes of pure exploration: "stop when certain" and "maximize accuracy over a fixed horizon". Further note that the problem we propose to solve extends classical ASHT by allowing environment-specific, history-dependent observation kernels: $x_{t+1} \sim P_t(\cdot|\mathcal{D}_t, a_t)$. Standard formulations assume memoryless dependence only on $(H^\star, a_t)$([Naghshvar & Javidi, 2013](); [Garivier & Kaufmann, 2016]()). Moreover, whereas ASHT/BAI typically use known estimators (e.g., maximum likelihood), *we learn the inference rule from data*. Consequently, both the sampling policy $\pi$ and the inference rule $I$ can depend on entire histories.

## 3 ICPE: In-Context Pure Explorer

In this section we describe `ICPE`, a meta-RL approach for solving eqs. (1) and (2). The implementation of `ICPE` is motivated from the theory. We first show that learning an optimal inference rule $I$ amounts to computing a posterior distribution. Secondly, the policy $\pi$ can be learned using RL with an appropriate reward function.

Importantly, the reward function used for training $\pi$ *emerges* naturally from the problem formulation, and it *is not* a user-chosen criterion, making it a principled information-theoretical reward function. We now describe the theory, and then describe the practical implementation of `ICPE`.

### 3.1 Theoretical Results

Our theoretical results highlight that the main quantity of interest, in both regimes in eqs. (1) and (2), is the posterior distribution over the true hypothesis $\mathbb{P}(H^\star = H|\mathcal{D}_t)$. First, the *optimal inference rule $I^\star$ is based on this posterior*. Secondly, *this posterior naturally defines a reward function* that can characterize the optimality of a data-collection policy.

Throughout this section, we assume that $\mathcal{X} \subset \mathbb{R}$ is compact and $\mathcal{A}, \mathcal{H}$ are finite. We instantiate $\mathcal{M}$ via a parametrized family $\{(P_\omega, \rho_\omega) : \omega \in \Omega\}$ with $\Omega$ compact and $\omega \mapsto (P_\omega, \rho_\omega)$ continuous, so a prior on $\Omega$ induces a prior on $\mathcal{M}$. For the sake of brevity, we provide informal statements here, and refer the reader to app. B.1 for all the details.

We have the following result about the optimality of the inference, proved in app. B.1.2.

**Proposition 3.1** (Inference Rule Optimality). *Let $t \geq 1$ and a policy $\pi$. The optimal inference rule to $\sup_{I_t} \mathbb{P}^\pi(H^\star = I_t(\mathcal{D}_t))$ is given by $I_t^\star(z) = \arg\max_{H \in \mathcal{H}} \mathbb{P}(H^\star = H|\mathcal{D}_t = z)$.*

Concretely, prop. 3.1 identifies the optimal inference rule as the *maximum a posteriori* estimator based on $\mathbb{P}(H^\star = H | \mathcal{D}_t)$, so that learning $I_\phi$ amounts to learning this posterior. Based on this we can now differentiate between the two settings.

**Fixed budget.** We begin with the simpler fixed budget case. The key idea is to show that the optimal policy $\pi^\star$ maximizes an action-value function $Q$ (Sutton & Barto, 2018). First, define the following reward function: for $t < N$ let $r_t(\mathcal{D}_t) := 0$, and for $t = N$ set $r_N(\mathcal{D}_N) = \max_H \mathbb{P}(H^\star = H | \mathcal{D}_N)$. In words, we assign a reward equal to the maximum value of the posterior distribution at the last time step and 0 otherwise.

Then, define $V_N(\mathcal{D}_N) = r_N(\mathcal{D}_N)$ to be the optimal value at the last timestep $t = N$. From this definition, we can recursively define the $Q$-function as follows:

$$Q_t(\mathcal{D}_t, a) = \mathbb{E}_{x_{t+1}|(\mathcal{D}_t,a)}[V_{t+1}(\underbrace{(\mathcal{D}_t, a, x_{t+1})}_{=\mathcal{D}_{t+1}})] \quad \text{and} \quad V_t(\mathcal{D}_t) = \max_{a \in \mathcal{A}} Q_t(\mathcal{D}_t, a) \quad \forall t \leq N - 1.$$

where "$x_{t+1}|(\mathcal{D}_t, a)$" denotes the posterior distribution of $x_{t+1}$ given $(\mathcal{D}_t, a)$. Optimizing with respect to this reward function yields an optimal solution to (2), which we formalize in the following result proved in app. B.1.3.

**Theorem 3.2** (Policy Optimality for Fixed Budget). *For all $t \geq 1$, define the policy $\pi_t^\star(\mathcal{D}_t) = \arg\max_{a \in \mathcal{A}} Q_t(\mathcal{D}, a)$. Then, $(\pi^\star, I_N^\star)$ (where $I_N^\star$ is as in prop. 3.1) are an optimal solution of eq. (2), and we have that*

$$\sup_{\pi, I} \mathbb{P}^\pi (I_N(\mathcal{D}_N) = H^\star) = \mathbb{E}^{\pi^\star}[r_N(\mathcal{D}_N)]. \tag{3}$$

Simply speaking, thm. 3.2 indicates that an optimal exploration policy in the fixed-budget setting is obtained by a greedy policy with respect to a $Q$-function whose terminal reward is the maximum posterior mass $r_N(\mathcal{D}_N) = \max_H \mathbb{P}(H^\star = H \mid \mathcal{D}_N)$ (and zero reward for all other timesteps). A similar principle also holds for the fixed confidence setting.

**Fixed confidence.** In the fixed confidence setting, we first simplify the problem by noting that the stopping time $\tau$ can be simply embedded as a stopping action $a_{\text{stop}}$ in the policy $\pi$ (see app. B.1.4 for a formal justification). Hence, we extend the action set as $\mathcal{A} \leftarrow \mathcal{A} \cup \{a_{\text{stop}}\}$ and $\tau = \inf\{t \in \mathbb{N} : a_t = a_{\text{stop}}\}$. Then, as in classical ASHT literature (Naghshvar & Javidi, 2013), we study the dual problem of eq. (1), that is:

$$\inf_{\lambda \geq 0} \sup_{\pi, I} V_\lambda(\pi, I), \quad \text{where} \quad V_\lambda(\pi, I) := -\mathbb{E}^\pi[\tau] + \lambda \left[\mathbb{P}^\pi (I_\tau(\mathcal{D}_\tau) = H^\star) - 1 + \delta\right]. \tag{4}$$

To show optimality of a policy, and satisfaction of the correctness constraint, there are 2 key observations to make: (1) one can show that the optimal inference rule $I^\star$ remains as in prop. B.2; (2) solving eq. (4) amounts to solving an RL problem in $\pi$.

Indeed, similarly to the the fixed budget setting, for $t \geq 1$ define the reward model as

$$r_{t,\lambda}(\mathcal{D}_t, a) = -\mathbf{1}_{\{a \neq a_{\text{stop}}\}} + \lambda \mathbf{1}_{\{a = a_{\text{stop}}\}} \max_H \mathbb{P}(H^\star = H | \mathcal{D}_t), \tag{5}$$

which simply penalizes the policy for each extra timestep, accompanied by a reward proportional to the maximum posterior value at the stopping time. Accordingly, we define the $Q$-function as

$$Q_{t,\lambda}(\mathcal{D}_t, a) = r_{t,\lambda}(\mathcal{D}_t, a) + \mathbf{1}_{\{a \neq a_{\text{stop}}\}} \mathbb{E}_{x_{t+1}|(\mathcal{D}_t,a)} \left[\max_{a'} Q_{t+1,\lambda}((\mathcal{D}_t, a, x_{t+1}), a')\right]. \tag{6}$$

Then, we have the following result (see app. B.1.5 and app. B.1.6 for a proof) indicating that optimizing with respect to this reward function yields an optimal solution to (1).

**Theorem 3.3** (Policy Optimality for Fixed Confidence). *Let $\pi_{t,\lambda}^\star(\mathcal{D}_t) = \arg\max_{a \in \mathcal{A}} Q_{t,\lambda}(\mathcal{D}_t, a)$ and $\pi_\lambda^\star = (\pi_{t,\lambda})_t$. Then, for $\lambda \geq 0$ the pair $(I^\star, \pi_\lambda^\star)$, with $I^\star = (I_t^\star)_t$ defined as in prop. 3.1, is an optimal solution of $\sup_{\pi, I} V_\lambda(\pi, I)$. Furthermore, under suitable identifiability conditions (see assum. 2), any maximizer $\lambda^\star$ of eq. (4) guarantees that $\pi_{\lambda^\star}^\star$ satisfies the $\delta$-correctness criterion.*

Intuitively, for the fixed-confidence setting, we first recast the constrained problem in eq. (1) via a Lagrangian dual, then prove that *any* admissible stopping rule $\tau$ can be represented as the selection time of an absorbing stopping action $a_{\text{stop}}$. In thm. 3.3, we show that the resulting dual problem is

---

**Algorithm 1 `ICPE` (In-Context Pure Explorer)**

---

1: **Input:** Tasks distribution $\mathcal{P}$; confidence $\delta$; horizon $N$; initial $\lambda$ and hyper-parameter $T_\phi, T_\theta$.
     `// Training phase`
2: Initialize buffer $\mathcal{B}$, networks $Q_\theta, I_\phi$ and set $\bar{\theta} \leftarrow \theta, \bar{\phi} \leftarrow \phi$.
3: **while** Training is not over **do**
4:      Sample environment $M \sim \mathcal{P}$ with hypothesis $H^\star$, observe $x_1 \sim \rho$ and set $t \leftarrow 1$.
5:      **repeat**
6:          Execute action $a_t = \arg\max_a Q_\theta(\mathcal{D}_t, a)$ in $M$ and observe $x_{t+1}$.
7:          Add partial trajectory $(\mathcal{D}_t, a_t, x_{t+1}, H^\star)$ to $\mathcal{B}$ and set $t \leftarrow t + 1$.
8:      **until** $a_{t-1} = a_{\text{stop}}$ (fixed confidence) or $t > N$ (fixed budget).
9:      In the fixed confidence, update $\lambda$ according to eq. (11).
10:      Sample batch $B \sim \mathcal{B}$ and update $\theta, \phi$ using $\mathcal{L}_{\text{inf}}(B; \phi)$ (eq. (7)) and $\mathcal{L}_{\text{policy}}(B; \theta)$ (eq. (8) or eq. (9)).
11:      Every $T_\phi$ steps set $\bar{\phi} \leftarrow \phi$ (similarly, every $T_\theta$ steps set $\bar{\theta} \leftarrow \theta$).
12: **end while**

---

     `// Inference phase`
13: Sample unknown environment $M \sim \mathcal{P}$.
14: Collect a trajectory $\mathcal{D}_N$ (or $\mathcal{D}_\tau$ in fixed confidence) according to a policy $\pi_t(\mathcal{D}_t) = \arg\max_a Q_\theta(\mathcal{D}_t, a)$, until $t = N$ (or $a_t = a_{\text{stop}}$).
15: **Return** $\hat{H}_N = \arg\max_H I_\phi(H|\mathcal{D}_N)$ (or $\hat{H}_\tau = \arg\max_H I_\phi(H|\mathcal{D}_\tau)$ in the fixed confidence)

---

solved by a greedy policy on the $Q$-function defined via the reward in eq. (5), and that such policy achieves the desired level of correctness, $1 - \delta$. This result establishes that both an optimal $\delta$-aware stopping rule and exploration strategy can be learned on the extended action space $\mathcal{A} \cup \{a_{\text{stop}}\}$. In the next section, we describe the practical implementation of `ICPE` based on these results using the Transformer architecture.

### 3.2 Practical Implementation: the ICPE algorithm

We instantiate `ICPE` with two learners: an *inference network* $I_\phi(H|\mathcal{D}_t)$, parametrized by $\phi$, that approximates the posterior $\mathbb{P}(H^\star = H|\mathcal{D}_t)$ (cf. prop. 3.1) and a *Q-network* $Q_\theta(\mathcal{D}_t, a)$, parametrized by $\theta$, whose greedy policy defines $\pi_\theta$ (and includes $a_{\text{stop}}$ in the fixed confidence setting only). Both networks are implemented using Transformer architectures, and, for practical reasons, we impose a maximum trajectory length of $N$. This architecture handles both *fixed budget* (eq. (2)) and *fixed confidence* (eq. (1)) settings. However, we find it important to explicitly note that while algorithm 1 abstracts the main ideas of `ICPE` in a unified way, in practice we train separate models for the fixed-budget and fixed-confidence regimes, each with their own reward and $Q$-function as derived in section 3.1.

**Training phase.** At training time `ICPE` interacts with an online environment: each episode draws an instance $M \sim \mathcal{P}$ and generates a trajectory. We maintain a buffer $\mathcal{B}$ with tuples $(\mathcal{D}_t, a_t, x_{t+1}, H^\star_M)$, where $H^\star$ is the true hypothesis for the sampled environment $M$ (from a single tuple we also obtain $\mathcal{D}_{t+1} = (\mathcal{D}_t, a_t, x_{t+1})$). This buffer is used to sample mini-batches $B \subset \mathcal{B}$ to train $(\theta, \phi)$. Lastly, we treat each optimization in $(\phi, \theta)$ (and $\lambda$ too for the fixed confidence) separately, treating the other variables as fixed.

**Training of $I_\phi$.** We train $I_\phi$ to learn the posterior by SGD on the negative log-likelihood on a a batch $B \subset \mathcal{B}$ of partial trajectories sampled from the buffer:

$$\mathcal{L}_{\text{inf}}(\phi) = -\frac{1}{|B|} \sum_{(\mathcal{D}_t, a_t, x_{t+1}, H^\star) \in B} \log I_\phi(H^\star|\mathcal{D}_{t+1}). \tag{7}$$

In expectation this is (up to an additive constant) equivalent to minimizing the KL-divergence between $\mathbb{P}(H^\star = H|\mathcal{D})$ and $I_\phi(H|\mathcal{D})$ (a similar loss is also used in (Lee et al., 2023)). Lastly, we also set $\hat{H}_t = \arg\max_H I_\phi(H|\mathcal{D}_t)$ to be predicted hypothesis with data $\mathcal{D}_t$.

**Training in the Fixed Budget**. In the fixed budget we train $\theta$ using DQN (Mnih et al., 2015) and the rewards defined in the previous section. We denote the target network $Q_{\bar{\theta}}$, which is parameterized by $\bar{\theta}$. Since rewards are defined in terms of $I_\phi$, to improve training stability we introduce a separate target inference network $I_{\bar{\phi}}$, parameterized by $\bar{\phi}$, which provides feedback for training $\theta$. These target networks are periodically updated, setting $\bar{\phi} \leftarrow \phi$ every $T_\phi$ steps (similarly, $\bar{\theta} \leftarrow \theta$ every $T_\theta$ steps).

Hence, in the fixed budget, for a batch $B \sim \mathcal{B}$, we update $\theta$ by performing SGD on the following loss

$$\mathcal{L}_{\text{policy}}(B; \theta) = \frac{1}{|B|} \sum_{(\mathcal{D}_t, a_t, x_{t+1}) \in B} \left( \max_H I_{\bar{\phi}}(H|\mathcal{D}_t) \cdot \mathbf{1}_{\{t=N\}} + \max_a Q_{\bar{\theta}}(\mathcal{D}_{t+1}, a) - Q_\theta(\mathcal{D}_t, a_t) \right)^2 .$$
(8)

**Training in the Fixed Confidence**. In this setting we train $\theta$ similarly to the fixed budget setting. However, we also have a dedicated stop-action $a_{\text{stop}}$ whose value depends solely on history. Thus, its $Q$-value can be updated at any time, allowing retrospective evaluation of stopping. In other words, $Q_\theta(\mathcal{D}_t, a_{\text{stop}})$ can be updated for *any* sampled transition $z \in \mathcal{B}$, even if the logged action $a_t \neq a_{\text{stop}}$ (i.e., a "pretend to stop" update). This allows the model to retro-actively evaluate the quality of stopping earlier in a trajectory.

Then, based on eq. (6), we update $\theta$ by performing SGD on the following $Q$-loss

$$\mathcal{L}_{\text{policy}}(B; \theta) = \frac{1}{|B|} \sum_{(\mathcal{D}_t, a_t, x_{t+1}) \in B} \left[ \mathbf{1}_{\{a_t \neq a_{\text{stop}}\}} \cdot \left( -1 + \max_a Q_{\bar{\theta}}(\mathcal{D}_{t+1}, a) - Q_\theta(\mathcal{D}_t, a_t) \right)^2 \right. \quad (9)$$

$$\left. + \left( \lambda \max_H I_{\bar{\phi}}(H|\mathcal{D}_t) - Q_\theta(\mathcal{D}_t, a_{\text{stop}}) \right)^2 \right], \quad (10)$$

and note that the loss depends on $\lambda$. We learn $\lambda$ using a gradient descent update, which depends on the correctness of the predicted hypothesis. We sample $K$ trajectories $\{(\mathcal{D}_\tau^{(i)}, H_i^\star)\}_{i=1}^K$ with fixed $(\theta, \phi)$ and update $\lambda$ with a small learning rate $\beta$:

$$\lambda \leftarrow \max\left[ 0, \lambda - \beta \left( \hat{p} - 1 + \delta \right) \right], \quad \text{where} \quad \hat{p} = \frac{1}{K} \sum_{i=1}^K \mathbf{1}_{\{\arg\max_H I_\phi(H|\mathcal{D}_\tau^{(i)})\} = H_i^\star\}}. \quad (11)$$

The quantity $\hat{p}$ can be used to assess when to stop training by checking its empirical convergence. In the fixed confidence, in practice we can stop whenever $\hat{p} \geq 1 - \delta$ is stable and $\lambda$ is almost a constant. However, to obtain rigorous guarantees care must be taken. In app. B.1.6 we discuss how to provide formal guarantees on the $\delta$-correctness of the resulting method, bsaed on a sequential testing procedure.

**Inference phase.** At inference time `ICPE` operates by simple forwards passes. An unknown task $M \sim \mathcal{P}$ is sampled, and actions are selected according to $a = \arg\max_a Q_\theta(\mathcal{D}_t, a)$. At the last timestep a hypothesis is predicted using $\hat{H}_N = \arg\max_H I_\phi(H|\mathcal{D}_N)$ (or $\hat{H}_\tau = \arg\max_H I_\phi(H|\mathcal{D}_\tau)$ at the stopping time for the fixed confidence setting).

**Theoretical guarantees and training correctness.** In prop. B.12, we describe how to decide when to stop training in order to guarantee that the resulting $(\pi_\theta, I_\phi)$ are $\delta$-correct. Furthermore, we derive finite-sample guarantees for the fixed-budget `ICPE` meta-learning phase in a stylized setting in app. B.2. In thm. B.14 we derive a bound on the sub-optimality of the policy $\pi^{(k)}$ at training epoch $k$ in terms of stage-wise Bellman residuals and concentrability coefficients. In thm. B.15, we additionally show how these residuals are controlled by an approximation term (capturing how well the function class can represent the Bellman update) and an estimation term that decays with the number and size of training batches. Together, these results yield an explicit finite-sample performance bound for `ICPE` in an ideal scenario.

## 4 EMPIRICAL EVALUATION

We evaluate `ICPE` on a range of tasks: BAI on bandit problems, hypothesis testing in MDPs, and general search problems (pixel sampling and binary search). For bandits, we consider different reward structures: deterministic, stochastic, with feedback graphs and with hidden information. Due to space limitations, refer to app. D for the results on bandit problem with feedback graphs and MDPs. Also refer to app. C for details on the algorithms. In all experiments *we use a target accuracy value* of $\delta = 0.1$, and shaded areas indicate $95\%$ confidence intervals computed via hierarchical bootstrapping (see app. D for details).

## 4.1 BANDIT PROBLEMS

We now apply `ICPE` to the classical BAI problem within MAB tasks. For the MAB setting we have a finite number of actions $\mathcal{A} = \{1, \ldots, K\}$, corresponding to the actions in the MAB problem $M$. For each action $a$, we define a corresponding reward distribution $P(\cdot|a)$ from which rewards are sampled i.i.d. Then, $\mathcal{P}$ is a prior distribution on the actions' rewards distributions. For the BAI problem, we let the true hypothesis be $H^\star = \arg\max_a \mathbb{E}_{x \sim P(\cdot|a)}[x]$, so that the goal is to identify the best action (and thus $\mathcal{H} = \mathcal{A}$).

**Stochastic Bandit Problems.** We evaluate `ICPE` on stochastic bandit environments for both the fixed confidence and fixed budget setting (with $N = 30$). Each action's reward distribution is normally distributed $\nu_a = \mathcal{N}(\mu_a, 0.5^2)$, with $(\mu_a)_{a \in \mathcal{A}}$ drawn from $\mathcal{P}$. In this case $\mathcal{P}$ is a uniform distribution over problems with minimum gap $\max_a \mu_a - \max_{b \neq a} \mu_a \geq \Delta_0$, with $\Delta_0 = 0.4$. Hence, an algorithm could exploit this property to infer $H^\star$ more quickly. For this case, we also derive some sample complexity bounds in app. B.

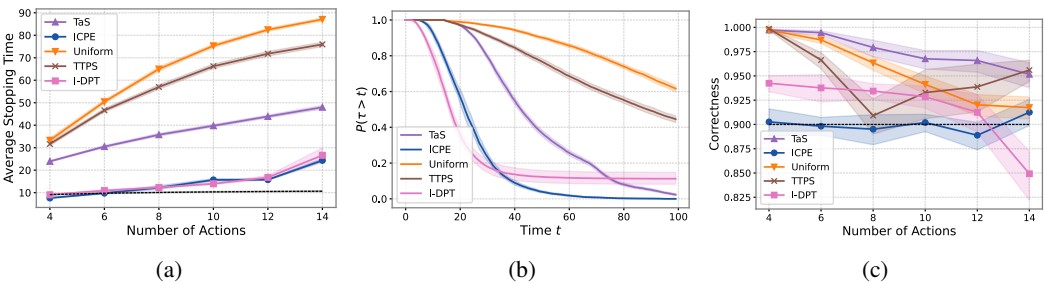

(a)          (b)          (c)

Figure 2: Results for stochastic MABs with fixed confidence $\delta = 0.1$ and $N = 100$: **(a)** average stopping time $\tau$; **(b)** survival function of $\tau$; **(c)** probability of correctness $\mathbb{P}^\pi(\hat{H}_\tau = H^\star)$.[3]

We compare against pure exploration baselines: **TaS** (Track-and-Stop) (Garivier & Kaufmann, 2016) and **TTPS** (Top-Two) (Russo et al., 2018), which are principled choices for hypothesis testing (asymptotically optimal or close to optimal allocations that target the most confusable hypotheses). We also include an ablation, "*I*-**DPT**", which uses our learned inference $I_\phi(H|D_t)$ as in DPT (Lee et al., 2023) and acts greedily with respect to the posterior (and a simple confidence-threshold stop); this isolates the value of learning a query policy versus relying on posterior-driven greedy control. Details for *I*-DPT are in app. C.

In fig. 2 are reported results for the fixed confidence. In fig. 2a we see how `ICPE` is able to find an efficient strategy compared to other techniques. Interestingly, also *I*-DPT seems to achieve relatively small sample complexities. However, the tail distribution of its $\tau$ is rather large compared to `ICPE` (fig. 2b) and the correctness is smaller than $1 - \delta$ for large values of $K$. Methods like TaS and TTPS achieve larger sample complexity, but also larger correctness values (fig. 2c). This is a well known fact: theoretically-sound stopping rules, such as the ones used by TaS and TTPS, tend to be are overly conservative (Garivier & Kaufmann, 2016).

Lastly, we verified the robustness of `ICPE` to distribution shifts. We trained `ICPE` in the stochastic fixed-confidence bandit setting as described above, and then evaluated the trained model on bandit instances drawn from *shifted* environment distributions. We report the results in app. D.1.2. Across all experiments, we observed that both correctness and stopping time remain remarkably stable, with only minor fluctuations within the reported confidence intervals. This suggests that ICPE is not excessively sensitive to moderate shifts in the environment distribution around the training family.

Finally, for the sake of space, we refer the reader to app. D.1.2 for the results in the fixed budget setting.

**Deterministic Bandits.** We also evaluated `ICPE` in deterministic bandit environments with a fixed budget $K$, equal to the number of actions. Thus, `ICPE` needs to learn to select each action only once to determine the optimal action. Since the rewards are deterministic, we cannot compare to classical BAI methods, which are tailored for stochastic environments. Instead, we compare to: (i) a uniform policy that uses a maximum likelihood estimator to estimate the best arm; (ii) **DQN** (Mnih et al., 2013), which uses $\mathcal{D}_t$ as the state, and trains an $I$ network to infer the true hypothesis; (iii) and *I*-**DPT**, acts greedily with respect to the posterior of $I_\phi$, as in DPT. Figure 3

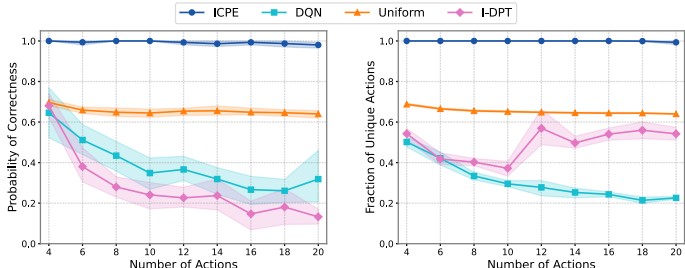

Figure 3: Deterministic bandits: (left) probability of correctly identifying the best action vs. $K$; (right) average fraction of unique actions selected during exploration vs. $K$.

reports the results: **ICPE** consistently identifies optimal actions (correctness $\approx 1$) and learns optimal sampling strategies (fraction of unique actions $\approx 1$). Without being explicitly instructed to "choose each action exactly once", **ICPE** discovers on its own that sampling every action is exactly what yields enough information to identify the best. While the optimal exploration strategy in this setting is intuitive, baseline performance degrades sharply as the number of actions grows, illustrating that existing exploration methods can fail even in such simple environments.

**Bandit Problems with Hidden Information.** To evaluate **ICPE** in structured settings, we introduce bandit environments with latent informational dependencies, termed *magic actions*. In the single magic action case, the magic action $a_m$'s reward is distributed according to $\mathcal{N}(\mu_{a_m}, \sigma_m^2)$, where $\sigma_m \in (0, 1)$ and $\mu_{a_m} := \phi(\arg\max_{a \neq a_m} \mu_a)$ encodes information about the optimal action's identity through an invertible mapping $\phi$ that is unknown to the learner. The index $a_m$ is fixed, and the mean rewards of the other actions $(\mu_a)_{a \neq a_m}$ are sampled from $\mathcal{P}$, a uniform distribution over models guaranteeing that $a_m$, as defined above, is not optimal (see apps. B.5 and D.1.3 for more details). Then, we define the reward distribution of the non-magic actions as $\mathcal{N}(\mu_a, (1 - \sigma_m)^2)$.

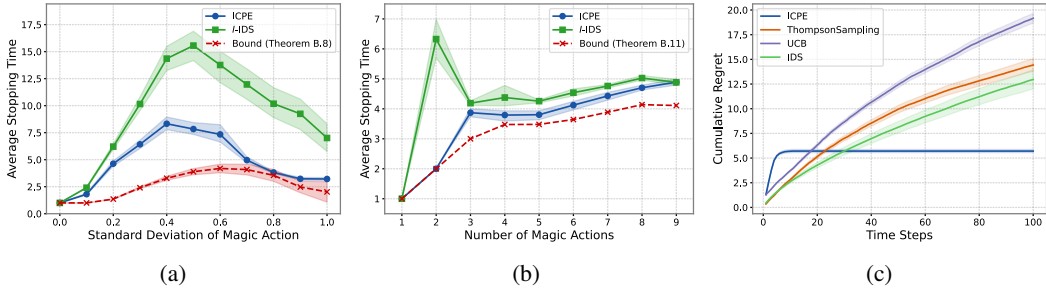

(a)             (b)             (c)

Figure 4: **(a)** Single magic action: average stopping time and the theoretical lower bound across varying $\sigma_m$. **(b)** Magic chain: average stopping time between **ICPE**, $I$-IDS vs. number of magic actions. **(c)** **ICPE** in a regret minimization task, with $\sigma_m = 0.1$.

In our first experiment, we vary the standard deviation $\sigma_m$ in $[0, 1]$. This problem isolates whether **ICPE** can detect and exploit latent informational dependencies (via a single diagnostic action that encodes the optimal arm) and balance sampling across action based on varying uncertainty levels.

Regarding the baselines, applying classical baselines (e.g., TaS) here is nontrivial: the magic action is coupled to the optimal arm via an *unknown* map $\phi$, which would need to be encoded as inductive bias. Instead, we compare **ICPE** to "$I$-**IDS**", which is standard pure exploration IDS (Russo & Van Roy, 2018) instantiated on top of ICPE's trained inference $I_\phi$ for exploiting the magic action.

We evaluate in a fixed-confidence setting with error rate $\delta = 0.1$. Figure 4a compares **ICPE**'s sample complexity against a theoretical lower bound (see app. B). **ICPE** achieves sample complexities close to the theoretical bound across all tested noise levels, consistently outperforming $I$-IDS.

Additionally, in fig. 4c we evaluate **ICPE** in a cumulative regret minimization setting, despite not being explicitly optimized for regret minimization. At the stopping $\tau$, **ICPE** commits to the identified best action (i.e., explore-then-commit strategy). As shown in the results, **ICPE** outperforms classic algorithms such as UCB, Thompson Sampling, and standard IDS initialized with Gaussian priors.

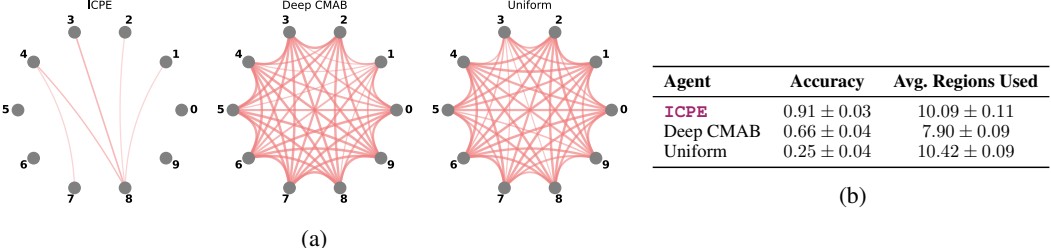

(a)

| Agent | Accuracy | Avg. Regions Used |
|---|---|---|
| ICPE | $0.91 \pm 0.03$ | $10.09 \pm 0.11$ |
| Deep CMAB | $0.66 \pm 0.04$ | $7.90 \pm 0.09$ |
| Uniform | $0.25 \pm 0.04$ | $10.42 \pm 0.09$ |

(b)

Figure 5: MNIST pixel-sampling task: **(a)** A chord between two digits indicates that their distributions were not significantly different ($p$-value $> 0.05$, based on a pairwise chi-squared test), with thicker chords representing higher $p$-values; **(b)** accuracy and performance (mean $\pm$ 95% CI)

To further challenge **ICPE**, we introduce a *multi-layered "magic chain" bandit* environment, where there is a sequence of $n$ magic actions $\mathcal{A}_m := \{a_{i_1}, \ldots, a_{i_n}\} \subset \mathcal{A}$ such that $\mu_{a_{i_j}} = \phi(\mu_{a_{i_{j+1}}})$, and $\mu_{a_{i_n}} = \phi(\arg\max_{a \notin \mathcal{A}_m} \mu_a)$. The first index $i_1$ is known, and by following the chain, an agent can uncover the best action in $n$ steps. However, the optimal sample complexity depends on the ratio of magic actions to non-magic arms. Varying the number of magic actions from 1 to 9 in a 10-actions environment, Figure 4b demonstrates **ICPE**'s empirical performance, outperforming $I$-IDS.

## 4.2 General Search Problems: Pixel Sampling and Probabilistic Binary Search

We now evaluate the applicability of **ICPE** to general search problems, including structured real-world examples.

**Pixel sampling as generalized search.** We introduce a classification task inspired by active perception settings. We consider the MNIST images (LeCun et al., 1998), each partitioned into a set of 36 distinct pixel patches, corresponding to the query space $\mathcal{A} = \{1, \ldots, 36\}$. The agent starts from a blank (masked) image and, patch by patch, reveals pixels to quickly discover "what the image is about." After choosing a query $a_t \in \mathcal{A}$ the agent observes $x_t$ (the revealed patch) and accumulates a partially observed image. After a budget $N = 12$, the agent outputs the predicted digit $\hat{H}_N \in \{0, \ldots, 9\}$.

For this setting we consider a slight variation of **ICPE** that may be of interest: we consider an inference net $I$ that is a pre-trained classifier, trained on fully revealed images from $\mathcal{P}$. Using this network, we benchmark **ICPE** against two baselines: standard uniform random sampling and Deep Contextual Multi-Armed Bandit (**Deep CMAB**) (Collier & Llorens, 2018), which employs Bayesian neural networks to sample from a posterior distribution (Deep CMAB uses as rewards the correctness probabilities computed by $I$). Importantly, we cannot compare to methods such as DPT since $\mathcal{A} \neq \mathcal{H}$, the hypothesis space is different from the query space.

Table 5b reports the classification accuracy and number of regions sampled. **ICPE** achieves substantially better performance than both baselines using fewer regions. However, to analyze whether **ICPE** learns a sampling strategy that adapts to the context of the task, we compare region selection distributions across digit classes using pairwise chi-squared tests. **ICPE** exhibits significantly more variation across classes than either baseline, as visualized in Figure 5a. This suggests **ICPE** adapts its exploration to class-conditional structure, rather than applying a generic sampling policy.

| $K$ | Min Accuracy | Mean Stop Time | Max Stop Time | $\log_2 K$ |
|---|---|---|---|---|
| 8 | 1.00 | $2.13 \pm 0.12$ | 3 | 3 |
| 16 | 1.00 | $2.93 \pm 0.12$ | 4 | 4 |
| 32 | 1.00 | $3.71 \pm 0.15$ | 5 | 5 |
| 64 | 1.00 | $4.50 \pm 0.21$ | 6 | 6 |
| 128 | 1.00 | $5.49 \pm 0.23$ | 7 | 7 |
| 256 | 1.00 | $6.61 \pm 0.26$ | 8 | 8 |

Table 1: **ICPE** performance on the binary search task as $K$ increases.

**Probabilistic binary search.** We also evaluated **ICPE**'s capabilities to autonomously meta-learn binary search. We define an action space of $\mathcal{A} = \{1, \ldots, K\}$, with $H^\star \in \mathcal{A}$. Pulling an arm above or below $H^\star$ yields a observation $x_t = -1$ or $x_t = +1$, respectively, providing directional feedback. In tab. 1 we report results on 100 held-out tasks per setting. **ICPE** consistently achieves perfect

accuracy with worst-case stopping times that match the optimal $\log_2(K)$ rate, demonstrating that it has successfully learned binary search. While simple, this task illustrates `ICPE`'s broader potential to learn efficient search strategies in domains where no hand-designed algorithm is available.

## 5    DISCUSSION AND CONCLUSIONS

Our results position `ICPE` within a broader line of work on *active sequential hypothesis testing* (Naghshvar & Javidi, 2013) and its close ties to exploration in RL (Sutton & Barto, 2018). Regarding exploration, note that classical regret-minimization methods, including UCB variants (Auer et al., 2002), posterior sampling (Osband et al., 2013; Russo & Van Roy, 2014), and regret-focused IDS (Russo et al., 2018), optimize long-run reward, not hypothesis identification. On the other hand, pure-exploration formulations in BAI (Audibert & Bubeck, 2010) yield sharp, instance-dependent procedures for hypothesis testing in fixed-confidence regimes (e.g., Track-and-Stop, Garivier & Kaufmann, 2016). However, these approaches assume to know the problem structure, which is not always possible if the user is not aware of such structure. Furthermore, computing an optimal data-collection policy remains a challenge in more general scenarios (Al Marjani et al., 2021), and we discuss some of these challenges in app. C.3.1.

`ICPE` uses Transformers (Vaswani et al., 2017) to learn, in-context, a data collection policy and and inference rule. Transformers have demonstrated remarkable in-context learning capabilities (Brown et al., 2020; Garg et al., 2022). In-context learning (Moeini et al., 2025) is a form of meta-RL (Beck et al., 2023), where agents can solve new tasks without updating any parameters by simply conditioning on histories. Building on this approach, Transformers can mimic posterior sampling from offline data, as in DPT (Lee et al., 2023), or perform return-conditioning for regret minimization (e.g., ICEE Dai et al., 2024). However, these approaches primarily target cumulative reward and typically lack a learned, $\delta$-aware stopping rule; applying them to hypothesis testing would require altering objectives, data-collection protocol, and add stopping semantics. Moreover, in generalized search where $\mathcal{A} \neq \mathcal{H}$, additional modeling is needed to map hypotheses to actions.

`ICPE` addresses these gaps by *learning* to acquire information in-context. `ICPE` targets *pure exploration for identification*: it splits inference and control, using a supervised inference network to provide task-relevant information signals, while an RL-trained Transformer learns acquisition policies that maximize information gain. This separation makes it possible to exploit rich, non-tabular structures that are difficult to encode in hand-designed tests or confidence bounds.

Empirically, `ICPE` is competitive on unstructured bandits and extends naturally to structured and deterministic settings. The results on the MNIST dataset highlights a key strength: `ICPE` adapts sampling to the class-conditional structure. More broadly, `ICPE` suggests a path for *data-driven generalized search*.

Limitations point to concrete avenues for future work. First, scaling to continuous or combinatorial hypothesis spaces to deal with more general scenarios is an important direction. However, such extensions require substantial further theoretical development, as rigorous formalisms for continuous hypothesis-testing frameworks remain an active area of research, even in classical pure-exploration settings (see, e.g., (Garivier & Kaufmann, 2021)). Second, extending `ICPE` to offline datasets is also a promising research direction. When offline data can be used to construct a reliable simulator, `ICPE` can already be applied directly. Moreover, even without such a simulator, `ICPE` could in principle be meta-trained purely from logged data using offline RL methods (e.g., IQL, CQL), and a systematic study of this offline regime is an important question for future work. Third, while the main focus of this work is to introduce and analyze `ICPE` as a general framework that can address a broad family of pure exploration problems, and we validate it on numerous BAI and active search tasks, we view real-world experiments as a natural next step. `ICPE` holds the promise to discover novel exploration and search algorithms in complex domains that do not offer a concrete way of finding an optimal solution a priori, such as determining efficient sequences of proteins to test in a lab (Amin et al., 2024), minimizing the number of tests required to detect cancer (Gan et al., 2021), and expediting the design of materials with desired properties (Talapatra et al., 2018). In sum, we believe `ICPE` advances pure exploration by leveraging in-context learning to discover task-adaptive acquisition strategies, and it opens a route toward unifying classical sequential testing with learned, structure-aware search policies that scale to real problems.

## REPRODUCIBILITY STATEMENT

We have taken several measures to ensure the reproducibility of our results. All model architectures, optimization procedures, and hyperparameters are described in detail in the paper (see Sections 2–3 and Appendix C–D). Experiments were conducted using Python 3.10.12 and standard libraries including NumPy, SciPy, PyTorch, Pandas, Seaborn, Matplotlib, CVXPY, and Gurobi.

To facilitate replication, we provide our full source code at https://github.com/rssalessio/icpe under the MIT license. The code contains (i) implementations of ICPE and all baselines, (ii) configuration files specifying the hyperparameters for each experiment, and (iii) detailed instructions in the README.md file for installing dependencies and running all experiments. Running the provided scripts will reproduce the main results reported in the paper, including bandit, MDP, and generalized search benchmarks.

## ACKNOWLEDGMENTS

The authors are pleased to acknowledge that the computational work reported on in this paper was performed on the Shared Computing Cluster administered by Boston University's Research Computing Services and computing resources from the Laboratory for Information and Decision Systems at MIT. R.W. was supported by a Master of Engineering fellowship by the Eric and Wendy Schmidt Center at the Broad Institute.

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
