# Appendix

CONTENTS

APPENDIX

LIMITATIONS AND BROADER IMPACT

**Finite vs continuous sets of hypotheses.** A limitation of this work is the assumption that $\mathcal{H}$ is finite. This is a common assumption in active sequential hypothesis testing, and the continuous case is also referred to as *active regression* (Mukherjee et al., 2022). We believe our framework can be extended to this case with a proper parametrization of the inference mapping $I$ that allows to sample from a continuous set.

**On the prior set of problems $\mathcal{P}$.** One limitation of our approach is the assumption of access to a prior set of problems $\mathcal{P}$. Such set may lack a common structure, and need not be stationary. Nonetheless, we view this as a useful starting point for developing more sophisticated methods. A natural direction for future work is to extend our framework to an adversarial setting, in which problem instances can evolve or even be chosen to thwart the learner.

**Online training.** Another limitation arises from assuming access to an online simulator from which we can sample $M \sim \mathcal{P}$ and training `ICPE`. Implicitly, this assumes access to $H^\star$ during training. Learning how to generalize to setting where $H^\star$ is not perfectly known at training time is an exciting research direction. Furthermore, our main focus is in the sequential process of starting from "no data", to being able to predict the right hypothesis as quickly as possible (see the MNIST example). We believe this framework to be valuable when one can build verifiable simulations to train policies that transfer to real-world problems.

**Practical limitations and transformers.** A limitation of `ICPE` is the current limit $N$ on the horizon of the trajectory. This is due to the computation cost of training and using transformer architectures. Future work could investigate how to extend this limit, or completely remove it.

Another technical limitation of `ICPE` is the hardness to scaling to larger problems. This is closely related to the above limitation, and it is mainly an issue of investigating how to improve the current architecture of `ICPE` and/or distribute training.

Lastly, we believe that `ICPE` does not use the full capabilities of transformer architectures. For example, during training and evaluation, we always use the last hidden state of the transformer to make prediction, while the other hidden states are left untouched.

**Bayesian BAI.** Some of our work falls within the Bayesian Best Arm Identification theoretical framework. However, the Bayesian setting is less known compared to the frequentist one, and only recently some work (Jang et al., 2024) studied the unstructured Gaussian case. Future work should compare `ICPE` more thoroughly with Bayesian techniques once the Bayesian setting is more developed.

**Broader impact.** This paper primarily focuses on foundational research in pure exploration problems. Although we do not directly address societal impacts, we recognize their importance. The methods proposed here improve the sample efficiency of active sequential hypothesis testing procedures, and could be applied in various contexts with societal implications. For instance, our technique could be used in decision-making systems in healthcare, finance, and autonomous vehicles, where biases or errors could have significant consequences. Therefore, while the immediate societal impact of our work may not be evident, we urge future researchers and practitioners to carefully consider the ethical implications and potential negative impacts in their specific applications

## A  EXTENDED RELATED WORK

**Exploration for Regret Minimization.**   The problem of exploration is particular relevant in RL (Sutton & Barto, 2018), and many strategies have been introduced, often with the goal of minimizing regret. Notably, approaches based on Posterior Sampling (Kaufmann et al., 2012; Osband et al., 2013; Russo & Van Roy, 2014; Gopalan et al., 2014) and Upper Confidence Bounds (Auer et al., 2002; 2008; Cappé et al., 2013; Lattimore & Hutter, 2012; Auer, 2002) have received significant attention. However, the problem of minimizing regret is a relevant objective only when one cares about the rewards accumulated so far, and does not answer the problem of how to efficiently gather data to reach some desired goal. In this context, *Information-Directed Sampling* (IDS) (Russo & Van Roy, 2014; Russo et al., 2018) has been proposed to strike a balance between minimizing regret and maximizing information gain, where the latter is quantified as the mutual information between the true optimal action and the subsequent observation. However, when the information structure is unknown, it effectively becomes a significant challenge to exploit it. Importantly, if the state does not encode the structure of the problem, RL techniques may not be able to exploit hidden information.

**In-Context Learning, LLMs and Return Conditioned Learning.**   Recently, Transformers (Vaswani et al., 2017; Chen et al., 2021) have demonstrated remarkable in-context learning capabilities (Brown et al., 2020; Garg et al., 2022). In-context learning (Moeini et al., 2025) is a form of meta-RL (Beck et al., 2023), where agents can solve new tasks without updating any parameters by simply conditioning on additional context such as their action-observation histories. When provided with a few supervised input-output examples, a pretrained model can predict the most likely next token (Lee et al., 2023). Building on this ability, Lee et al. (2023) recently showed that Transformers can be trained in a supervised manner using offline data to mimic posterior sampling in reinforcement learning. In (Krishnamurthy et al., 2024) the authors investigate the extent to which LLMs (Achiam et al., 2023) can perform in-context exploration in multi-armed bandit problems. Similarly, other works (Coda-Forno et al., 2023; Monea et al., 2024; Nie et al., 2024; Harris & Slivkins, 2025; Sun et al., 2025) evaluate the in-context learning capabilities of LLMs in sequential decision making problems, with (Harris & Slivkins, 2025) showing that LLMs can help at exploring large action spaces with inherent semantics. On a different note, in (Arumugam & Griffiths, 2025) investigate how to use LLMs to implement PSRL, leveraging the full expressivity and fluidity of natural language to express the prior and current knowledge about the problem.

In (Dai et al., 2024) the authors presente ICEE (In-Context Exploration Exploitation), a method closely related to `ICPE`. ICEE uses Transformer architectures to perform in-context exploration-exploration for RL. ICEE tackles this challenge by expanding the framework of return conditioned RL with in-context learning (Chen et al., 2021; Emmons et al., 2021). Return conditioned learning is a type of technique where the agent learns the return-conditional distribution of actions in each state. Actions are then sampled from the distribution of actions that receive high return. This methodoloy was first proposed for the online RL setting by work on Upside Down RL (Srivastava et al., 2019) and Reward Conditioned Policies (Kumar et al., 2019). Lastly, we note the important contribution of $RL^2$ (Duan et al., 2016), which proposes to represent an RL policy as the hidden state of an RNN, whose weights are learned via RL. `ICPE` employs a similar idea, but focuses on a different objective (identification), and splits the process into a supervised inference network that provides rewards to an RL-trained transformer network that selects actions to maximize information gain.

**Active Pure Exploration in Bandit and RL Problems.**   Other strategies consider the *pure exploration problem* (Even-Dar et al., 2006; Audibert & Bubeck, 2010; Bubeck et al., 2011; Kaufmann et al., 2016), or Best Arm Identification (BAI), in which the samples collected by the agent are no longer perceived as rewards, and the agent must actively optimize its exploration strategy to identify the optimal action. In this pure exploration framework, the task is typically formulated as a hypothesis testing problem: given a desired goal, the agent must reject the hypothesis that the observed data could have been generated by any environment whose behavior is fundamentally inconsistent with the true environment (Garivier & Kaufmann, 2016). This approach leads to instance-dependent exploration strategies that adapt to the difficulty of the environment and has been extensively studied in the context of bandit problems under the fixed confidence setting (Even-Dar et al., 2006; Garivier & Kaufmann, 2016; Degenne et al., 2019; Jang et al., 2024; Russo & Proutiere, 2023b; Shukla & Basu, 2024; Carlsson et al., 2024; Karthik et al., 2024; Russo & Vannella, 2024; Poiani et al., 2025; Russo et al., 2025), where the objective is to identify the optimal policy using the fewest number

of samples while maintaining a specified level of confidence. Similar ideas have been applied to Markov Decision Processes for identifying the best policy (Marjani & Proutiere, 2021; Marjani et al., 2021; Russo & Proutiere, 2023a; Tuynman et al., 2024; Russo & Vannella, 2025), policy testing (Ariu et al., 2025), or rapidly estimating the value of a given policy (Russo & Pacchiano, 2025). Another setting is that of of identifying the best arm in MAB problems with a fixed horizon. In this case characterizing the complexity of the problem is challenging, and this is an area of work that is less developed compared to the fixed confidence one (Wang et al., 2023; Karnin et al., 2013; Audibert & Bubeck, 2010; Atsidakou et al., 2022; Nguyen et al., 2025; Ghosh et al., 2024). Because of this reason, we believe `ICPE` can help better understand the nuances of this specific setting.

However, while BAI strategy are powerful, they may be suboptimal when the underlying information structure is not adequately captured within the hypothesis testing framework. Hence, the issue of leveraging hidden environmental information, or problem with complex information structure remains a difficult problem. Although IDS and BAI techniques offer frameworks to account for such structure, extending these approaches to Deep Learning is difficult, particularly when the information structure is unknown to the learner.

A closely related work is that of (Liu et al., 2024). In (Liu et al., 2024) the authors present empirical evidence of skills and directed exploration emerging from using RL with a sparse reward and a contrastive loss. They define a goal state, and encode a sparse reward using that goal state. Their objective, which maximizes the probability of reaching the goal state, is similar to ours, where in our framework the goal state would be a hypothesis. Note, however, that they do not learn an inference network, and we do not assume the observations to possess the Markov property.

**Active Learning and Active Sequential Hypothesis Testing** In the problem of active sequential hypothesis testing (Chernoff, 1992; Ghosh, 1991; Lindley, 1956; Naghshvar & Javidi, 2013; Naghshvar et al., 2012; Mukherjee et al., 2022; Gan et al., 2021), a learner is tasked with adaptively performing a sequence of actions to identify an unknown property of the environment. Each action yields noisy feedback about the true hypothesis, and the goal is to minimize the number of samples required to make a confident and correct decision. Similarly, active learning (Cohn et al., 1996; Chen et al., 2023) studies the problem of data selection, and, closely related, Bayesian Active Learning (Golovin & Krause, 2011), or Bayesian experimental design (Rainforth et al., 2024), studies how to adaptively select from a number of expensive tests in order to identify an unknown hypothesis sampled from a known prior distribution.

Active sequential hypothesis testing generalizes the pure exploration setting in bandits and RL by allowing for the identification of arbitrary hypotheses, rather than just the optimal action. However, most existing approaches assume full knowledge of the observation model (Naghshvar & Javidi, 2013), which is the distribution of responses for each action under each hypothesis. While some work has attempted to relax this assumption to partial knowledge (Cecchi & Hegde, 2017), it remains highly restrictive in practice. As in bandit settings, real-world exploration and hypothesis testing often proceed without access to the true observation model, requiring strategies that can learn both the structure and the hypothesis from interaction alone.

**Algorithm Discovery.** Our method is also closely related to the problem of discovering algorithms (Oh et al., 2020). In fact, one can argue that `ICPE` is effectively discovering active sampling techniques. This is particularly important for BAI and Best Policy Identification (BPI) problems, where often one needs to solve a computationally expensive optimization technique numerous times. For BPI the problem is even more exacerbated, since the optimization problem is usually non-convex (Marjani & Proutiere, 2021; Russo & Pacchiano, 2025).

**Cognitive Theories of Exploration.** Our approach draws inspiration from cognitive theories of exploration. Indeed, in animals, exploration arises naturally from detecting mismatches between sensory experiences and internal cognitive maps—mental representations encoding episodes and regularities within environments (O'keefe & Nadel, 1979; Nadel & Peterson, 2013). Detection of novelty prompts updates of these cognitive maps, a function strongly associated with the hippocampus (Nadel, 1991; Lisman et al., 2017). Conversely, exploration can also be explicitly goal-directed: psychological theories posit that an internal representation of goals, combined with cognitive maps formed through experience, guides adaptive action selection (Kagan, 1972; Morris et al., 2022). `ICPE` embodies these cognitive principles computationally: the exploration ($\pi$) network learns

an internal model (analogous to a cognitive map), while the inference ($I$) network encodes goal-directed evaluation. This interplay enables `ICPE` to effectively manage exploration as an adaptive, structure-sensitive behavior.

# B    THEORETICAL RESULTS

In this section we provide different theoretical results: first, we describe the theoretical results for `ICPE`. Then, we discuss some sample complexity results for different MAB problems with structure.

## B.1    ICPE: THEORETICAL RESULTS

In this subsection we present the theoretical results of `ICPE`. We begin by describing the problem setup. After that, we present results for the fixed budget and fixed confidence regimes respectively.

### B.1.1    PROBLEM SETUP

We now provide a formal definition of the underlying probability measures of the problem we consider. To that aim, it is important to formally define what a model $M$ is, as well as the definition of policy $\pi$ and inference rule $I$ (infernece rules are also known as recommendation rules).

**Spaces and $\sigma$-fields.**    We let $\mathcal{X} \subset \mathbb{R}$ be nonempty, compact, and endowed with its Borel $\sigma$-field $\mathcal{B}(\mathcal{X})$. Let $\mathcal{A} = \{1, \dots, K\}$ be a finite action set with the discrete $\sigma$-field $2^{\mathcal{A}}$, and let $\mathcal{H}$ be a finite hypothesis set with the discrete $\sigma$-field. Write $\Delta(\mathcal{X})$ for the set of Borel probability measures on $(\mathcal{X}, \mathcal{B}(\mathcal{X}))$, equipped with the topology of weak convergence and its Borel $\sigma$-field $\mathcal{B}(\Delta(\mathcal{X}))$.

For $t \in \mathbb{N}$, define the trajectory space

$$\mathcal{Z}_t := (\mathcal{X} \times \mathcal{A})^{t-1} \times \mathcal{X}, \qquad z_t = (x_1, a_1, \dots, a_{t-1}, x_t),$$

with the product topology and Borel $\sigma$-field $\mathcal{B}(\mathcal{Z}_t)$. Since $\mathcal{X}$ is compact metric and $\mathcal{A}$ is finite, each $(\mathcal{Z}_t, \mathcal{B}(\mathcal{Z}_t))$ is standard Borel (in fact compact Polish). Set $\mathcal{Z}_\infty := \mathcal{X} \times (\mathcal{A} \times \mathcal{X})^{\mathbb{N}}$ with the product $\sigma$-field $\mathcal{B}(\mathcal{Z}_\infty)$.

**Observation dynamics and parameterization.**    To define the dynamics $(\rho, P)$ we introduce a parametrization in $\omega \in \Omega$. We take $(\Omega, d)$ to be a compact metric with Borel $\sigma$-field $\mathcal{B}(\Omega)$ and metric $d$. For each $\omega \in \Omega$, we assume that $\rho$ and $P$ are functionals of $\omega$:

- $\rho_\omega \in \Delta(\mathcal{X})$ is the initial observation law (a Borel probability measure on $\mathcal{X}$).
- $P_s^\omega(\cdot | z_s, a_s) \in \Delta(\mathcal{X})$ is a Borel probability *kernel* for each round $s \geq 1$: for every $(z_s, a_s)$, $P_s^\omega(\cdot | z_s, a_s)$ is a probability measure on $(\mathcal{X}, \mathcal{B}(\mathcal{X}))$, and for every $C \in \mathcal{B}(\mathcal{X})$ the map $(z_s, a_s) \mapsto P_s^\omega(C | z_s, a_s)$ is measurable.

We assume the following *weak continuity* in $\omega$: for every bounded continuous $f : \mathcal{X} \to \mathbb{R}$,

$$\omega \mapsto \int f \, d\rho_\omega \quad \text{and} \quad (\omega, z_s, a_s) \mapsto \int f(x') \, P_s^\omega(dx' | z_s, a_s) \quad \text{are continuous.}$$

Equivalently, $\omega \mapsto \rho_\omega$ and $(\omega, z_s, a_s) \mapsto P_s^\omega(\cdot | z_s, a_s)$ are continuous maps into $\Delta(\mathcal{X})$ with the weak topology.

**Set of models (environments).**    To define the set of models, consider a mapping $\phi : \Omega \to \Delta(\mathcal{X}) \times \prod_{s \geq 1} \left( \Delta(\mathcal{X})^{\mathcal{Z}_s \times \mathcal{A}} \right)$ so that $\phi(\omega) = (\rho_\omega, P^\omega)$, where $P^\omega := (P_s^\omega)_{s \geq 1}$. Set $\mathcal{M}^\sharp := \phi(\Omega)$ with the product of weak topologies. We indicate by $(\rho, P) \in \mathcal{M}^\sharp$ a model in this set (hence $P = P^\omega$ for some $\omega$). By continuity of $\phi$ and compactness of $\Omega$, $\mathcal{M}^\sharp$ is compact.

We also let $h^\star : \Delta(\mathcal{X}) \times \prod_{s \geq 1} \left( \Delta(\mathcal{X})^{\mathcal{Z}_s \times \mathcal{A}} \right) \to \mathcal{H}$, with $\mathcal{H}$ finite (with the discrete topology), to be a Borel measurable mapping defining the ground truth hypothesis $H^\star = h^\star(\rho, P)$ for a pair $(\rho, P) \in \mathcal{M}^\sharp$. Then, we define $\mathcal{M}$ as

$$\mathcal{M} := \left\{ (\rho, P, h^\star(\rho, P)) : (\rho, P) \in \mathcal{M}^\sharp \right\}$$

to be the push-forward set of environments[4]. Therefore, a prior $Q$ on $\Omega$ induces a prior on $\mathcal{M}$ (and $\mathcal{M}^\sharp$) by pushforward: $\mathcal{P} := Q \circ (\omega \mapsto (\phi(\omega), h^\star(\phi(\omega))))^{-1}$ and $\mathcal{P}^\sharp := Q \circ \phi^{-1}$. In the following, we mainly work with $\mathcal{M}^\sharp$ and use $\mathcal{P}$ and $\mathcal{P}^\sharp$ interchangeably whenever clear from the context.

---

[4] We omit $\mathcal{X}, \mathcal{A}$ from the definition since these sets are the same for all models in $\mathcal{M}^\sharp$.

**Policies and inference (recommendation) rules.** A (possibly randomized) policy $\pi = (\pi_s)_{s \geq 1}$ is a sequence of Borel probability kernels $\pi_s(\cdot | z_s) \in \Delta(\mathcal{A})$, $s \geq 1$, with $\pi_s : (\mathcal{Z}_s, \mathcal{B}(\mathcal{Z}_s)) \to (\Delta(\mathcal{A}), \mathcal{B}(\Delta(\mathcal{A})))$ measurable. Deterministic policies are the special case $\pi_s(\cdot | z_s) = \delta_{\alpha_s(z_s)}$ for some measurable $\alpha_s : \mathcal{Z}_s \to \mathcal{A}$. An inference rule at timestep $t$ is defined as any Borel map $I_t : \mathcal{Z}_t \to \mathcal{H}$. We also define an inference rule as the collection $I := (I_s)_{s \geq 1}$.

**Path laws and probability measures.** Fix $M \in M^\sharp$, with $M = (\rho, (P_s)_s)$, and a policy $\pi = (\pi_s)_{s \geq 1}$. By the Ionescu–Tulcea theorem, there exists a unique probability measure $\mathbb{P}^\pi_{M,t}$ on $(\mathcal{Z}_t, \mathcal{B}(\mathcal{Z}_t))$ such that for all cylinder sets $C = C_1 \times B_1 \times \cdots \times B_{t-1} \times C_t$, with $C_i \in \mathcal{B}(\mathcal{X})$ and $B_i \subset \mathcal{A}$,

$$\mathbb{P}^\pi_{M,t}(C) = \int_{C_1} \rho(\mathrm{d}x_1) \prod_{s=1}^{t-1} \left[ \int_{B_s} \pi_s(\mathrm{d}a_s | z_s) \int_{C_{s+1}} P_s(\mathrm{d}x_{s+1} | z_s, a_s) \right],$$

equivalently,

$$\mathbb{P}^\pi_{M,t}(dz_t) = \rho(dx_1) \prod_{s=1}^{t-1} \left[ \pi_s(da_s | z_s) \, P_s(dx_{s+1} | z_s, a_s) \right].$$

Analogously, one obtains a unique path measure $\mathbb{P}^\pi_M$ on $(\mathcal{Z}_\infty, \mathcal{B}(\mathcal{Z}_\infty))$.

Now, define the joint law on $\mathcal{M}^\sharp \times \mathcal{Z}_t$ by

$$\mathbf{P}^\pi_t(\mathrm{d}M, \mathrm{d}z_t) := \mathcal{P}(\mathrm{d}M) \, \mathbb{P}^\pi_{M,t}(\mathrm{d}z_t),$$

and the trajectory marginal

$$\mathbb{P}^\pi_t(A) := \int_{\mathcal{M}^\sharp} \mathbb{P}^\pi_{M,t}(A) \, \mathcal{P}(\mathrm{d}M), \qquad A \in \mathcal{B}(\mathcal{Z}_t).$$

Similarly, we also define $\mathbf{P}^\pi(\mathrm{d}M, \mathrm{d}z)$ on $M^\sharp \times \mathcal{Z}_\infty$ and $\mathbb{P}^\pi(A)$ for $A \in \mathcal{B}(\mathcal{Z}_\infty)$.

Lastly, since $\Omega$ and $\mathcal{Z}_t$ are standard Borel, regular conditional probabilities exist on $\Omega \times \mathcal{Z}_t$; by pushforward through $\phi$ they induce regular conditional probabilities on $\mathcal{M}^\sharp \times \mathcal{Z}_t$, for all $t \geq 1$.

### B.1.2 POSTERIOR DISTRIBUTION OVER THE TRUE HYPOTHESIS AND INFERENCE RULE OPTIMALITY

We first record a domination assumption that allows us to express likelihoods w.r.t. fixed reference measures and obtain continuity.

**Assumption 1** (Domination and joint continuity). *There exist probability measures $\lambda_0, \lambda$ on $(\mathcal{X}, \mathcal{B}(\mathcal{X}))$ such that, for all $(\rho, P) \in \mathcal{M}^\sharp$, $s \in \mathbb{N}$, and $(z, a) \in \mathcal{Z}_s \times \mathcal{A}$,*

$$\rho(\cdot) \ll \lambda_0(\cdot) \quad and \quad P_s(\cdot | z, a) \ll \lambda(\cdot).$$

*We also let $p_0^\omega(x) := \frac{d\rho_\omega}{d\lambda_0}(x)$ and $p_s^\omega(x | z, a) := \frac{dP_s^\omega(\cdot | z, a)}{d\lambda}(x)$ be the corresponding densities (versions chosen jointly measurable).*

*Remark.* For compact $\mathcal{X} \subset \mathbb{R}$, such a dominating pair always exists (e.g., Lebesgue on $\mathcal{X}$).

Under assum. 1, define the (policy-independent) likelihood for $(\omega, z) \in \Omega \times \mathcal{Z}_t$:

$$\ell_t(M, z) := p_0(x_1) \prod_{s=1}^{t-1} p_s(x_{s+1} | z_s, a_s).$$

We now give a posterior kernel representation that is independent of $\pi$.

**Lemma B.1** (Posterior kernel). *For each $t \in \mathbb{N}$ there exists a probability kernel $R_t : \mathcal{Z}_t \times \mathcal{B}(M^\sharp) \to [0, 1]$, independent of $\pi$, such that for every policy $\pi$ and all $A \in \mathcal{B}(M^\sharp)$, $Z \in \mathcal{B}(\mathcal{Z}_t)$,*

$$\mathbf{P}^\pi_t(M \in A, \, \mathcal{D}_t \in Z) = \int_Z R_t(A|z) \, \mathbb{P}^\pi_{M \sim \mathcal{P}, t}(dz).$$

*Consequently, for $B \subset \mathcal{H}$,*

$$\mathbb{P}_t(H^\star \in B | \mathcal{D}_t = z) := R_t\big(\{(\rho, P) \in \mathcal{M}^\sharp : h^\star(\rho, P) \in B\} | z\big) \quad \textit{for } \mathbb{P}^\pi_t\textit{-a.e. } z.$$

*Proof.* Fix $\pi$ and $t$. Define the reference measure on $\mathcal{Z}_t$ (depending on $\pi$)

$$\nu_t^\pi(dz) := \lambda_0(\mathrm{d}x_1) \prod_{s=1}^{t-1} \left[ \pi_s(\mathrm{d}a_s | z_s)\, \lambda(\mathrm{d}x_{s+1}) \right].$$

By construction and assum. 1, $\mathbb{P}_{M,t}^\pi \ll \nu_t^\pi$ for every $\omega$, and the Radon–Nikodym density is

$$\frac{d\mathbb{P}_{M,t}^\pi}{d\nu_t^\pi}(z) = \ell_t(M, z),$$

which does not depend on $\pi$. Therefore,

$$\mathbf{P}_t^\pi(M \in A, \mathcal{D}_t \in Z) = \int_A \int_Z \ell_t(M, z)\, \nu_t^\pi(\mathrm{d}z)\, \mathcal{P}(\mathrm{d}M),$$

and

$$\mathbb{P}_t^\pi(Z) = \int_Z \int_{\mathcal{M}^\sharp} \ell_t(M, z)\, \mathcal{P}(\mathrm{d}M)\, \nu_t^\pi(\mathrm{d}z).$$

Absolute continuity $\mathbf{P}_t^\pi(A, \cdot) \ll \mathbb{P}_t^\pi(\cdot)$ follows immediately, and the Radon–Nikodym derivative is the displayed ratio, which we denote $R_t(A|z)$. Standard arguments show $R_t(\cdot|z)$ is a probability measure and $z \mapsto R_t(A|z)$ is measurable; independence of $\pi$ is evident from the formula. Mapping through $h^\star$ yields the posterior on $\mathcal{H}$. $\square$

Define then

$$\mathbb{P}_t^\pi(H^\star = H) := \int_{\mathcal{Z}_t} \mathbb{P}_t(H^\star = H | \mathcal{D}_t = z) \mathbb{P}_t^\pi(\mathrm{d}z). \tag{12}$$

We now provide a proof of the optimality of an inference rule. In the following, we use the following quantity

$$r_t(z) := \max_{H \in \mathcal{H}} \mathbb{P}_t(H^\star = H | \mathcal{D}_t = z), \tag{13}$$

which is the maximum value of the posterior distribution at time $t$ for some dataset $z$.

**Proposition B.2.** *Consider a fixed policy $\pi$. Let $t \in \mathbb{N}$ and $z \sim \mathbb{P}_t^\pi$. For any $t$ the optimal inference rule to $\sup_{I_t} \mathbb{P}_t^\pi(H^\star = I_t(\mathcal{D}_t))$ is given by $I_t^\star(z) = \arg\max_{H \in \mathcal{H}} \mathbb{P}_t(H^\star = H | \mathcal{D}_t = z)$ (break ties according to some fixed ordering).*

*Proof.* Fix a policy $\pi$ and an inference rule $I_t$ at timestep $t$. By definition, we have

$$\mathbb{P}_t^\pi(H^\star = I_t(\mathcal{D}_t)) = \int_{\mathcal{Z}_t} \sum_{H \in \mathcal{H}} \mathbf{1}_{\{I_t(z)=H\}} \mathbb{P}_t(H^\star = H | \mathcal{D}_t = z) \mathbb{P}_t^\pi(\mathrm{d}z),$$

$$\text{(Posterior independent of } \pi\text{)}$$

$$\leq \int_{\mathcal{Z}_t} \max_{H \in \mathcal{H}} \mathbb{P}_t(H^\star = H | \mathcal{D}_t = z) \mathbb{P}_t^\pi(\mathrm{d}z),$$

$$= \int_{\mathcal{Z}_t} r_t(z) \mathbb{P}_t^\pi(\mathrm{d}z).$$

However, for any $\mathcal{D}_t = z$ choosing $I_t(z) = \arg\max_{H \in \mathcal{H}} \mathbb{P}_t^\pi(H^\star = H | \mathcal{D}_t = z)$ (break ties according to some fix ordering $\Rightarrow$ hence $I_t(z)$ is Borel measurable) yields that

$$\mathbb{P}_t^\pi(H^\star = I_t(\mathcal{D}_t)) = \int_{\mathcal{Z}_t} r_t(z) \mathbb{P}_t^\pi(\mathrm{d}z),$$

which concludes the proof. $\square$

### B.1.3 FIXED BUDGET SETTING: OPTIMAL POLICY

We now turn our attention to the fixed budget setting. In particular, we prove that an optimal policy $\pi_t^\star$ in $\mathcal{D}_t$ attains the optimal value defined as $V_t(\mathcal{D}_t) = \max_a \mathbb{E}_{x_{t+1}|(\mathcal{D}_t,a)}[V_{t+1}((\mathcal{D}_t,a,x_{t+1})|\mathcal{D}_t,a]$ with $V_N(\mathcal{D}_N) = \max_H \mathbb{P}_t(H^\star = H|\mathcal{D}_N)$ (see a rigorous definition of $x_{t+1}|(\mathcal{D}_t,a)$ below).

First, note that from prop. B.2 we can deduce that the optimal objective in the fixed budget satisfies, for all $t \geq 1$,

$$\sup_{\pi, I_t} \mathbb{P}_t^\pi(H^\star = I_t(\mathcal{D}_t)) = \sup_\pi \mathbb{E}_t^\pi[r_t(\mathcal{D}_t)]$$

where $r_t(z) := \max_{H \in \mathcal{H}} \mathbb{P}_t(H^\star = H|\mathcal{D}_t = z)$.

We now show that there exists an optimal deterministic policy $\pi^\star$ that optimally solves the fixed budget regime. First, define the following posterior mixture for any $t \in \mathbb{N}, X \in \mathcal{B}(\mathcal{X}), (z, a) \in \mathcal{Z}_t \times \mathcal{A}$:

$$\bar{P}_t(x' \in X|z, a) := \int_{\mathcal{M}^\sharp} P_t(X|z, a) R_t(\mathrm{d}M|z), \tag{14}$$

where $P_t(\cdot|z, a)$ is the transition function at step $t$ in $(M, z, a)$. This is simply the posterior $x'|(z, a)$, that in the main text of the manuscript is denoted by $x_{t+1}|(\mathcal{D}_t, a)$.

**Optimal value.** Define the value at $N \in \mathbb{N}$ as $V_N(z_N) = r_N(z_N)$ for any $z_N \in \mathcal{Z}_N$, and define the value function for $z_t \in \mathcal{Z}_t, a \in \mathcal{A}$ as

$$V_t(z_t) = \max_{a \in \mathcal{A}} Q_t(z_t, a), \quad Q_t(z_t, a) = \int_\mathcal{X} V_{t+1}(\underbrace{z_t, a, x'}_{=z_{t+1}}) \bar{P}_t(\mathrm{d}x'|z_t, a), \quad t = 1, \ldots, N-1, \tag{15}$$

For some ordering on $\mathcal{A}$, for $z \in \mathcal{Z}_t$ define $\pi_t^\star(z) \in \arg\max_{a \in \mathcal{A}} Q_t(z, a)$ (break ties according to the ordering). We have the following result.

**Proposition B.3.** *For any $t \in \{1, \ldots, N-1\}, z \in \mathcal{Z}_t$, the policy $\pi_t^\star(z) \in \arg\max_{a \in \mathcal{A}} Q_t(z, a)$ (break ties according to a fixed ordering) is an optimal policy, that is*

$$\sup_{\pi, I_N} \mathbb{P}_N^\pi(H_M^\star = I_N(\mathcal{D}_N)) = \mathbb{E}_N^{\pi^\star}[r_N(\mathcal{D}_N)]. \tag{16}$$

*Proof.* To prove the result, we use lem. B.4, which shows that $\mathbb{E}_N^\pi[V_N(\mathcal{D}_N)|\mathcal{D}_t] \leq V_t(\mathcal{D}_t)$ holds $\mathbb{P}_t^\pi$-almost surely for any policy $\pi$ and $t \in [T]$, with equality if $\pi = \pi^\star$. Then, using this inequality we can show that for $t = 1$, with $z_1 \in \mathcal{X}$, we have

$$\mathbb{E}_N^\pi[V_N(\mathcal{D}_N)|\mathcal{D}_1 = z_1] \leq V_1(z_1) \Rightarrow \mathbb{E}_N^\pi[r_N(\mathcal{D}_N)] \leq \mathbb{E}_1[V_1(\mathcal{D}_1)],$$

with equality if $\pi = \pi^\star$, implying $\pi^\star$ is optimal since we can attain the r.h.s. (note that it does not depend on $\pi$). Hence

$$\sup_{\pi, I_N} \mathbb{P}_N^\pi(H_M^\star = I_N(\mathcal{D}_N)) = \sup_\pi \mathbb{E}_N^\pi[r_N(\mathcal{D}_N)] = \mathbb{E}_N^{\pi^\star}[r_N(\mathcal{D}_N)].$$

where the first equality follows from prop. B.2. $\qquad \square$

We now prove the result used in the proof.

**Lemma B.4.** *Consider the fixed budget setting with horizon $N$. For any policy $\pi = (\pi_s)_{s \geq 1}$, $t \in \{1, \ldots, N\}, z \in \mathcal{Z}_t$, we have*

$$\mathbb{E}_N^\pi[V_N(\mathcal{D}_N)|\mathcal{D}_t = z] \leq V_t(z) \quad \mathbb{P}_{M \sim \mathcal{P}, t}^\pi\text{-almost surely}, \tag{17}$$

*with equality when $\pi = \pi^\star$.*

*Proof.* We prove it by backward induction. For $t = N$ the equality holds by definition. Assume it holds for $t + 1$, then at time $t$ for any policy $\pi$ we have

$$\begin{aligned}
\mathbb{E}_N^\pi[V_N(\mathcal{D}_N)|\mathcal{D}_t = z] &= \mathbb{E}_N^\pi[\mathbb{E}_N^\pi[V_N(\mathcal{D}_N)|\mathcal{D}_{t+1}]|\mathcal{D}_t = z], \\
&\leq \mathbb{E}_{t+1}^\pi[V_{t+1}(\mathcal{D}_{t+1})|\mathcal{D}_t = z], \\
&= \mathbb{E}_t^\pi[Q_t(z, \pi_t(z))|\mathcal{D}_t = z], \\
&\leq V_t(z),
\end{aligned}$$

and if $\pi = \pi^\star$ then both inequalities hold since they hold at $t + 1$. Hence the result holds also for $t$, which concludes the induction argument. $\qquad \square$

### B.1.4 FIXED CONFIDENCE SETTING: DUAL PROBLEM FORMULATION

In the fixed confidence setting we are interested in solving the following problem.

$$\inf_{\tau,\pi,I} \mathbb{E}^\pi[\tau] \quad \text{subject to} \quad \mathbb{P}^\pi(H^\star = I_\tau(\mathcal{D}_\tau)) \geq 1 - \delta, \ \mathbb{E}^\pi[\tau] < \infty. \tag{18}$$

where $\tau$ is a stopping time adapted to $(\sigma(\mathcal{D}_t))_t$ (recall that it counts the total number of observations; thus, if $\tau = t \Rightarrow$ we have observations $x_1, \ldots, x_t$), $\pi = (\pi_s)_{s\geq 1}$, is a collection of policies, and $I = (I_s)_{s\geq 1}$, is a sequence of recommendation rules. Furthermore, we have that $\mathbb{P}^\pi(\tau < \infty) = 1$ (this follows from $\mathbb{E}^\pi[\tau] < \infty$).

**Dual problem and optimal recommendation rule** In the following we focus on the dual problem of eq. (18). First, we show what is the dual problem, and what is the optimal recomendation rule.

**Proposition B.5.** *The Lagrangian dual of the problem in eq.* (18) *is given by*

$$\sup_{\lambda\geq 0} \inf_{\pi,\tau,I} V_\lambda(\pi,\tau,I) = \sup_{\lambda\geq 0} \inf_{\pi,\tau,I} \lambda(1-\delta) + \mathbb{E}^\pi\left[\tau - \lambda\mathbb{P}_\tau(H^\star = I_\tau(\mathcal{D}_\tau)|\mathcal{D}_\tau)\right], \tag{19}$$

*where $\lambda \geq 0$ is the Lagrangian variable.*

*Proof.* Since any feasible solution stops almost surely, we can also write

$$\mathbb{P}^\pi(H_M^\star = I_\tau(\mathcal{D}_\tau)) = \sum_{t\geq 1} \mathbb{P}_t^\pi(I_t(\mathcal{D}_t) = H^\star, \tau = t), \qquad \text{(law of total probability)}$$

$$= \sum_{t\geq 1} \mathbb{E}_t^\pi\left[\mathbf{1}_{\{\tau=t\}}\mathbf{1}_{\{H^\star=I_t(\mathcal{D}_t)\}}\right],$$

$$= \sum_{t\geq 1} \mathbb{E}_t^\pi\left[\mathbb{E}_t^\pi\left[\mathbf{1}_{\{\tau=t\}}\mathbf{1}_{\{H^\star=I_t(\mathcal{D}_t)\}}\Big|\mathcal{D}_t\right]\right], \qquad \text{(tower rule)}$$

$$= \sum_{t\geq 1} \mathbb{E}_t^\pi\left[\mathbf{1}_{\{\tau=t\}}\mathbb{E}_t^\pi\left[\mathbf{1}_{\{H^\star=I_t(\mathcal{D}_t)\}}\Big|\mathcal{D}_t\right]\right], \qquad (\{\tau = t\} \in \sigma(\mathcal{D}_t))$$

$$= \sum_{t\geq 1} \mathbb{E}_t^\pi\left[\mathbf{1}_{\{\tau=t\}}\mathbb{P}_t(H^\star = I_t(\mathcal{D}_t)|\mathcal{D}_t)\right]$$

$$\text{(Outer expectation integrates over } \mathcal{D}_t)$$

where we used the fact that the posterior distribution does not depend on $\pi$ (lem. B.1). Using this decomposition, for $\lambda \geq 0$ (the dual variable) we can write the Lagrangian dual of the problem as

$$V_\lambda(\pi,\tau,I) := \mathbb{E}^\pi\left[\tau + \lambda\left(1 - \delta - \sum_{t\geq 1}\mathbf{1}_{\{\tau=t\}}\mathbb{P}_t(H^\star = I_t(\mathcal{D}_t)|\mathcal{D}_t)\right)\right],$$

$$= \lambda - \lambda\delta + \mathbb{E}^\pi\left[\sum_{t\geq 1}\mathbf{1}_{\{t\leq\tau\}} - \lambda\mathbf{1}_{\{\tau=t\}}\mathbb{P}_t(H^\star = I_t(\mathcal{D}_t)|\mathcal{D}_t)\right],$$

where we used that $\mathbb{E}^\pi[\tau] = \mathbb{E}^\pi\left[\sum_{t=1}^\tau 1\right] = \mathbb{E}^\pi\left[\sum_{t=1}^\infty \mathbf{1}_{\{t\leq\tau\}}\right]$. $\square$

For the dual problem, we now show we can embed the stopping rule as a stopping action. Define the extended action space $\bar{\mathcal{A}} := \mathcal{A} \cup \{a_{\text{stop}}\}$, where $a_{\text{stop}}$ is absorbing. We show that for every $(\tau, \pi, I)$ there exists a policy $\bar\pi = (\bar\pi_s)_{s\geq 1}$, with $\bar\pi_s : \mathcal{Z}_s \to \Delta(\bar{\mathcal{A}})$, such that $V_\lambda(\pi,\tau,I) = V_\lambda(\bar\pi,I)$, where

$$V_\lambda(\bar\pi, I) := \lambda(1 - \delta) + \mathbb{E}^{\bar\pi}\left[\bar\tau - \lambda\mathbb{P}_{\bar\tau}(H^\star = I_{\bar\tau}(\mathcal{D}_{\bar\tau})|\mathcal{D}_{\bar\tau})\right].$$

and

$$\bar\tau = \inf\{t : a_t = a_{\text{stop}}\}.$$

At the beginning of round $t$, given $\mathcal{D}_t$, the learner may stop by choosing $a_{\text{stop}}$ (termination at $t$, no new observation) or continue by choosing $a_t \neq a_{\text{stop}}$ and then observing $x_{t+1}$. If the learner decides to stop after seeing $x_{t+1}$, this is equivalent to choosing $a_{\text{stop}}$ at round $t + 1$, leading to $\bar\tau = t + 1 = \tau$.

**Lemma B.6.** *For every $(\pi, \tau, I)$ with $\tau \geq 1$ a.s., there exists a policy $\bar{\pi}$ on $\bar{\mathcal{A}} = \mathcal{A} \cup \{a_{\text{stop}}\}$ with stopping rule $\bar{\tau} = \inf\{t : a_t = a_{\text{stop}}\}$ such that $V_\lambda(\pi, \tau, I) = V_\lambda(\bar{\pi}, I)$.*

*Proof.* Since $\{\tau = t\} \in \sigma(\mathcal{D}_t)$ note that there exists $A_t \in \mathcal{B}(\mathcal{Z}_t)$ such that $\{\tau = t\} = \{\mathcal{D}_t \in A_t\}$. Define, for all $t \geq 1$ and $z \in \mathcal{Z}_t$,

$$\bar{\pi}_t(a_{\text{stop}}|z) = \mathbf{1}_{\{z \in A_t\}}, \qquad \bar{\pi}_t(a|z) = \pi_t(a|z) \ (a \in \mathcal{A}).$$

Clearly we have $\{\bar{\tau} = t\} = \{a_t = a_{\text{stop}}\} = \{\tau = t\}$, then

$$V_\lambda(\bar{\pi}, I) = \lambda(1 - \delta) + \mathbb{E}^{\bar{\pi}}\left[\bar{\tau} - \lambda \mathbb{P}_{\bar{\tau}}(H^\star = I_{\bar{\tau}}(\mathcal{D}_{\bar{\tau}})|\mathcal{D}_{\bar{\tau}})\right],$$

$$= \lambda(1 - \delta) + \mathbb{E}^{\bar{\pi}}\left[\sum_{t \geq 1} \mathbf{1}_{\{t \leq \bar{\tau}\}} - \lambda \mathbf{1}_{\{\bar{\tau}=t\}} \mathbb{P}_t(H^\star = I_t(\mathcal{D}_t)|\mathcal{D}_t)\right],$$

$$= \lambda(1 - \delta) + \mathbb{E}^{\bar{\pi}}\left[\sum_{t \geq 1}^{\bar{\tau}} 1 - \lambda \mathbf{1}_{\{a_t=a_{\text{stop}}\}} \mathbb{P}_t(H^\star = I_t(\mathcal{D}_t)|\mathcal{D}_t)\right],$$

$$= \lambda(1 - \delta) + \mathbb{E}^{\bar{\pi}}\left[\sum_{t \geq 1}^{\tau} 1 - \lambda \mathbf{1}_{\{\tau=t\}} \mathbb{P}_t(H^\star = I_t(\mathcal{D}_t)|\mathcal{D}_t)\right],$$

$$= \lambda(1 - \delta) + \mathbb{E}^{\pi}\left[\sum_{t \geq 1} \mathbf{1}_{\{t \leq \tau\}} - \lambda \mathbf{1}_{\{\tau=t\}} \mathbb{P}_t(H^\star = I_t(\mathcal{D}_t)|\mathcal{D}_t)\right],$$

$$= \lambda(1 - \delta) + \mathbb{E}^{\pi}\left[\tau - \lambda \mathbb{P}_\tau(H^\star = I_\tau(\mathcal{D}_\tau)|\mathcal{D}_\tau)\right].$$

$\square$

Then, in the following, we assume to work with the extended space $\bar{\mathcal{A}}$, and indicate by $\tau = \inf\{t : a_t = a_{\text{stop}}\}$ the stopping time. We avoid the bar notation for simplicity.

We now show what is the optimal inference rule.

**Proposition B.7.** *For any $t \in \mathbb{N}$ the optimal inference rule satisfies $I_t(\mathcal{D}_t) \in \arg\max_{H \in \mathcal{H}} \mathbb{P}_t(H^\star = H|\mathcal{D}_t = z_t)$ (break ties according to some fixed ordering), where $z_t \in \mathcal{Z}_t$. Moreover, we also have*

$$\sup_{\lambda \geq 0} \inf_{\pi, I} V_\lambda(\pi, I) = \sup_{\lambda \geq 0} \inf_{\pi} \lambda(1 - \delta) + \mathbb{E}^{\pi}\left[\tau - \lambda r_\tau(\mathcal{D}_\tau)\right], \tag{20}$$

*where $\tau = \inf\{t : a_t = a_{\text{stop}}\}$ and $r_t(z_t) := \max_H \mathbb{P}_t(H^\star = H|\mathcal{D}_t = z_t)$.*

*Proof.* First, we optimize over recommendation rules. For any $t \in \mathbb{N}, z_t \in \mathcal{Z}_t$ define $r_t(z_t) = \max_{H \in \mathcal{H}} \mathbb{P}_t(H^\star = H|\mathcal{D}_t = z_t)$ as before. Then, for fixed $(\pi, \tau)$ we have that

$$\inf_I V_\lambda(\pi, I) = \lambda - \lambda\delta + \mathbb{E}^{\pi}\left[\inf_I \sum_{t \geq 1} \mathbf{1}_{\{t \leq \tau\}} - \lambda \mathbf{1}_{\{\tau=t\}} \mathbb{P}_t(H^\star = I_t(\mathcal{D}_t)|\mathcal{D}_t)\right],$$

$$= \lambda - \lambda\delta + \mathbb{E}^{\pi}\left[\sum_{t \geq 1} \mathbf{1}_{\{t \leq \tau\}} - \lambda \mathbf{1}_{\{\tau=t\}} \sup_{I_t} \mathbb{P}_t(I_t(\mathcal{D}_t) = H^\star|\mathcal{D}_t)\right],$$

$$= \lambda - \lambda\delta + \mathbb{E}^{\pi}\left[\sum_{t \geq 1} \mathbf{1}_{\{t \leq \tau\}} - \lambda \mathbf{1}_{\{\tau=t\}} r_t(\mathcal{D}_t)\right],$$

where the last step follows from prop. B.2. Therefore, we can also conclude that

$$\sup_{\lambda \geq 0} \inf_{\pi, I} V_\lambda(\pi, I) = \sup_{\lambda \geq 0} \inf_{\pi} \lambda - \lambda\delta + \mathbb{E}^{\pi}\left[\tau - \lambda r_\tau(\mathcal{D}_\tau)\right].$$

$\square$

### B.1.5 FIXED CONFIDENCE SETTING: OPTIMAL POLICY

We now optimize over policies. Recall that $\mathcal{A}$ includes the stopping action, and $\tau = \inf\{t : a_t = a_{\mathrm{stop}}\}$. For $t \in \mathbb{N}, z \in \mathcal{Z}_t$ define the optimal value to go

$$V_t(z; \lambda) := \inf_{\pi : \tau \geq t} \lambda - \lambda\delta + \mathbb{E}^\pi \left[ \tau - t - \lambda r_\tau(\mathcal{D}_\tau) | \mathcal{D}_t = z \right].$$

Also define the following optimal $Q$-function for $z \in \mathcal{Z}_t, a \neq a_{\mathrm{stop}}$

$$Q_t(z, a; \lambda) := 1 + \int_{\mathcal{X}} V_{t+1}(\underbrace{z, a, x'}_{=z'}; \lambda) \, \bar{P}_t(\mathrm{d}x'|z, a)$$

where $\bar{P}_t$ is the posterior mixture defined in eq. (14). We also set

$$Q_t(z, a_{\mathrm{stop}}; \lambda) := \lambda(1 - \delta - r_t(z)).$$

Consider then the policy $\pi_\lambda^\star = (\pi_t^\star)_t$, where $\pi_t^\star(z; \lambda) \in \arg\min_{a \in \mathcal{A}} Q_t(z, a; \lambda)$, where we break ties according to some fixed ordering over $\mathcal{A}$. We have then the following result.

**Proposition B.8.** $\pi_\lambda^\star$ *is a $\lambda$-optimal policy. Furthermore, the optimal value for $z \in \mathcal{Z}_t$ satisfies*

$$V_t(z; \lambda) = \min_a Q_t(z, a; \lambda). \tag{21}$$

*Proof.* Fix $\mathcal{D}_t = z, z \in \mathcal{Z}_t$. Assume the optimal stopping action stops at $\tau = t$ for such $z$. Then $V_t(z; \lambda) = \lambda - \lambda\delta - \lambda r_t(z) = Q_t(z, a_{\mathrm{stop}}; \lambda)$.

Otherwise, assume the optimal stopping rule stops for $\tau > t$. Then

$$V_t(z; \lambda) = \inf_{\pi : \tau > t} \lambda - \lambda\delta + \mathbb{E}^\pi \left[ \tau - t - \lambda r_\tau(\mathcal{D}_\tau) | \mathcal{D}_t = z \right],$$

$$= \inf_{\pi : \tau > t} 1 + \mathbb{E}_a^\pi \left[ \int_{\mathcal{X}} V_{t+1}(z, a, x'; \lambda) \, \bar{P}_t(\mathrm{d}x'|z, a) \right],$$

$$= \min_{a \neq a_{\mathrm{stop}}} Q_t(z, a; \lambda).$$

We then clearly obtain the lower bound

$$V_t(z; \lambda) \geq \min \left\{ \lambda(1 - \delta - r_t(z)), \min_{a \in \mathcal{A}} Q_t(z, a; \lambda) \right\}.$$

We now show that $\pi^\star$ achieves this value, and thus is optimal.

1. If $\tau = t$, then $\lambda(1 - \delta - r_t(z) \leq \min_{a \neq a_{\mathrm{stop}}} Q_t(z, a; \lambda)$ and the value to go for $\pi^\star$ is exactly $\lambda(1 - \delta - r_t(z))$.

2. if $\tau \neq t$, then $\lambda(1 - \delta - r_t(z) \geq \min_{a \neq a_{\mathrm{stop}}} Q_t(z, a; \lambda)$, and the value to go is $Q_t(z, \pi_t(z); \lambda) = \min_a Q_t(z, a; \lambda) = V_t(z; \lambda)$.

Therefore, the value to go for $\pi^\star$ at time $t$ in $\mathcal{D}_t = z$ attains the lower bound $\min_{a \in \mathcal{A}} Q_t(z, a; \lambda)$, and is thus optimal. Applying the result from the previous proposition leads to the desired result. $\square$

### B.1.6 FIXED CONFIDENCE SETTING: IDENTIFIABILITY AND CORRECTNESS

Lastly, to verify the correctness, we need to make an explicit identifiability assumption.

**Assumption 2.** *For every $\delta > 0$ there exists a policy $\pi$ with $\mathbb{E}^\pi[\tau] < \infty$, such that $\mathbb{E}^\pi[r_\tau(\mathcal{D}_\tau)] \geq 1 - \delta$.*

Now we show that the optimization problem solved by ICPE can lead to a $\delta$-correct policy and stopping rule. To that aim, define

$$\mathcal{S}(\lambda) := \arg\min_\pi V_\lambda(\pi), \quad \text{where } V_\lambda(\pi) := \lambda(1 - \delta) + \mathbb{E}^\pi[\tau - \lambda r_\tau(\mathcal{D}_\tau)].$$

Observe that the set $\mathcal{S}(\lambda)$ is not empty since we know that $\pi_\lambda^\star$ belongs to it. We have the following result.

**Lemma B.9.** *Define the set* $\Phi(\lambda) = \{\mathbb{E}^{\pi}[r_{\tau}(\mathcal{D}_{\tau})] : \pi \in \mathcal{S}(\lambda)\}$. *Then, any* $\phi(\lambda) \in \Phi(\lambda)$ *is non-decreasing and under assum.* [2] *any* $\phi(\lambda) \in \Phi(\lambda)$ *satisfies* $\lim_{\lambda \to \infty} \phi(\lambda) = 1$.

*Proof.* We first prove the limit, and then prove the monotonicity.

*Step 1:* $\lim_{\lambda \to \infty} \phi(\lambda) = 1$. For $\epsilon > 0$ consider a policy $\pi_{\epsilon}$ such that $\mathbb{E}^{\pi_{\epsilon}}[r_{\tau}(\mathcal{D}_{\tau})] \geq 1 - \epsilon$.

Define $g(\lambda) := \inf_{\pi, I} V_{\lambda}(\pi, I) = \inf_{\pi} V_{\lambda}(\pi)$.

Now, assume that some feasible minimizer $\pi \in \mathcal{S}(\lambda)$ satisfies $\mathbb{E}^{\pi}[r_{\tau}(\mathcal{D}_{\tau})] \leq 1 - 2\epsilon$. We proceed by contradiction and show that this is not possible. First, note that

$$V_{\lambda}(\pi_{\epsilon}) - g(\lambda) = \mathbb{E}^{\pi_{\epsilon}}[\tau] - \mathbb{E}^{\pi}[\tau] + \lambda(\mathbb{E}^{\pi}[r_{\tau}(\mathcal{D}_{\tau})] - \mathbb{E}^{\pi_{\epsilon}}[r_{\tau}(\mathcal{D}_{\tau})]).$$

Observe then that $\mathbb{E}^{\pi}[r_{\tau}(\mathcal{D}_{\tau})] - \mathbb{E}^{\pi_{\epsilon}}[r_{\tau}(\mathcal{D}_{\tau})] \leq -\epsilon < 0$. Therefore, we obtain that whenever $\lambda > \frac{\mathbb{E}^{\pi_{\epsilon}}[\tau]}{\epsilon}$ we have that

$$V_{\lambda}(\pi_{\epsilon}) - g(\lambda) \leq \mathbb{E}^{\pi_{\epsilon}}[\tau] - \lambda\epsilon < 0,$$

which is however a contradiction to $g(\lambda)$ being a minimum. Hence, any feasible solution $\pi \in \mathcal{S}(\lambda)$ must satisfy $\mathbb{E}^{\pi}[r_{\tau}(\mathcal{D}_{\tau})] > 1 - 2\epsilon$ for $\lambda > \mathbb{E}^{\pi_{\epsilon}}[\tau]/\epsilon$. Since any $\pi \in \mathcal{S}(\lambda)$ satisfies $\mathbb{E}^{\pi}[\tau] < \infty$, we have that for any fixed $\epsilon > 0$ we get $\lim_{\lambda \to \infty} \inf_{\pi \in \mathcal{S}(\lambda)} \mathbb{E}^{\pi}[r_{\tau}(\mathcal{D}_{\tau})] > 1 - 2\epsilon$. Since the statement holds for any $\epsilon > 0$, letting $\epsilon \to 0$ yields the desired result.

*Step 2: Monotonicity.* Consider two feasible optimal solutions $\pi_1 \in \mathcal{S}(\lambda_1)$ and $\pi_2 \in \mathcal{S}(\lambda_2)$, with $\lambda_2 > \lambda_1$. We have that

$$g(\lambda_2) - V_{\lambda_1}(\pi_1) \leq V_{\lambda_2}(\pi_1) - V_{\lambda_1}(\pi_1) = (\lambda_2 - \lambda_1)(1 - \delta - \mathbb{E}^{\pi_1}[r_{\tau_1}(\mathcal{D}_{\tau_1})])$$

and

$$g(\lambda_1) - V_{\lambda_2}(\pi_2) \leq V_{\lambda_1}(\pi_2) - V_{\lambda_2}(\pi_2) = (\lambda_1 - \lambda_2)(1 - \delta - \mathbb{E}^{\pi_2}[r_{\tau_2}(\mathcal{D}_{\tau_2})]).$$

Summing up, and using that $g(\lambda_i) = V_{\lambda_i}(\pi_i)$ we have that

$$0 \leq (\lambda_2 - \lambda_1)(\mathbb{E}^{\pi_2}[r_{\tau_2}(\mathcal{D}_{\tau_2})] - \mathbb{E}^{\pi_1}[r_{\tau_1}(\mathcal{D}_{\tau_1})]).$$

Since $\lambda_2 > \lambda_1$, we must have that $\mathbb{E}^{\pi_2}[r_{\tau_2}(\mathcal{D}_{\tau_2})] - \mathbb{E}^{\pi_1}[r_{\tau_1}(\mathcal{D}_{\tau_1})] \geq 0$. Since we chose the elements in $\mathcal{S}$ arbitrarily, it implies that any $\phi(\lambda) \in \Phi(\lambda)$ is non-decreasing. $\qquad\square$

Lastly, to verify the correctness, we use the fact that the sub-gradient of the optimal value of the dual problem is non-decreasing. To show this result, we employ the following proposition from (Hantoute & López, 2008) (see Prop. 3.1 therein), which characterizes the subdifferential of the supremum of a family of affine functions.

**Proposition B.10** (Subdifferential of the supremum of affine functions (Hantoute & López, 2008)). *Given a non-empty set* $\{(a_t, b_t) : t \in \mathcal{T}\} \subset \mathbb{R}^2$, *and the supremum function* $f(x) : \mathbb{R} \to \mathbb{R} \cup \{\infty\}$

$$f(x) = \sup\{a_t x - b_t : t \in \mathcal{T}\},$$

*for every* $x \in \mathrm{dom} f$ *we have*

$$\partial f(x) = \cap_{\epsilon > 0} \mathrm{cl}\left(\mathrm{conv}\{a_t : t \in \mathcal{T}_{\epsilon}(x)\} + B(x)\right)$$

*with*

$$\mathcal{T}_{\epsilon}(x) := \{t \in \mathcal{T} : a_t x - b_t \geq f(x) - \epsilon\},$$

*and*

$$B(x) := \{y \in \mathbb{R} : (y, yx) \in (\overline{\mathrm{conv}}\{(a_t, b_t) : t \in \mathcal{T}\})_{\infty}\},$$

*where* $C_{\infty}$ *is the recession cone of a set* $C$ *and* $\overline{\mathrm{conv}}(\cdot)$ *denotes the closed convex hull of a set. In particular, if* $x \in \mathrm{int}(\mathrm{dom} f)$ *we have*

$$\partial f(x) = \cap_{\epsilon > 0} \overline{\mathrm{conv}}\{a_t : t \in \mathcal{T}_{\epsilon}(x)\}.$$

This last proposition permits us to define the subdifferential of the supremum of affine functions, and, as we see later, we can also find a lower bound on any subdifferential $d \in \partial f(x)$.

We are now ready to state the identifiability result.

**Proposition B.11.** *Consider assum. 2, and, for simplicity, assume the set of optimal policies $\mathcal{S}(\lambda)$ is a singleton for each $\lambda$. Then, an optimal solution $(\lambda^\star, \pi^\star_{\lambda^\star})$ satisfies*

$$\boxed{\mathbf{P}^{\pi^\star_{\lambda^\star}}(H^\star_M = \hat{H}_{\tau^\star_{\lambda^\star}}) \geq 1 - \delta,} \tag{22}$$

*for any critical point $\lambda^\star \in \arg\max_{\lambda \geq 0} \inf_{\pi, I} V_\lambda(\pi, I)$.*

*Proof.* Define $g(\lambda) := \inf_{\pi, I} V_\lambda(\pi, I) = \inf_\pi V_\lambda(\pi)$. Clearly $V_\lambda$ is differentiable with respect to $\lambda$ for all $\pi$, and we have $\partial V_\lambda(\pi)/\partial \lambda = 1 - \delta - \mathbb{E}^\pi[r_\tau(\mathcal{D}_\tau)]$.

*Part 1: application of prop. B.10.* We now derive the subdifferential of $g(\lambda)$ for $\lambda > 0$. For $t = \pi \in \mathcal{T}$, let $a_t = -(1 - \delta - \mathbb{E}^\pi[r_\tau(\mathcal{D}_\tau)])$ and $b_t = \mathbb{E}^\pi[\tau]$. Then

$$-g(\lambda) = \sup\{a_t \lambda - b_t : t \in \mathcal{T}\}.$$

By prop. B.10 it follows that for $\lambda \in \mathbb{R}$

$$\partial(-g(\lambda)) = \cap_{\epsilon > 0} \overline{\text{conv}}\{a_t : t \in \mathcal{T}_\epsilon(\lambda)\}, \quad \mathcal{T}_\epsilon(\lambda) := \{t \in \mathcal{T} : a_t \lambda - b_t \geq -g(\lambda) - \epsilon\},$$

where $\partial(-g(\lambda))$ is the subdifferential of $-g$.

Using that $\mathcal{S}(\lambda) \subseteq \mathcal{T}_\epsilon(\lambda)$ for all $\epsilon \geq 0$, we conclude that for any $d \in \partial(-g(\lambda))$ we have

$$-d \geq \inf_{\pi \in \mathcal{S}(\lambda)} 1 - \delta - \mathbb{E}^\pi[r_\tau(\mathcal{D}_\tau)] = 1 - \delta - \mathbb{E}^{\pi^\star_\lambda}[r_\tau(\mathcal{D}_\tau)].$$

Defining $\phi(\lambda) = \mathbb{E}^{\pi^\star_\lambda}[r_\tau(\mathcal{D}_\tau)]$, we note that $\phi(\lambda) \in \Phi(\lambda)$.

Next, consider the case $\lambda = 0$. From prop. B.10, we have

$$B(0) := \{y \in \mathbb{R} : (y, 0) \in (\overline{\text{conv}}\{(a_t, b_t) : t \in \mathcal{T}\})_\infty\}.$$

Let $C = \overline{\text{conv}}\{(a_t, b_t) : t \in \mathcal{T}\}$. For a nonempty closed convex set $C \subset \mathbb{R}^2$ the recession cone is defined as $C_\infty = \{y \in \mathbb{R}^2 | \forall x \in C, \forall t \geq 0 : x + yt \in C\}$. By contradiction, assume $(y, 0) \in C_\infty$, then for any $(a, b) \in C, t \geq 0$ we have $(a + yt, b) \in C$. However, $a_t \in [-1 + \delta, \delta]$ for all $t \in \mathcal{T}$, bounded. Hence, there exists $t > 0$ such that $a + yt \notin [-1 + \delta, \delta]$, which is a contradiction. Since $y$ is arbitrary, only the 0 element satisfies the condition, and thus $B(0) = \{0\}$. Therefore the set of subdifferentials in 0 is simply given by $\partial(-g(0)) = \cap_{\epsilon > 0} \overline{\text{conv}}\{a_t : t \in \mathcal{T}_\epsilon(0)\}$.

*Part 2: critical points.* Define the following value:

$$\bar{\lambda} := \inf\{\lambda \geq 0 : \phi(\lambda) \geq 1 - \delta\}.$$

By lem. B.9, since $\phi(\lambda) \in \Phi(\lambda)$ we know that $\bar{\lambda} < \infty$. Then, for any $0 \leq \lambda < \bar{\lambda}$ we have that any $d \in \partial(-g(\lambda))$ satisfies

$$-d \geq 1 - \delta - \phi^+(\lambda) > 0$$

hence $-d > 0$ for $0 < \lambda < \bar{\lambda}$. Since $-d$ is a superdifferential (we are maximizing $g$!), any critical solution $\lambda^\star \in \arg\max g(\lambda)$ satisfies $\lambda^\star \in [\bar{\lambda}, \infty)$. Furthermore, such critical point exists: as $\lambda \to \infty$, the differential $-d$ becomes negative (since $\phi^+(\lambda) \to 1$ by lem. B.9), implying that $g(\lambda)$ decreases. Hence, the maximum is attained in $\lambda^\star \in [0, \infty)$.

Then, since $\lambda^\star \in [\bar{\lambda}, \infty)$, we have that

$$1 - \delta \leq \phi(\lambda^\star) = \mathbb{E}^{\pi^\star_{\lambda^\star}}[r_{\tau^\star_{\lambda^\star}}(\mathcal{D}_{\tau^\star_{\lambda^\star}})] = \mathbb{E}^{\pi^\star_{\lambda^\star}}[\mathbf{P}_{\tau^\star_{\lambda^\star}}(H^\star_M = \hat{H}_{\tau^\star_{\lambda^\star}}|\mathcal{D}_{\tau^\star_{\lambda^\star}})] = \mathbf{P}^{\pi^\star_{\lambda^\star}}(H^\star_M = \hat{H}_{\tau^\star_{\lambda^\star}}).$$

$\square$

**Training-time certification and stopping.** To obtain formal guarantees on correctness, note that sequentially testing the accuracy $\hat{p}$ during training, where

$$\hat{p} = \frac{1}{K} \sum_{i=1}^{K} \mathbf{1}_{\{H_i^\star = \arg\max_H I_\phi(H | \mathcal{D}_\tau^{(i)})\}},$$

may not imply $\delta$-correctness, unless we adopt the correct sequential test. Alternatively, one can simply avoid to sequentially test the accuracy of the model, and simply stop training at a fixed number of epochs $T_E$, where $T_E$ is fixed a priori. Then, the user can test the model $(\theta_{T_E}, \phi_{T_E})$ on a number of i.i.d. trajectories to evaluate a lower bound on the accuracy of the model (e.g., through a simple Hoeffding bound).

On the other hand, if we want to stop training as soon as the model is $\delta$-correct, then we should employ a sequential testing procedure to decide when to stop. To that aim, we need to introduce an additional confidence $\delta' \in (0, 1/2)$. This value becomes the desired correctness of the method, while $\delta$ is chosen to satisfy $\delta < \delta'$, with $\delta' - \delta$ sufficiently large. The reason is simple: by forcing the model to be more accurate, it becomes easier (for the test that we use) to detect that the accuracy crossed the threshold $1 - \delta'$.

We employ the following procedure.

- At epoch $t = 1, 2, \ldots$ we evaluate $(\theta_t, \phi_t, \lambda_t)$ on $K$ i.i.d. rollouts (independent of the training updates at epoch $t$, and sampled on $K$ different environments $M_i \sim \mathcal{P}$).
  - For each $n \in \{1, \ldots, K\}$ let $Z_{t,n} \in \{0, 1\}$ indicate whether the returned hypothesis on that rollout equals $H_i^\star$ on the $i$-th environment, and set $X_t = \frac{1}{K} \sum_{n=1}^{K} Z_{t,n}$ with conditional mean $p_t := \mathbb{E}[Z_{t,1} \mid \mathcal{F}_{t-1}]$.
- We adopt the rule: fix $\eta \in (0, 1)$ and stop at the first epoch

$$T := \inf\left\{t \geq 1 : \frac{1}{t}\sum_{s=1}^{t} X_s \geq (1 - \delta') + \frac{1}{t}\sqrt{2\left(1 + \frac{1-\delta'}{B}t\right)\ln\left(\frac{\sqrt{1 + \frac{1-\delta'}{B}t}}{\eta}\right)}\right\},$$

then *freeze* the parameters and return $(\theta_T, \phi_T, \lambda_T)$. The proposition below (an anytime bound via a mixture-martingale) guarantees

$$\mathbb{P}\left(\exists t : \frac{1}{t}\sum_{s=1}^{t} X_s \text{ crosses the boundary} \,\Big|\, \sup_{t \geq 1} p_t \leq 1 - \delta'\right) \leq \eta,$$

so, with probability at least $1 - \eta$, we only stop when the global null "$p_t \leq 1 - \delta'$ for all epochs" is false, i.e., there exists some $s \leq T$ with $p_s > 1 - \delta'$. If, in addition, the epochwise performance is nondecreasing ($p_1 \leq p_2 \leq \cdots$, a property that typically arises when the method converges), then $p_T \geq 1 - \delta'$, and the returned model is $\delta'$-correct with confidence $1 - \eta$.

**Proposition B.12** (Training correctness). *Let $(\mathcal{F}_t)_{t \geq 1}$ be the training filtration, with $\mathcal{F}_t = \sigma(x_1, a_1, x_2, \ldots, a_{t-1}, x_t)$. For each epoch $t$ let $Z_{t,1}, \ldots, Z_{t,K}$ be conditionally i.i.d. $\mathrm{Ber}(p_t)$ given $\mathcal{F}_{t-1}$, with $X_t := K^{-1}\sum_{n=1}^{K} Z_{t,n}$ and $p_t := \mathbb{E}[Z_{t,1}|\mathcal{F}_{t-1}]$. For $\eta \in (0, 1)$, define the stopping time*

$$T := \inf\left\{t \geq 1 : \frac{1}{t}\sum_{s=1}^{t} X_s \geq (1 - \delta') + \frac{1}{t}\sqrt{2\left(1 + \frac{1-\delta'}{K}t\right)\ln\left(\frac{\sqrt{1 + \frac{1-\delta'}{K}t}}{\eta}\right)}\right\}.$$

*Assume further that with probability at least $1 - \xi$ there exists a (finite, $\mathcal{F}_t$-stopping) time $t_0$ such that*

$$p_{t+1} \geq p_t \quad \forall t \geq t_0, \quad \text{and} \quad \sup_{t \geq t_0} p_t \geq 1 - \delta. \tag{23}$$

*Then*

$$\mathbb{P}(T < \infty, p_T \geq 1 - \delta') \geq \mathbb{P}(T < \infty) - (\eta + \xi).$$

*Proof.* Let $\mathcal{E}$ be the event in (23). The idea is to construct an event $\mathcal{G}$ such that $\{T < \infty\} \cap \mathcal{E} \cap \mathcal{G} \subseteq \{T < \infty, p_T \geq 1 - \delta'\}$. On $\mathcal{E}$, define the stopping time $S = \inf\{t \geq t_0 : p_t \geq 1 - \delta'\}$.

We let $\mathcal{G} = \{T \geq S\}$. Clearly, by prop. B.13 we have that on $\mathcal{E}$ the event $\mathcal{G}$ happens with probability at-least $1 - \eta$. Therefore, on $\{T < \infty\} \cap \mathcal{E} \cap \mathcal{G}$ we have that $T < \infty$ and $p_T \geq p_S \geq 1 - \delta'$, hence $\{T < \infty\} \cap \mathcal{E} \cap \mathcal{G} \subseteq \{T < \infty, \, p_T \geq 1 - \delta'\}$.

Let $A = \{T < \infty\}$. Using the following decomposition of $A$ in disjoint regions

$$A = (A \cap \mathcal{E} \cap \mathcal{G}) \cup (A \cap \mathcal{E} \cap \mathcal{G}^c) \cup (A \cap \mathcal{E}^c),$$

we obtain

$$\begin{aligned}
\mathbb{P}(T < \infty, \, p_T \geq 1 - \delta') &\geq \Pr(\{T < \infty\} \cap \mathcal{E} \cap \mathcal{G}), \\
&= \mathbb{P}(\{T < \infty\}) - \mathbb{P}(\{T < \infty\} \cap \mathcal{E} \cap \mathcal{G}^c) - \mathbb{P}(\{T < \infty\} \cap \mathcal{E}^c), \\
&\geq \mathbb{P}(\{T < \infty\}) - \mathbb{P}(\mathcal{E} \cap \mathcal{G}^c) - \mathbb{P}(\mathcal{E}^c), \\
&= \mathbb{P}(\{T < \infty\}) - \eta - \xi.
\end{aligned}$$

$\square$

*Remark.* Condition (23) is one natural way to formalize "monotone convergence from some epoch $t_0$ with high probability." Under (23) the first epoch $S$ with $p_S \geq 1 - \delta'$ exists a.s., and the anytime validity ensures we do not stop before $S$ except with probability at most $\eta$. Hence, upon stopping, the returned snapshot is $\delta'$-correct with probability at least $1 - \eta - \xi$. We also note that the event $\mathcal{E}$ is a consequence of lem. B.9, from the monotonicity of $\mathbb{E}^\pi[r_\tau(\mathcal{D}_\tau)]$ in $\lambda$.

Lastly, note that the test that we use considers the average over epochs of $X_n$. If $\delta' = \delta$, this average may take a long time to converge to $1 - \delta$, and even to cross the threshold. Hence, we practically run the algorithm with confidence $\delta$, with $\delta < \delta'$ (where $\delta'$ is the desired accuracy), so that $(1/t) \sum_n X_n$ converges to $1 - \delta > 1 - \delta'$ (and this fact can help the test trigger earlier).

Lastly, we prove an anytime bound via a mixture-martingale on the repeated tests on $p_t$.

**Proposition B.13.** *For all $t \geq 1, B \in \mathbb{N}$, let $X_t = \frac{1}{B} \sum_{n=1}^{B} Z_{t,n}$, where, for each $t$, $(Z_{t,n})_{n=1}^{B}$ are conditionally i.i.d. Bernoulli random variables with mean $p_t$ given $\mathcal{F}_{t-1}$, where $\mathcal{F}_t = \sigma(x_1, a_1, x_2, \ldots, a_{t-1}, x_t)$. Assume that $\sup_{t \geq 1} p_t \leq 1 - \delta'$. Then, for all $\eta \in (0, 1)$ we have*

$$\mathbb{P}\left( \exists t \geq 1 : \frac{1}{t} \sum_{n=1}^{t} X_n \geq (1 - \delta') + \frac{1}{t} \sqrt{2\left(1 + \frac{1 - \delta'}{B}t\right) \ln\left(\frac{\sqrt{1 + \frac{1-\delta'}{B}t}}{\eta}\right)} \right) \leq \eta.$$

*Proof.* Let $S_t = \sum_{i=1}^{t} X_i - p_i$.

For any $\lambda \geq 0, \alpha > 0$, let $\phi_t(\lambda) = \frac{\alpha B}{p_t(1-p_t)} \ln \mathbb{E}[e^{\frac{\lambda}{B}(Z - p_t)} | \mathcal{F}_{t-1}]$ be the (normalized) CGF of $Z \sim \text{Ber}(p_t)$. Define $V_t = \frac{p_t(1-p_t)}{\alpha}$ be a measure of variance. Then, for $M_t(\lambda) = \exp\left(\lambda S_t - \sum_{i=1}^{t} \phi_i(\lambda) V_i\right)$ we get that

$$\begin{aligned}
\mathbb{E}\left[M_t(\lambda) | \mathcal{F}_{t-1}\right] &= \mathbb{E}\left[\exp\left(\sum_{i=1}^{t} \lambda(X_i - p_i) - \phi_i(\lambda) V_i\right) | \mathcal{F}_{t-1}\right], \\
&= M_{t-1}(\lambda) \mathbb{E}\left[\exp\left(\lambda(X_t - p_t) - \phi_t(\lambda) V_t\right) | \mathcal{F}_{t-1}\right], \\
&= M_{t-1}(\lambda) \mathbb{E}\left[\exp\left(\frac{\lambda(\sum_{n=1}^{B} Z_{t,n} - p_t)}{B} - B \ln \mathbb{E}[e^{\frac{\lambda}{B}(Z - p_t)} | \mathcal{F}_{t-1}]\right) | \mathcal{F}_{t-1}\right], \\
&= M_{t-1}(\lambda) \frac{\mathbb{E}\left[\exp\left(\frac{\lambda(\sum_{n=1}^{B} Z_{t,n} - p_t)}{B}\right) | \mathcal{F}_{t-1}\right]}{\mathbb{E}[\exp\left(\frac{\lambda}{B}(Z - p_t)\right) | \mathcal{F}_{t-1}]^B}, \\
&= M_{t-1}(\lambda) \frac{\mathbb{E}\left[\exp\left(\frac{\lambda}{B}(Z - p_t)\right) | \mathcal{F}_{t-1}\right]^B}{\mathbb{E}[\exp\left(\frac{\lambda}{B}(Z - p_t)\right) | \mathcal{F}_{t-1}]^B} = M_{t-1}(\lambda).
\end{aligned}$$

Since $M_t \geq 0, \lambda \geq 0$, we have that $M_t(\lambda)$ is a non-negative martingale (hence, also a super-martingale).

We use the method of mixtures to integrate $M_t(\lambda)$ over a prior over $\lambda$. To do so, we need to find an appropriate lower bound on $M_t$. Consider then $\phi_t(\lambda)$: we can use the fat that $\phi_t(\lambda) \leq \alpha\lambda^2/(2B)$ from the sub-gaussianity of $Z$. Then, choose a prior $\pi(\mathrm{d}\lambda) = \sqrt{2/\pi}e^{-\lambda^2/2}\mathrm{d}\lambda$ (a half normal). We obtain

$$
\begin{aligned}
\int_0^\infty M_t(\lambda)\,\pi(\mathrm{d}\lambda) &= \int_0^\infty \exp\left(\lambda S_t - \sum_{i=1}^t \phi_i(\lambda)V_i\right)\pi(\mathrm{d}\lambda), \\
&\geq \int_0^\infty \exp\left(\lambda S_t - \sum_{i=1}^t \frac{\alpha\lambda^2}{2B}\frac{1-\delta'}{\alpha}\right)\pi(\mathrm{d}\lambda), \quad (V_i \leq p_t/\alpha \leq (1-\delta')/\alpha) \\
&= \sqrt{2/\pi}\int_0^\infty \exp\left(\lambda S_t - \frac{\lambda^2}{2}\left[1 + \frac{(1-\delta')}{B}t\right]\right)\mathrm{d}\lambda.
\end{aligned}
$$

Since the Gaussian integral satisfies

$$
\int_0^\infty e^{-a\lambda^2 + b\lambda}\,\mathrm{d}\lambda = e^{\frac{b^2}{4a}}\int_0^\infty e^{-a\left(\lambda - \frac{b}{2a}\right)^2}\,\mathrm{d}\lambda \geq e^{\frac{b^2}{4a}}\int_0^\infty e^{-ax^2}\,\mathrm{d}x = e^{\frac{b^2}{4a}}\frac{1}{2}\sqrt{\frac{\pi}{a}},
$$

for $v_t = (1-\delta')t/B$ we can lower bound the integral over $M_t(\lambda)$ as

$$
\int_0^\infty M_t(\lambda)\,\pi(\mathrm{d}\lambda) \geq \frac{1}{\sqrt{1+v_t}}e^{\frac{S_t^2}{2(1+v_t)}}.
$$

Therefore, by Ville's inequality we obtain

$$
\mathbb{P}\left(\exists t \geq 1 : \int_0^\infty M_t(\lambda)\,\pi(\mathrm{d}\lambda) \geq \frac{1}{\eta}\right) \leq \eta.
$$

Therefore, with probability $1 - \eta$ for all $t \geq 1$ we have

$$
\frac{1}{\sqrt{1+v_t}}e^{\frac{S_t^2}{2(1+v_t)}} < \frac{1}{\eta} \Rightarrow S_t < \sqrt{2(1+v_t)\ln\left(\frac{\sqrt{1+v_t}}{\eta}\right)}
$$

Since $S_t \geq \sum_{i=1}^t X_i - (1-\delta')$, we obtain the desired result. $\qquad\square$

## B.2 META-TRAINING: FINITE SAMPLE ANALYSIS

---

**Algorithm 2** Finite-budget idealized ICPE training

---

**Inputs:** Value function space $\mathcal{F}$, posterior rewards $r_N$, reference measure $\mu$.

**Init:** choose $Q^{(0)} \in \mathcal{F}$; for $k \geq 0$ set $\pi^{(k+1)} = \mathcal{G}(Q^{(k)})$ with $\pi_t^{(k+1)}(z) \in \arg\max_a Q_t^{(k)}(z, a)$.

1: **for** $k = 0, \ldots$ **do**
2:    **Sampling:** draw a batch $B_k = \{(z^{(i)}, a^{(i)}, t^{(i)})\}_{i=1}^B$ i.i.d. from $\mu$.
3:    **for** each $(z, a, t) \in B_k$ **do**
4:       sample $x' \sim \bar{P}_t(\cdot \mid z, a)$ and let $z' = (z, a, x')$.
5:       set targets:

$$\hat{Q}_t^{(k+1)}(z, a) \leftarrow \begin{cases} Q_{t+1}^{(k)}(z', \pi_{t+1}^{(k+1)}(z')), & t < N, \\ r_N(z'), & t = N. \end{cases}$$

6:    **end for**
7:    **Regression:** for each $t = 1, \ldots, N$, fit

$$Q_t^{(k+1)} \in \arg\min_{Q \in \mathcal{F}_t} \widehat{\mathcal{L}}_t(Q, \hat{Q}^{(k+1)}; B_{k,t}).$$

8: **end for**

---

We work in the Bayes/history MDP induced by the prior over environments. Let $\{\mathcal{Z}_t\}_{t=1}^N$ be the history spaces (as in ICPE), with $z \in \mathcal{Z}_t$ encoding the full trajectory prefix up to stage $t$. The terminal (posterior) reward is

$$r_N(z) = \max_{H \in \mathcal{H}} \mathbb{P}_N(H^\star = H | \mathcal{D}_N = z),$$

with $z \in \mathcal{Z}_N$.

**Reference sampling law.** Let $\mu$ be a probability distribution on triples $(z, a, t) \in \bigcup_{t=1}^N (\mathcal{Z}_t \times \mathcal{A} \times \{t\})$, with stage marginals $\mu_t$ on $\mathcal{Z}_t \times \mathcal{A}$. During training, all regression samples are drawn i.i.d. from $\mu$: this measure represents sampling from idealized replay buffer.

The next sample is then sampled according to $x' \sim \bar{P}_t(\cdot \mid z, a)$, where

$$\bar{P}_t(x' \in X | z_t, a) := \int_{\mathcal{M}^\sharp} P_t(X | z_t, a) \, R_t(\mathrm{d}M | z_t), \tag{24}$$

so that the next history is $z_{t+1} = (z_t, a, x')$.

**Function class and Stage-wise Bellman operators.** We let $\mathcal{F} \subset \prod_{t=1}^N \{\mathcal{Z}_t \times \mathcal{A} \to [0, 1]\}$ be the $Q$-function class. For $Q = (Q_t)_{t=1}^N \in \mathcal{F}$, define the stagewise greedy policy

$$\pi_t = \mathcal{G}(Q_t) \in \arg\max_{a \in \mathcal{A}} Q_t(\cdot, a) \qquad (t = 1, \ldots, N).$$

For a nonstationary policy $\pi = (\pi_1, \ldots, \pi_N)$ and a $Q$-array $Q = (Q_1, \ldots, Q_N)$, define for $t = 1, \ldots, N, z \in \mathcal{Z}_t$ the operator

$$[\Gamma_t^\pi Q](z, a) := \mathbb{E}_{x' \sim \bar{P}_t(\cdot | z, a)} [Q_{t+1}(z', \pi_{t+1}(z'))], \qquad z' = (z, a, x').$$

At the last stage, with terminal posterior reward $r_N$

$$[\Gamma_N^\pi Q](z, a) := r_N(z), \quad \forall a.$$

We also define the optimal operator

$$\Gamma_t^\star Q(z, a) := \mathbb{E}\left[ \mathbf{1}_{\{t < N\}} \max_{a'} Q_{t+1}(z', a') + \mathbf{1}_{\{t = N\}} r_N(z) \right].$$

In the following we write $\Gamma^\pi Q = (\Gamma_1^\pi Q, \ldots, \Gamma_N^\pi Q)$ and similarly for $\Gamma^\star, \mathcal{G}$.

Given a policy $\pi$ we also indicate by $Q_t^\pi(z, a)$ the true $Q$-value of $\pi$ at $(z, a, t)$. Similarly, we define the value as $V_t^\pi(z) = Q_t^\pi(z, \pi_t(z))$. We similarly define the optimal value $V_t^\star$.

**Concentrability (w.r.t. $\mu$).**   Let $\nu_t^\pi$ be the occupancy measure on $(z_t)$ at stage $t$ under policy $\pi$, when the initial history is sampled from the prior-induced initial distribution $\rho$ (where $\rho$ is the initial observation distribution in $M$), that is

$$\nu_t^\pi(\cdot) := \mathbb{E}_{M \sim \mathcal{P}}^\pi \left[ \rho P_1 \cdots P_{t-1}(\cdot) \right].$$

with $\nu_1(\cdot) = \mathbb{E}_{M \sim \mathcal{P}}[\rho(\cdot)]$.

**Assumption 3.** *For all $t = 1, \ldots, N$ we assume that $\nu_t^\pi \ll \mu_t^Z$, where $\mu_t^Z$ is the marginal of $\mu$ on $\mathcal{Z}_t$.*

Define then the concentrability coefficients

$$c_\infty(t) := \sup_\pi \left\| \frac{\mathrm{d}\nu_t^\pi}{\mathrm{d}\mu_t^Z} \right\|_\infty.$$

Recall also assumption 1, which states that there exist probability measures $\lambda_0, \lambda$ on $(\mathcal{X}, \mathcal{B}(\mathcal{X}))$ such that, for all $(\rho, P) \in \mathcal{M}^\sharp$, $s \in \mathbb{N}$, and $(z, a) \in \mathcal{Z}_s \times \mathcal{A}$,

$$\rho(\cdot) \ll \lambda_0(\cdot) \quad \text{and} \quad P_s(\cdot|z, a) \ll \lambda(\cdot).$$

We make the following additional assumption.

**Assumption 4.** *For all $(\rho, P) \in M^\sharp, s \in \mathbb{N}$ and $(z, a) \in \mathcal{Z}_s \times A$ we assume that $\mathrm{d}\rho_M(\cdot)/\mathrm{d}\lambda_0(\cdot)$ and $\mathrm{d}P_s(\cdot|z, a)/\mathrm{d}\lambda(\cdot)$ are upper semicontinuous.*

Hence, by compactness and upper semicontinuity there exist $L_0, L_1$ such that

$$\sup_{\rho \in M^\sharp} \sup_{x \in \mathcal{X}} \frac{\mathrm{d}\rho}{\mathrm{d}\lambda_0}(x) \le L_0, \quad \max_{t=1,\ldots,N} \sup_{P \in M^\sharp} \sup_{x \in \mathcal{X}, z \in \mathcal{Z}_t, a \in \mathcal{A}} \frac{\mathrm{d}P_t(\cdot|z, a)}{\mathrm{d}\lambda}(x) \le L_1.$$

Consequently, one can bound $c_\infty$ as follows

$$c_\infty(t) \le L_0 L_1^t.$$

**Function class and losses.**   Let $\mathcal{F}_t$ be a hypothesis class for $Q_t$. We indicate by $B_k \subset (\mathcal{Z} \times \mathcal{A} \times [N])$ a batch of samples, and by $B_{k,t} = \{(z, a, s) \in B_k : s = t\}$. Hence, for a batch $B_k$ with targets $\hat{Q}^{(k+1)}$, define the empirical squared loss

$$\widehat{\mathcal{L}}_t(Q, \hat{Q}^{(k+1)}; B_{k,t}) := \frac{1}{|B_{k,t}|} \sum_{(z_t, a) \in B_{k,t}} \left( Q_t(z_t, a) - \hat{Q}_t^{(k+1)}(z_t, a) \right)^2,$$

and the Monte Carlo targets are

$$\hat{Q}_t^{(k+1)}(z, a) = \begin{cases} Q_{t+1}^{(k)}(z', \pi_{t+1}^{(k+1)}(z')), & t < N, \\ r_N(z'), & t = N, \end{cases} \quad z' = (z, a, x'), \ x' \sim \bar{P}_t(\cdot|z, a).$$

with $\pi^{(k+1)} = \mathcal{G}(Q^{(k)})$. We also define the true loss

$$\mathcal{L}(Q^{(k)}, Q^{(k-1)}) := \mathbb{E}_{(z, a, t) \sim \mu} \left[ \left( \Gamma_t^{\pi^{(k)}} Q^{(k-1)}(z, a) - Q_t^{(k)}(z, a) \right)^2 \right],$$

and $\mathcal{L}_t(Q^{(k)}, Q^{(k-1)}) := \mathbb{E}_{(z, a) \sim \mu_t} \left[ \left( \Gamma_t^{\pi^{(k)}} Q^{(k-1)}(z, a) - Q_t^{(k)}(z, a) \right)^2 \right]$. In the following, for simplicity, we also write $\mathcal{L}_{k,t} := \mathcal{L}_t(Q^{(k)}, Q^{(k-1)})$.

In each epoch $k$ a regression problem is solved, where the training set $\{(z^{(i)}, a^{(i)}, t^{(i)}, \hat{Q}^{(k+1)})\}$ and $\hat{Q}_{t^{(i)}}^{(k+1)}(z^{(i)}, a^{(i)})$ is an unbiased estimate of the target defined by $\Gamma_t Q$.

### B.2.1   MAIN RESULTS

The main results are the following ones.

**Error propagation.** We first obtain a result on the error propagation that bounds the sub-optimality of the policy at training epoch $k$. This result holds for a general function space $\mathcal{F} = (\mathcal{F}_t)_{t=1}^N$. In the following, we denote the overlal value of a policy $\pi$ by $J(\pi) = \mathbb{E}_{\mathcal{P}}^\pi[r_N(z_N)]$ and define $\pi^\star \in \arg\sup_\pi J(\pi)$.

**Theorem B.14** (Sub-optimality of policy $\pi^{(k)}$). *Let $J(\pi) = \mathbb{E}_{\mathcal{P}}^\pi[r_N(z_N)]$ and $\pi^\star \in \arg\sup_\pi J(\pi)$. For $k \geq N + 1$, we have that*

$$|J(\pi^\star) - J(\pi^{(k)})| \leq \|w\|_2 \left[ \sqrt{S_{k-1}^{(1,N)}} + 2\sqrt{(N+1)\sum_{u=k-N}^{k} S_u^{(2,N)} + \sqrt{D_k^{(1,N)}}} \right]$$

*where $w = (w_u)_{u=1}^N$ is the vector of concentrability coefficients, with $w_u := c_\infty(u)\kappa_u$; $S_m^{(a,b)} = \sum_{u=a}^b \mathcal{L}_{m,u}$ is the sum of losses for epoch $m$ along the timesteps $(a, a+1, \ldots, b)$; $D_m^{(a,b)} = \sum_{u=a}^b \mathcal{L}_{m-u,u}$ is the diagonal sum of losses.*

**Finite-sample performance bound.** We now show how the losses that appear in the previous result can be bounded to derive a finite-sample performance bounds.

To approximate the target, for each $t = 1, \ldots, N$ we consider a linear function space $\mathcal{F}_t$ of dimension $d_t$ with bounded basis function $\{\varphi_{t,i}\}_{i=1}^{d_t} \|\varphi_{t,i}\|_\infty \leq C_b$. For each $t$ we consider a linear family with parameter $\alpha_t \in \mathbb{R}^{d_t}$ and features $\phi_t : \mathcal{Z}_t \times \mathcal{A} \to \mathbb{R}^{d_t}$, thus $\mathcal{F}_t = \{(z, a) \mapsto \phi_t(z, a)^\top \alpha_t : \alpha_t \in \mathbb{R}^{d_t}\}$.

At epoch $k$ regression returns a linear predictor $\tilde{Q}_t^{(k)}$. We then define the $Q$-function used by the algorithm as the truncation $Q_t^{(k)} = \mathbb{T}(\tilde{Q}_t^{(k)})$. In the analysis, $Q_t^{(k)}$ always denotes this truncated version.

**Theorem B.15** (Fixed-budget finite-sample training error). *Fix $\delta \in (0, 1)$ and choose $\delta' = \delta/(4kN)$. Suppose (i) the features are bounded, $\sup_{z,a} \|\phi_t(z,a)\|_2 \leq C_b$; (ii) concentrability holds with coefficients $c_\infty(t)$ and $\kappa_t$; and (iii) the batch size satisfies*

$$B \geq \frac{2}{p_{\min}\eta^2} \log \frac{4kN}{\delta}.$$

*for some $\eta \in (0, 1)$. Then, for $k \geq N + 1$, with probability at least $1 - \delta$,*

$$|J(\pi^\star) - J(\pi^{(k)})| \leq O\left( NC_0 \left[ \sqrt{\sum_{t=1}^N \beta_t^2} + \sqrt{\sum_{t=1}^N \frac{d_t}{(1-\eta)p_{\min}B} \log \frac{4kN}{\delta}} \right] \right),$$

*where $\beta_t = \sup_{Q \in \mathcal{F}, \pi} \inf_{f \in \mathcal{F}_t} \|\Gamma_t^\pi Q - f\|_{\mu_t}$ and $C_0 = \left( \sum_{t=1}^N c_\infty(t)^2 \kappa_t^2 \right)^{1/2}$, and $p_{\min} = \min_t \mu(t)$, where $\mu(t)$ is the marginal over timesteps of the buffer distribution.*

**Intuition.** thm. B.14 shows that the performance gap $J(\pi^\star) - J(\pi^{(k)})$ is controlled by how well each step approximates the Bellman update: the terms $S_{k-1}^{(1,N)}$, $\sum_{u=k-N}^{k} S_u^{(2,N)}$, and $D_k^{(1,N)}$ aggregate the single–step squared Bellman residuals $\mathcal{L}_{k,t}$ across time and across a window of epochs, and the concentrability vector $w = (c_\infty(t) \cdot \kappa_t)_{t=1}^N$ measures how much these local errors can be amplified when propagated along the trajectory distribution. The finite-sample bound in thm. B.15 then replaces these abstract residuals with explicit statistical quantities: each $\mathcal{L}_{k,t}$ is bounded by an *approximation* term $\beta_t$ (how well the function class can represent an exact update) plus an *estimation* term that decays as $\sqrt{d_t/((1-\eta)p_{\min}B)}$. In other words, the final rate cleanly separates an approximation error, captured by $\sqrt{\sum_t \beta_t^2}$, from a sample error, captured by $\sqrt{\sum_t d_t/((1-\eta)p_{\min}B)}$, and both are scaled by the horizon $N$ and the concentrability constant $C_0$, which quantify how errors accumulate along the history MDP.

### B.2.2 Convergence Analysis: Proof of thm. B.14

To prove thm. B.14, we follow an analysis similar to the one in (Scherrer et al., 2012). However, note that their setting is quite different from ours: we do not have the classical discounted Bellman operator, and as a consequence the proofs are different.

We begin by defining the following key quantities :

1. At iteration $k$ we indicate by $\pi_t^{(k)} = \mathcal{G}(Q_t^{(k-1)})$ the greedy policy.

2. The one-step evaluation $Q_t^{(k)} = \Gamma_t^{\pi^{(k)}} Q^{(k-1)} + \epsilon_t^{(k)}$, and $\epsilon_t^{(k)}$ is the regression error and $Q_t^{(k)}$ is computed according to $Q_t^{(k)} \in \arg\min_{Q \in \mathcal{F}_t} \widehat{\mathcal{L}}_t(Q, \hat{Q}^{(k)})$ for all $t = 1, \ldots, N$. We also write $V_t^{(k)}(z) = [\Gamma_t^{\pi^{(k)}} Q^{(k-1)}](z, \pi_t^{(k)}(z))$.

3. We define $V_t^{\pi^{(k)}}$ to be the value of $\pi^{(k)}$ under $\Gamma$, that is, the true value of $\pi^{(k)}$ with rewards $r$. Similarly, we define $Q_t^{\pi^{(k)}}$ to be the $Q$-value.

4. The Bellman residual w.r.t. the next greedy policy: $b_t^{(k)} = Q_t^{(k)} - \Gamma_t^{\pi^{(k+1)}} Q^{(k)}$

5. The performance gap $\ell_t^{(k)} = V_t^\star - V_t^{\pi^{(k)}} \geq 0$.

6. Distance before approximation: $d_t^{(k)} = V_t^\star - V_t^{(k)}$.

7. The shift: $s_t^{(k)} = V_t^{(k)} - V_t^{\pi^{(k)}}$.

Therefore $\ell_t^{(k)} = s_t^{(k)} + d_t^{(k)}$: this is the quantity we wish to bound for $t = 1$. The proof of thm. B.14 is based on bounding $s_t$ and $d_t$ separately. We begin by proving a lemma that we use repeatedly in all of the proofs.

**Lemma B.16.** *Let $\kappa_t := \sqrt{\operatorname{esssup}_z \max_a \frac{1}{\mu_t(a|z)}}$. Let $\mu_t^Z$ be the marginal of $\mu_t$ on $\mathcal{Z}_t$. Then, for any $t$, measurable function $f_t : \mathcal{Z}_t \to \mathcal{A}$, we have that*

$$\mathbb{E}_{z \sim \mu_t^Z}[|\epsilon_t^{(k)}(z, f_t(z))|] \leq \kappa_t \sqrt{\mathcal{L}_t(Q^{(k)}, Q^{(k-1)})}.$$

*Proof.* Consider $|\epsilon_t^{(k)}(z, f_t(z))|$, then

$$\mathbb{E}_{z \sim \mu_t^Z}[|\epsilon_t^{(k)}(z, f_t(z))|] = \mathbb{E}_{\mu_t^Z}\left[\sum_a \mathbf{1}_{\{f_t(z)=a\}} |\epsilon_t^{(k)}(z, a)|\right],$$

$$= \mathbb{E}_{z \sim \mu_t^Z}\left[\sum_a \sqrt{\frac{\mu_t(a|z)}{\mu_t(a|z)}} \mathbf{1}_{\{f_t(z)=a\}} |\epsilon_t^{(k)}(z, a)|\right],$$

$$\leq \sqrt{\mathbb{E}_{z \sim \mu_t^Z}\left[\sum_a \frac{\mathbf{1}_{\{f_t(z)=a\}}}{\mu_t(a|z)}\right] \mathbb{E}_{z \sim \mu_t^Z}\left[\sum_a |\epsilon_t^{(k)}(z, a)|^2 \mu_t(a|z)\right]},$$

$$\text{(Cauchy-Schwartz)}$$

$$\leq \sqrt{\mathbb{E}_{z \sim \mu_t^Z}\left[\frac{1}{\mu_t(f_t(z)|z)}\right] \mathbb{E}_{(z,a) \sim \mu_t}\left[|\epsilon_t^{(k)}(z, a)|^2\right]},$$

$$\leq \kappa_t \sqrt{\mathcal{L}_t(Q^{(k)}, Q^{(k-1)})}. \qquad \text{(by definition)}$$

$\square$

We now have the bound on $d_t^{(k+1)}$.

**Lemma B.17.** *For $t = 1, \ldots, N$, and all $k \geq 1$ we have that*

$$\mathbb{E}_{z \sim \nu_t^{\pi^{(k+1)}}}\left[d_t^{(k+1)}(z)\right] \leq \mathbb{E}_{z \sim \nu_t^{\pi^{(k+1)}}}\left[b_t^{(k)}(z, \pi_t^{(k+1)}(z))\right] + \sum_{j=0}^{N-t} c_\infty(t+j) \kappa_{t+j} \sqrt{\mathcal{L}_{k-j, t+j}}.$$

*Proof.* Consider $d_N^{(k)}(z) = V_N^\star(z) - V_N^{(k)}(z) = r_N(z) - r_N(z) = 0$. Then $d_N^{(k)}(z) = 0$ for all $z \in \mathcal{Z}_N$.

For $t < N$ we have

$$
\begin{aligned}
d_t^{(k+1)}(z) &= V_t^\star(z) - [\Gamma_t^{\pi^{(k+1)}} Q^{(k)}](z, \pi_t^{(k+1)}(z)), \\
&= \max_a Q_t^\star(z, a) - [\Gamma_t^{\pi^{(k+1)}} Q^{(k)}](z, \pi_t^{(k+1)}(z)), \\
&= \max_a Q_t^\star(z, a) - [\Gamma_t^{\pi^{(k+1)}} Q^{(k)}](z, \pi_t^{(k+1)}(z)) \pm Q_t^{(k)}(z, \pi_t^{(k+1)}(z)), \\
&= \max_a Q_t^\star(z, a) - Q_t^{(k)}(z, \pi_t^{(k+1)}(z)) + Q_t^{(k)}(z, \pi_t^{(k+1)}(z)) - [\Gamma_t^{\pi^{(k+1)}} Q^{(k)}](z, \pi_t^{(k+1)}(z)), \\
&= \max_a Q_t^\star(z, a) - Q_t^{(k)}(z, \pi_t^{(k+1)}(z)) + b_t^{(k)}(z, \pi_t^{(k+1)}(z)), \\
&\leq \max_a [Q_t^\star(z, a) - Q_t^{(k)}(z, a)] + b_t^{(k)}(z, \pi_t^{(k+1)}(z)),
\end{aligned}
$$

where the last step follows from the greediness of $\pi_t^{(k+1)}$ w.r.t. $Q_t^{(k)}$. Define $\Delta_t^{(k)}(z, a) = Q_t^\star(z, a) - Q_t^{(k)}(z, a)$ and $\Delta_t^{(k)}(z) = \max_a \Delta_t^{(k)}(z, a)$. Then

$$
d_t^{(k+1)}(z) \leq \Delta_t^{(k)}(z) + b_t^{(k)}(z, \pi_t^{(k+1)}(z)).
$$

We are now tasked with bounding $\Delta_t^{(k)}$. To that aim, observe that $Q_t^\star = \Gamma_t^\star Q^\star$, thus

$$
\begin{aligned}
\Delta_t^{(k)}(z, a) &= [\Gamma_t^\star Q^\star](z, a) - [\Gamma_t^{\pi^{(k)}} Q^{(k-1)}](z, a) - \epsilon_t^{(k)}(z, a), \\
&= \mathbb{E}_{z' \sim \bar{P}(\cdot | z, a)}[V_{t+1}^\star(z') - Q_{t+1}^{(k-1)}(z', \pi_{t+1}^{(k)}(z'))] - \epsilon_t^{(k)}(z, a), \\
&= \mathbb{E}_{z' \sim \bar{P}(\cdot | z, a)}[\max_{a'} Q_{t+1}^\star(z', a') - Q_{t+1}^{(k-1)}(z', \pi_{t+1}^{(k)}(z'))] - \epsilon_t^{(k)}(z, a), \\
&\leq \mathbb{E}_{z' \sim \bar{P}(\cdot | z, a)} \left[ \Delta_{t+1}^{(k-1)}(z') \right] - \epsilon_t^{(k)}(z, a). \qquad \text{(similarly to above)}
\end{aligned}
$$

Therefore, we have that

$$
\Delta_t^{(k)}(z) \leq \max_a \mathbb{E}_{z' \sim \bar{P}(\cdot | z, a)} \left[ \Delta_{t+1}^{(k-1)}(z') \right] + \max_a |\epsilon_t^{(k)}(z, a)|,
$$

from which we can recursively show that

$$
\mathbb{E}_{z \sim \nu_t^{\pi^{(k+1)}}} \left[ \Delta_t^{(k)}(z) \right] \leq \sum_{j=0}^{N-t} c_\infty(t + j) \kappa_{t+j} \sqrt{\mathcal{L}_{k-j, t+j}}
$$

using lem. B.16.

$\square$

We now have the bound on $s_t^{(k)}$.

**Lemma B.18.** *For all $t = 1, \ldots, N$ and $k$ we have that*

$$
\mathbb{E}_{z \sim \nu_t^{\pi^{(k)}}}[s_t^{(k)}(z)] = \sum_{j=1}^{N-t} \mathbb{E}_{z' \sim \nu_{t+j}^{\pi^{(k)}}} \left[ b_{t+j}^{(k-1)}(z', \pi_{t+j}^{(k)}(z')) \right].
$$

*Proof.* First, note that $s_N^{(k)}(z) = 0$. Then, for $t < N$ we have

$$
\begin{aligned}
s_t^{(k)}(z) &= V_t^{(k)}(z) - V_t^{\pi^{(k)}}(z), \\
&= [\Gamma_t^{\pi^{(k)}} Q^{(k-1)}](z, \pi_t^{(k)}(z)) - Q_t^{\pi^{(k)}}(z, \pi_t^{(k)}(z)), \\
&= \mathbb{E}_{x' | z, \pi_t^{(k)}(z)} \left[ Q_{t+1}^{(k-1)}(z', \pi_{t+1}^{(k)}(z')) - V_{t+1}^{\pi^{(k)}}(z') \,\Big|\, z' = (z, \pi_t^{(k)}(z), x') \right], \\
&= \mathbb{E}_{x' | z, \pi_t^{(k)}(z)} \left[ b_{t+1}^{(k-1)}(z', \pi_{t+1}^{(k)}(z')) + [\Gamma_{t+1}^{\pi^{(k)}} Q^{(k-1)}](z', \pi_{t+1}^{(k)}(z')) - V_{t+1}^{\pi^{(k)}}(z') \,\Big|\, z' = (z, \pi_t^{(k)}(z), x') \right], \\
&= \mathbb{E}_{x' | z, \pi_t^{(k)}(z)} \left[ b_{t+1}^{(k-1)}(z', \pi_{t+1}^{(k)}(z')) + V_{t+1}^{(k)}(z') - V_{t+1}^{\pi^{(k)}}(z') \,\Big|\, z' = (z, \pi_t^{(k)}(z), x') \right], \\
&= \mathbb{E}_{x' | z, \pi_t^{(k)}(z)} \left[ b_{t+1}^{(k-1)}(z', \pi_{t+1}^{(k)}(z')) + s_{t+1}^{(k)}(z') \,\Big|\, z' = (z, \pi_t^{(k)}(z), x') \right], \\
&= \sum_{j=1}^{N-t} \mathbb{E} \left[ b_{t+j}^{(k-1)}(z', \pi_{t+j}^{(k)}(z')) \,\Big|\, z_t = z, \text{ then follow } \pi^{(k)} \right].
\end{aligned}
$$

Therefore

$$\mathbb{E}_{z\sim\nu_t^{\pi^{(k)}}}[s_t^{(k)}(z)] = \sum_{j=1}^{N-t} \mathbb{E}_{z'\sim\nu_{t+j}^{\pi^{(k)}}} \left[ b_{t+j}^{(k-1)}(z', \pi_{t+j}^{(k)}(z')) \right].$$

$\square$

**Lemma B.19.** *For all $t = 1, \ldots, N, \forall a \in \mathcal{A}$ and epochs $k \geq N$ we have that*

$$\mathbb{E}_{z\sim\nu_t^{\pi^{(k)}}} \left[ b_t^{(k)}(z, a) \,\middle|\, \pi^{(k)}, \ldots, \pi^{(k-(N-t)+1)} \right]$$

$$\leq c_\infty(t)\kappa_t \sqrt{\mathcal{L}_{k,t}} + c_\infty(N)\kappa_N \left[ \sqrt{\mathcal{L}_{k-(N-t),N}} + \sqrt{\mathcal{L}_{k-(N-t-1),N}} \right]$$

$$+ \sum_{j=1}^{N-t-1} c_\infty(t+j)\kappa_{t+j} \left[ \sqrt{\mathcal{L}_{k-j,t+j}} + \sqrt{\mathcal{L}_{k-j+1,t+j}} \right].$$

*Proof. (One-step recursion).* For $t < N$ write

$$b_t^{(k)} = Q_t^{(k)} - \Gamma_t^{\pi^{(k+1)}} Q^{(k)},$$
$$= \Gamma_t^{\pi^{(k)}} Q^{(k-1)} - \Gamma_t^{\pi^{(k+1)}} Q^{(k)} + \epsilon_t^{(k)}.$$

Use the definition of $\Gamma_t^\pi$, we have that at time $t = N$ we get $b_N^{(k)} = \epsilon_N^{(k)}$. For $t < N$ we get

$$b_t^{(k)}(z, a) = \epsilon_t^{(k)}(z, a) + \mathbb{E}_{x'\sim\bar{P}_t(\cdot|z,a)} \left[ Q_{t+1}^{(k-1)} \left( z', \pi_{t+1}^{(k)}(z') \right) - Q_{t+1}^{(k)} \left( z', \pi_{t+1}^{(k+1)}(z') \right) \,\middle|\, z' = (z, a, x') \right],$$

$$= \epsilon_t^{(k)}(z, a) + \mathbb{E}_{x'\sim\bar{P}_t(\cdot|z,a)} \left[ Q_{t+1}^{(k-1)} \left( z', \pi_{t+1}^{(k)}(z') \right) - Q_{t+1}^{(k)} \left( z', \pi_{t+1}^{(k+1)}(z') \right) \pm Q_{t+1}^{(k)}(z', \pi_{t+1}^{(k)}(z')) \,\middle|\, z' = (z, a, x') \right],$$

$$= \epsilon_t^{(k)}(z, a) + \mathbb{E}_{x'\sim\bar{P}_t(\cdot|z,a)} \Big[ Q_{t+1}^{(k-1)} \left( z', \pi_{t+1}^{(k)}(z') \right) - Q_{t+1}^{(k)}(z', \pi_{t+1}^{(k)}(z'))$$
$$+ Q_{t+1}^{(k)}(z', \pi_{t+1}^{(k)}(z')) - Q_{t+1}^{(k)} \left( z', \pi_{t+1}^{(k+1)}(z') \right) \,\middle|\, z' = (z, a, x') \Big],$$

$$\leq \epsilon_t^{(k)}(z, a) + \mathbb{E}_{x'\sim\bar{P}_t(\cdot|z,a)} \left[ Q_{t+1}^{(k-1)} \left( z', \pi_{t+1}^{(k)}(z') \right) - Q_{t+1}^{(k)}(z', \pi_{t+1}^{(k)}(z')) \,\middle|\, z' = (z, a, x') \right],$$

where in the last inequality, we used that $Q_{t+1}^{(k)}(z', \pi_{t+1}^{(k+1)}(z')) \geq Q_{t+1}^{(k)}(z', \pi_{t+1}^{(k)}(z'))$ (since $\pi^{(k+1)} = \mathcal{G}(Q^{(k)})$). Now, using the definition $b_{t+1}^{(k)} = Q_{t+1}^{(k)} - \Gamma_{t+1}^{\pi^{(k+1)}} Q^{(k)}$, we continue with $Q_{t+1}^{(k)} = \Gamma_{t+1}^{\pi^{(k)}} Q^{(k-1)} + \epsilon_{t+1}^{(k)}$

$$= \epsilon_t^{(k)}(z, a) + \mathbb{E}_{x'\sim\bar{P}_t(\cdot|z,a)} \left[ Q_{t+1}^{(k-1)} \left( z', \pi_{t+1}^{(k)}(z') \right) - [\Gamma_{t+1}^{\pi^{(k)}} Q^{(k-1)} + \epsilon_{t+1}^{(k)}](z', \pi_{t+1}^{(k)}(z')) \,\middle|\, z' = (z, a, x') \right],$$

$$= \epsilon_t^{(k)}(z, a) + \mathbb{E}_{x'\sim\bar{P}_t(\cdot|z,a)} \left[ b_{t+1}^{(k-1)} \left( z', \pi_{t+1}^{(k)}(z') \right) - \epsilon_{t+1}^{(k)}(z', \pi_{t+1}^{(k)}(z')) \,\middle|\, z' = (z, a, x') \right].$$

Therefore

$$b_t^{(k)}(z, a) \leq \epsilon_t^{(k)}(z, a) + \mathbb{E}_{x'\sim\bar{P}_t(\cdot|z,a)} \left[ b_{t+1}^{(k-1)} \left( z', \pi_{t+1}^{(k)}(z') \right) - \epsilon_{t+1}^{(k)}(z', \pi_{t+1}^{(k)}(z')) \,\middle|\, z' = (z, a, x') \right].$$

Thus

$$b_t^{(k)}(z, a) \leq |\epsilon_t^{(k)}(z, a)| + \mathbb{E}_{x'\sim\bar{P}_t(\cdot|z,a)} \left[ b_{t+1}^{(k-1)} \left( z', \pi_{t+1}^{(k)}(z') \right) + |\epsilon_{t+1}^{(k)}(z', \pi_{t+1}^{(k)}(z'))| \,\middle|\, z' = (z, a, x') \right].$$

*(Unrolling).* Let $z_{t+1}$ be the state observed after taking action $a$ in $z$ in round $t$. Then denote the successive states by $z_{t+j}$, sampled by following $\pi^{(k)}$. Then, unrolling the last upper bound yields

$$b_t^{(k)}(z, a) \leq |\epsilon_t^{(k)}(z, a)| + \mathbb{E} \Big[ |\epsilon_{t+1}^{(k-1)}(z_{t+1}, \pi_{t+1}^{(k)}(z_{t+1}))| + b_{t+2}^{(k-2)} \left( z_{t+2}, \pi_{t+2}^{(k-1)}(z_{t+2}) \right)$$
$$+ |\epsilon_{t+2}^{(k-1)}(z_{t+2}, \pi_{t+2}^{(k-1)}(z_{t+2}))| + |\epsilon_{t+1}^{(k)}(z_{t+1}, \pi_{t+1}^{(k)}(z_{t+1}))| \,\middle|\, z_t = z, a_t = a \Big],$$

$$\leq \mathbb{E} \Big[ |\epsilon_t^{(k)}(z_t, a_t)| + |\epsilon_N^{(k-(N-t))}(z_N, \pi_N^{(k-(N-t)+1)}(z_N))| + |\epsilon_N^{(k-(N-t)+1)}(z_N, \pi_N^{(k-(N-t)+1)}(z_N))|$$
$$+ \sum_{j=1}^{N-t-1} |\epsilon_{t+j}^{(k-j)}(z_{t+j}, \pi_{t+j}^{(k-j+1)}(z_{t+j}))| + |\epsilon_{t+j}^{(k-j+1)}(z_{t+j}, \pi_{t+j}^{(k-j+1)}(z_{t+j}))| \,\middle|\, z_t = z, a_t = a \Big].$$

Therefore, using lem. B.16

$$\mathbb{E}_{z\sim\nu_t^{\pi^{(k)}}}\left[b_t^{(k)}(z,a)\,\Big|\,\pi^{(k)},\ldots,\pi^{(k-(N-t)+1)}\right]$$
$$\leq c_\infty(t)\kappa_t\sqrt{\mathcal{L}_{k,t}} + c_\infty(N)\kappa_N\left[\sqrt{\mathcal{L}_{k-(N-t),N}} + \sqrt{\mathcal{L}_{k-(N-t-1),N}}\right]$$
$$+ \sum_{j=1}^{N-t-1} c_\infty(t+j)\kappa_{t+j}\left[\sqrt{\mathcal{L}_{k-j,t+j}} + \sqrt{\mathcal{L}_{k-j+1,t+j}}\right].$$

$\square$

We now prove the bound on $J(\pi^\star) - J(\pi^{(k)})$ in thm. B.14, where $J(\pi) = \mathbb{E}_{\mathcal{P}}^\pi[r_N(z_N)]$.

*Proof of thm. B.14.* Note that $0 \leq J(\pi^\star) - J(\pi^{(k)}) = \mathbb{E}_{z\sim\nu_1}[\ell_1^{(k)}(z)]$. Using the decomposition $\ell_1^{(k)} = s_1^{(k)} + d_1^{(k)}$ and lems. B.17 and B.18, we obtain that

$$\mathbb{E}_{z\sim\nu_1}[\ell_1^{(k)}(z)] = \mathbb{E}_{z\sim\nu_1}\left[s_1^{(k)}(z)\right] + \mathbb{E}_{z\sim\nu_1}\left[d_1^{(k)}(z)\right],$$
$$\leq \sum_{u=2}^N \mathbb{E}_{z'\sim\nu_u^{\pi^{(k)}}}\left[b_u^{(k-1)}(z',\pi_u^{(k)}(z'))\right] + \mathbb{E}_{z\sim\nu_1^{\pi^{(k)}}}\left[b_1^{(k-1)}(z,\pi_1^{(k)}(z))\right] + \sum_{u=1}^N c_\infty(u)\kappa_u\sqrt{\mathcal{L}_{k-u,u}},$$
$$= \sum_{u=1}^N \mathbb{E}_{z'\sim\nu_u^{\pi^{(k)}}}\left[b_u^{(k-1)}(z',\pi_u^{(k)}(z'))\right] + \sum_{u=1}^N c_\infty(u)\kappa_u\sqrt{\mathcal{L}_{k-u,u}}.$$

From lem. B.19 we know that

$$\mathbb{E}_{z\sim\nu_t^{\pi^{(k)}}}\left[b_t^{(k)}(z,a)\,\Big|\,\pi^{(k)},\ldots,\pi^{(k-(N-t)+1)}\right] \leq c_\infty(t)\kappa_t\sqrt{\mathcal{L}_{k,t}} + c_\infty(N)\kappa_N\left[\sqrt{\mathcal{L}_{k-(N-t),N}} + \sqrt{\mathcal{L}_{k-(N-t-1),N}}\right]$$
$$+ \sum_{j=1}^{N-t-1} c_\infty(t+j)\kappa_{t+j}\left[\sqrt{\mathcal{L}_{k-j,t+j}} + \sqrt{\mathcal{L}_{k-j+1,t+j}}\right].$$

hence

$$\mathbb{E}_{z\sim\nu_t^{\pi^{(k)}}}\left[b_t^{(k-1)}(z,\pi_t^{(k)}(z))\right]$$
$$\leq c_\infty(t)\kappa_t\sqrt{\mathcal{L}_{k-1,t}} + c_\infty(N)\kappa_N\left[\sqrt{\mathcal{L}_{k-(N-t)-1,N}} + \sqrt{\mathcal{L}_{k-(N-t),N}}\right]$$
$$+ \sum_{j=1}^{N-t-1} c_\infty(t+j)\kappa_{t+j}\left[\sqrt{\mathcal{L}_{k-j-1,t+j}} + \sqrt{\mathcal{L}_{k-j,t+j}}\right].$$

Using the last inequality we obtain

$$\sum_{u=1}^N \mathbb{E}_{z'\sim\nu_u^{\pi^{(k)}}}\left[b_u^{(k-1)}(z',\pi_u^{(k)}(z'))\right] \leq \sum_{u=1}^N c_\infty(u)\kappa_u\sqrt{\mathcal{L}_{k-1,u}} + c_\infty(N)\kappa_N\left[\sqrt{\mathcal{L}_{k-(N-u)-1,N}} + \sqrt{\mathcal{L}_{k-(N-u),N}}\right]$$
$$+ \sum_{u=1}^N \sum_{j=1}^{N-u-1} c_\infty(u+j)\kappa_{u+j}\left[\sqrt{\mathcal{L}_{k-j-1,u+j}} + \sqrt{\mathcal{L}_{k-j,u+j}}\right] = (\star).$$

Re-indexing the last term by $s = u + j$, we have

$$(\star) = \sum_{u=1}^N c_\infty(u)\kappa_u\sqrt{\mathcal{L}_{k-1,u}} + c_\infty(N)\kappa_N\left[\sqrt{\mathcal{L}_{k-(N-u)-1,N}} + \sqrt{\mathcal{L}_{k-(N-u),N}}\right]$$
$$+ \sum_{s=2}^{N-1} c_\infty(s)\kappa_s\sum_{j=1}^{s-1}\left[\sqrt{\mathcal{L}_{k-j-1,s}} + \sqrt{\mathcal{L}_{k-j,s}}\right].$$

At this point, define $w_u = c_\infty(u)\kappa_u, w^{(a,b)} = (w_u)_{u=a}^b, Z_m^{(a,b)} = \sum_{u=a}^b \sqrt{\mathcal{L}_{u,m}}$ and $S_m^{(a,b)} = \sum_{u=a}^b \mathcal{L}_{m,u}$. Then,

$$(\star) \leq \|w^{(1,N)}\|_2 \sqrt{S_{k-1}^{(1,N)}} + w_N \left[ Z_N^{(k-N,k-1)} + Z_N^{(k-N+1,k)} \right] \qquad \text{(Applied Cauchy-Schwartz)}$$

$$+ \sum_{s=2}^{N-1} w_s \left[ Z_s^{(k-s,k-2)} + Z_s^{(k-s+1,k-1)} \right],$$

$$\leq \|w^{(1,N)}\|_2 \sqrt{S_{k-1}^{(1,N)}} + 2 \sum_{s=2}^N w_s Z_s^{(k-s,k)}, \qquad \text{(Increased the sum range of } Z\text{)}$$

$$\leq \|w^{(1,N)}\|_2 \left[ \sqrt{S_{k-1}^{(1,N)}} + 2 \sqrt{\sum_{s=2}^N \left( Z_s^{(k-s,k)} \right)^2} \right]. \qquad \text{(By } \|w^{(2,N)}\|_2 \leq \|w^{(1,N)}\|_2\text{)}$$

Now, observe that

$$\left( Z_s^{(k-s,k)} \right)^2 = \left( \sum_{u=k-s}^k \sqrt{\mathcal{L}_{u,s}} \right)^2 \leq (s+1) \left( \sum_{u=k-s}^k \mathcal{L}_{u,s} \right) \leq (N+1) \sum_{u=k-N}^k \mathcal{L}_{u,s},$$

therefore

$$\sum_{s=2}^N (N+1) \sum_{u=k-N}^k \mathcal{L}_{u,s} = (N+1) \sum_{u=k-N}^k \sum_{s=2}^N \mathcal{L}_{u,s} = (N+1) \sum_{u=k-N}^k S_u^{(2,N)}.$$

Thus

$$(\star) \leq \|w^{(1,N)}\|_2 \left[ \sqrt{S_{k-1}^{(1,N)}} + 2 \sqrt{(N+1) \sum_{u=k-N}^k S_u^{(2,N)}} \right].$$

We can plug this back into the original bound on $\ell_1^{(k)}$. Define $D_s^{(a,b)} = \sum_{u=a}^b \mathcal{L}_{s-u,u}$ to be the diagonal sum of losses, then

$$\mathbb{E}_{z \sim \nu_1}[|\ell_1^{(k)}(z)|] \leq \|w^{(1,N)}\|_2 \left[ \sqrt{S_{k-1}^{(1,N)}} + 2 \sqrt{(N+1) \sum_{u=k-N}^k S_u^{(2,N)}} \right] + \sum_{u=1}^N c_\infty(u) \kappa_u \sqrt{\mathcal{L}_{k-u,u}},$$

$$\leq \|w^{(1,N)}\|_2 \left[ \sqrt{S_{k-1}^{(1,N)}} + 2 \sqrt{(N+1) \sum_{u=k-N}^k S_u^{(2,N)}} \right] + \|w^{(1,N)}\|_2 \sqrt{D_k^{(1,N)}},$$

$$\leq \|w^{(1,N)}\|_2 \left[ \sqrt{S_{k-1}^{(1,N)}} + 2 \sqrt{(N+1) \sum_{u=k-N}^k S_u^{(2,N)}} + \sqrt{D_k^{(1,N)}} \right].$$

$\square$

### B.2.3 FINITE SAMPLE ANALYSIS: PROOF OF THM. B.15

*Proof of thm. B.15.* **Preliminaries.** We now consider the error due to the evaluation step. In each epoch $k$ a regression problem is solved, where the training set $\{(z^{(i)}, a^{(i)}, t^{(i)}, \hat{Q}^{(k+1)})\}$ and $\hat{Q}_{t^{(i)}}^{(k+1)}(z^{(i)}, a^{(i)})$ is an unbiased estimate of the target defined by $\Gamma_t Q$.

To approximate the target, for each $t = 1, \ldots, N$ we consider a linear function space $\mathcal{F}_t$ of dimension $d_t$ with bounded basis function $\{\varphi_{t,i}\}_{i=1}^{d_t} \|\varphi_{t,i}\|_\infty \leq C_b$. For each $t$ we consider a linear family with parameter $\alpha_t \in \mathbb{R}^{d_t}$ and features $\phi_t : \mathcal{Z}_t \times \mathcal{A} \to \mathbb{R}^{d_t}$, thus $\mathcal{F}_t = \{(z,a) \mapsto \phi_t(z,a)^\top \alpha_t : \alpha_t \in \mathbb{R}^{d_t}\}$.

Recall the losses

$$\mathcal{L}_{k,t} := \mathbb{E}_{(z,a)\sim\mu_t}\left[\left(Y_t^{(k)}(z,a) - Q_t^{(k)}(z,a)\right)^2\right],$$

where

$$Y_t^{(k)}(z,a) = [\Gamma_t^{\pi^{(k)}} Q^{(k-1)}](z,a)$$

and we also define the error $\epsilon_t^{(k)} = Q_t^{(k)} - Y_t^{(k)}$.

For a batch $B_k$ we denote by $B_{k,t} = \{(z,a,s) \in B_k : s = t\}$ the elements in that batch of size $t$, and let $n_{k,t} = |B_{k,t}|$.

We then let $Y_{k,t} = (Y_t^{(k)}(z,a))_{(z,a)\in B_{k,t}}$ (*true targets*) and $\hat{Q}_{k,t} = (\hat{Q}_t^{(k)}(z,a))_{(z,a)\in B_{k,t}}$ (*noisy targets*), and define $\mathcal{F}_{k,t} = \{\Phi_{k,t}\alpha_t : \alpha_t \in \mathbb{R}^{d_t}\}$, where $\Phi_{k,t} = (\phi_t(z,a)^\top)_{(z,a)\in B_{k,t}}$ is a matrix where each row corresponds to the features of some $(z,a) \in B_{k,t}$. We then denote by $\Pi_{k,t}$ the $L_2(\hat{\mu}_{k,t})$-projection on $\mathcal{F}_{k,t}$, where $\hat{\mu}_{k,t}(z,a) = \sum_{(z',a')\in B_{k,t}} \delta_{(z',a')}(z,a)$ is the empirical norm at epoch $k$ for timestep $t$. We also define $\Pi_t$ to be the $L_2(\mu_t)$ projection on $\mathcal{F}_t$, where $\mu_t$ is the marginal over trajectories of $\mu$ at timesteps $t$.

We set $\tilde{Q}_{k,t} := \Pi_{k,t}\hat{Q}_{k,t} = (\tilde{Q}_t^{(k)}(z,a))_{(z,a)\in B_{k,t}}$, where $\tilde{Q}_t^{(k)}$ is the result of linear regression and its truncation (by 1) is $Q_t^{(k)}$ ($Q_t^{(k)} = \mathbb{T}(\tilde{Q}_t^{(k)})$). Define also $\hat{Y}_{k,t} := \Pi_{k,t}Y_{k,t}$ and the errors $\xi_{k,t} := Y_{k,t} - \hat{Q}_{k,t}$ and $\hat{\xi}_{k,t} := \Pi_{k,t}\xi_{k,t}$. We note that $\xi_{k,t}$ has mean 0, and $|(\xi_{k,t})_i| \leq 1$.

In the following, we denote by $\|f\|_{\mu_t} = \sqrt{\int f(z,a)^2 \mathrm{d}\mu_t(z,a)}$ the $L_2(\mu_t)$-norm of $f$, and similarly we also indicate the $L_2(\hat{\mu}_{k,t})$-norm (empirical) by $\|f\|_{\hat{\mu}_{k,t}} = \sqrt{\frac{1}{n_{k,t}}\sum_{(z,a)\in B_{k,t}} f(z,a)^2}$.

**Bounding the error.** Our goal is to bound

$$\|e_t^{(k)}\|_{\mu_t} = \|Y_t^{(k)} - Q_t^{(k)}\|_{\mu_t} = \|Y_t^{(k)} - \mathbb{T}(\tilde{Q}_t^{(k)})\|_{\mu_t}.$$

By a variation of theorem 11.2 in (Györfi et al., 2002) (see (Lazaric et al., 2012) corollary 12), we also know that

$$\|Y_t^{(k)} - \mathbb{T}(\tilde{Q}_t^{(k)})\|_{\mu_t} - 2\|Y_{k,t} - \tilde{Q}_{k,t}\|_{\hat{\mu}_{k,t}} \leq 24\sqrt{\frac{2}{n_{k,t}}\Lambda(n_{k,t}, d_t, \delta')}.$$

with probability at least $1 - \delta'$, where $\Lambda(n_{k,t}, d_t, \delta') = 2(d_t + 1)\log(n_{k,t}) + \log(\frac{e}{\delta'}) + \log\left(9(12e)^{2(d_t+1)}\right)$. Therefore

$$\|Y_t^{(k)} - \mathbb{T}(\tilde{Q}_t^{(k)})\|_{\mu_t} \leq 2\|Y_{k,t} - \tilde{Q}_{k,t}\|_{\hat{\mu}_{k,t}} + 24\sqrt{\frac{2}{n_{k,t}}\Lambda(n_{k,t}, d_t, \delta')}.$$

So, for each $t$ the error is

$$\|Y_{k,t} - \tilde{Q}_{k,t}\|_{\hat{\mu}_{k,t}} \leq \|\tilde{Q}_{k,t} - \hat{Y}_{k,t}\|_{\hat{\mu}_{k,t}} + \|Y_{k,t} - \hat{Y}_{k,t}\|_{\hat{\mu}_{k,t}} = \|\hat{\xi}_{k,t}\|_{\hat{\mu}_{k,t}} + \|Y_{k,t} - \hat{Y}_{k,t}\|_{\hat{\mu}_{k,t}}.$$

Furthermore $\|\hat{\xi}_{k,t}\|_{\hat{\mu}_{k,t}}^2 = \langle\hat{\xi}_{k,t}, \hat{\xi}_{k,t}\rangle = \langle\xi_{k,t}, \hat{\xi}_{k,t}\rangle$ by the orthogonal projection, and, by an application of a variation of Pollard's inequality (Györfi et al., 2002) we have that

$$\langle\xi_{k,t}, \hat{\xi}_{k,t}\rangle \leq 4\|\hat{\xi}_{k,t}\|_{\hat{\mu}_{k,t}}\sqrt{\frac{2}{n_{k,t}}\log\left(\frac{3(9e^2 n_{k,t})^{d_t+1}}{\delta'}\right)}$$

holds with probability at least $1 - \delta'$. Therefore, we are left with bounding $\|Y_{k,t} - \hat{Y}_{k,t}\|_{\hat{\mu}_{k,t}}$.

Define $\hat{\alpha}_t^\star$ as the parameter satisfying $f_{\hat{\alpha}_t^\star} \in \mathcal{F}_t$ such that $f_{\hat{\alpha}_t^\star}(z,a) = [\Pi_{k,t}Y_t^{(k)}](z,a)$ for all $(z,a) \in B_{k,t}$. Also define $\alpha_t^\star$ to be the optimal projection (w.r.t. $\mu_t$) of $Y_t^{(k)}$ in $\mathcal{F}_t$, i.e., $f_{\alpha_t^\star} = \Pi_t Y_t^{(k)}$.

Then, again by a variation of Theorem 11.2 Györfi et al. (2002) (see also (Lazaric et al., 2012) corollary 13), we have the following sequence of inequality

$$
\begin{aligned}
\|Y_{k,t} - \hat{Y}_{k,t}\|_{\hat{\mu}_{k,t}} &= \|Y_{k,t} - f_{\hat{\alpha}_t^\star}\|_{\hat{\mu}_{k,t}}, \\
&\leq \|Y_{k,t} - f_{\alpha_t^\star}\|_{\hat{\mu}_t}, \\
&\leq 2\|Y_t^{(k)} - f_{\alpha_t^\star}\|_{\mu_t} + 12\left(1 + \|\alpha_t^\star\|_2 \sup_{(z,a)\in\mathcal{Z}_t\times\mathcal{A}} \|\phi_t(z,a)\|_2\right)\sqrt{\frac{2}{n_{k,t}}\log\left(\frac{3}{\delta'}\right)},
\end{aligned}
$$

that hold with probability at least $1 - \delta'$. In conclusion, we have shown that

$$
\begin{aligned}
\|e_t^{(k)}\|_{\mu_t} \leq &2\left[2\|Y_t^{(k)} - f_{\alpha_t^\star}\|_{\mu_t} + 12\left(1 + \|\alpha_t^\star\|_2 \sup_{(z,a)\in\mathcal{Z}_t\times\mathcal{A}} \|\phi_t(z,a)\|_2\right)\sqrt{\frac{2}{n_{k,t}}\log\left(\frac{3}{\delta'}\right)}\right. \\
&\left. + 4\sqrt{\frac{2}{n_{k,t}}\log\left(\frac{3(9e^2 n_{k,t})^{d_t+1}}{\delta'}\right)}\right] + 24\sqrt{\frac{2}{n_{k,t}}\Lambda(n_{k,t}, d_t, \delta')}.
\end{aligned}
$$

**Union bound for the random batch.** At this point, let $\mu(t)$ be the marginal of $\mu$ over the timesteps $t = 1, \ldots, N$. Let $p_{\min} = \min_t \mu(t)$. Then $n_{k,t} := |B_{k,t}| \sim \text{Binom}(B, \mu(t))$ and

$$
\begin{aligned}
\mathbb{P}(n_{k,t} \leq (1-\eta)\mu(t)B) &\leq \exp\left(-\frac{\mu(t)B\eta^2}{2}\right), \\
&\leq \exp\left(-\frac{p_{\min}B\eta^2}{2}\right).
\end{aligned}
$$

Therefore, for a fixed $t$ for $B = \frac{2}{\eta^2 p_{\min}}\log\frac{1}{\delta'}$ we obtain that

$$
\mathbb{P}(n_{k,t} \geq (1-\eta)\mu(t)B) \geq 1 - \delta'.
$$

Therefore, by setting $\delta = 4Nk\delta'$, through a union bound, we can conclude that

$$
\|e_t^{(k)}\|_{\mu_t} \leq 4\inf_{f\in\mathcal{F}_t}\|Y_{k,t} - f\|_{\mu_t} + \eta_t((1-\eta)p_{\min}B, d_t, \delta) + \eta_t'((1-\eta)p_{\min}B, d_t, \delta),
$$

holds with probability $1 - \delta$ for all $i = 1, \ldots, k$, $t = 1, \ldots, N$, where

$$
\eta_t(n, d_t, \delta) = 32\sqrt{\frac{2}{n}\log\left(\frac{4\cdot 27 Nk(12e^2 n)^{2(d_t+1)}}{3\delta}\right)},
$$

$$
\eta_t'(n, d_t, \delta) = 24\left(1 + \|\alpha_t^\star\|_2 \sup_{(z,a)\in\mathcal{Z}_t\times\mathcal{A}} \|\phi_t(z,a)\|_2\right)\sqrt{\frac{2}{n}\log\left(\frac{12Nk}{\delta}\right)}.
$$

**Bounding $S_k^{(a,b)}$ in terms of the error.** Let

$$
\beta_t = \sup_{Q\in\mathcal{F}_t,\pi} \inf_{f\in\mathcal{F}_t} \|\Gamma_t^\pi Q - f\|_{\mu_t}.
$$

Since $S_k^{(a,b)} = \sum_{u=a}^b \mathcal{L}_{k,u}$ and $\sqrt{\mathcal{L}_{k,t}} = \|e_t^{(k)}\|_{\mu_t}$, we have that

$$
\sqrt{S_k^{(a,b)}} = \left\|\left(\sqrt{\mathcal{L}_{k,t}}\right)_{t=a}^b\right\|_2 \leq 4\|(\beta_t)_{t=a}^b\|_2 + \|(\eta_t)_{t=a}^b\|_2 + \|(\eta_t')_{t=a}^b\|_2,
$$

with probability $1 - \delta$.

Similarly, we have that

$$
\begin{aligned}
\sqrt{(N+1)\sum_{u=k-N}^k S_u^{(2,N)}} &\leq \sqrt{N+1}\sqrt{\sum_{u=k-N}^k 16\|(\beta_t)_{t=2}^N\|_2^2 + \|(\eta_t)_{t=2}^N\|_2^2 + \|(\eta_t')_{t=2}^N\|_2^2}, \\
&\leq (N+1)\sqrt{16\|(\beta_t)_{t=2}^N\|_2^2 + \|(\eta_t)_{t=2}^N\|_2^2 + \|(\eta_t')_{t=2}^N\|_2^2}, \\
&\leq (N+1)\left[4\|(\beta_t)_{t=2}^N\|_2 + \|(\eta_t)_{t=2}^N\|_2 + \|(\eta_t')_{t=2}^N\|_2\right].
\end{aligned}
$$

which holds with probability $1 - \delta$.

Lastly, we consider $\sqrt{D_k^{(1,N)}}$ where $D_k^{(1,N)} = \sum_{u=1}^{N} \mathcal{L}_{k-u,u}$. We have with probability $1 - \delta$

$$\sqrt{D_k^{(1,N)}} = \sqrt{\sum_{u=1}^{N} \mathcal{L}_{k-u,u}} \leq 4\|(\beta_t)_{t=1}^{N}\|_2 + \|(\eta_t)_{t=1}^{N}\|_2 + \|(\eta_t')_{t=1}^{N}\|_2.$$

Therefore, in conclusion

$$\sqrt{S_{k-1}^{(1,N)}} + \sqrt{(N+1)\sum_{u=k-N}^{k} S_u^{(2,N)}} + \sqrt{D_k^{(1,N)}}$$

$$\leq 4\|(\beta_t)_{t=2}^{N}\|_2 + \|(\eta_t)_{t=2}^{N}\|_2 + \|(\eta_t')_{t=2}^{N}\|_2 + (N+1)\left[4\|(\beta_t)_{t=2}^{N}\|_2 + \|(\eta_t)_{t=2}^{N}\|_2 + \|(\eta_t')_{t=2}^{N}\|_2\right],$$

$$+ 4\|(\beta_t)_{t=1}^{N}\|_2 + \|(\eta_t)_{t=1}^{N}\|_2 + \|(\eta_t')_{t=1}^{N}\|_2,$$

$$\leq (N+3)\left[4\|(\beta_t)_{t=1}^{N}\|_2 + \|(\eta_t)_{t=1}^{N}\|_2 + \|(\eta_t')_{t=1}^{N}\|_2\right].$$

**Conclusions.** Therefore, we can conclude that, up to constants and logarithmic factors, we have that with probability $1 - \delta$

$$|J(\pi^\star) - J(\pi^{(k)})| \leq O\left(NC_0\left[C_1 + \sqrt{\sum_{t=1}^{N} \frac{d_t}{(1-\eta)p_{\min}B} \log \frac{4kN}{\delta}}\right]\right)$$

provided $B \geq \frac{2}{p_{\min}\eta^2} \log \frac{4kN}{\delta}$ where $\eta \in (0,1)$, $C_0 := \sqrt{\sum_{t=1}^{N} c_\infty(t)^2 \kappa_t^2}$ and $C_1 := \sqrt{\sum_{t=1}^{N} \beta_t^2}$.

$\square$

### B.3 COMPARISON WITH INFORMATION DIRECTED SAMPLING

In pure exploration IDS (Russo & Van Roy, 2018) the main objective is to maximize the *information gain*. For example, consider the BAI problem: we set $\alpha_t(a) = \mathbb{P}(\hat{H} = a|\mathcal{D}_t)$ to be the posterior distribution of the optimal arm. Then, the information gain is defined through the following quantity

$$g_t(a) = \mathbb{E}[H(\alpha_t) - H(\alpha_{t+1})|\mathcal{D}_t, a_t = a],$$

which measures the expected reduction in entropy of the posterior distribution of the best arm due to selecting arm $a$ at time $t$.

For the BAI problem, the authors in (Russo & Van Roy, 2018) propose a *myopic* sampling policy $a_t \in \arg\max_a g_t(a)$, which only considers the information gain from the next sample. The reason for using a greedy policy stems from the fact that such a strategy is competitive with the optimal policy in problems where the information gain satisfies a property named *adaptive submodularity* (Golovin & Krause, 2011), a generalization of submodular set functions to adaptive policies. For example, in the noiseless Optimal Decision Tree problem, it is known (Zheng et al., 2005) that a greedy strategy based on the information gain is equivalent to a nearly-optimal (Dasgupta, 2004; Golovin et al., 2010; Golovin & Krause, 2011) strategy named *generalized binary search* (GBS) (Nowak, 2008; Bellala et al., 2010) , which maximizes the expected reduction of the *version space* (the space of hypotheses consistent with the data observed so far). However, for the noisy case both strategies perform poorly (Golovin et al., 2010).

The myopic pure exploration IDS strategy $a_t \in \arg\max_a g_t(a)$ can perform poorly in environments where the sampling decisions influence the observation distributions, or where an action taken at time $t$ can greatly affect the complexity of the problem at a later stage (hence, IDS can perform poorly on credit assignments problems).

**First example.** As a first example, consider a bandit problem with $K$ arms, where the reward for each arm $a_i$ is distributed according to $\mathcal{N}(\mu_i, 1)$, with priors $\mu_1 = \delta_0$ and $\mu_i \sim \mathcal{U}([0,1])$ independently for each $i \in \{2, \dots, K\}$. Thus, almost surely, the optimal arm $a^\star$ lies within $\{2, \dots, K\}$, and the goal is to estimate $a^\star$

We introduce the following twist: if arm $a_1$ is sampled exactly twice, its reward distribution changes permanently to a Dirac delta distribution $\delta_{\phi(a^\star)}$, where $\phi$ is a known invertible mapping. Consequently, sampling arm $a_1$ twice fully reveals the identity of $a^\star$. However, if arm $a_1$ has not yet been sampled, the expected immediate information gain at any step $t$ is zero, i.e., $g_t(a_1) = 0$, since arm $a_1$ is already known to be suboptimal. In contrast, the immediate information gain for any other arm remains strictly positive. Therefore, under this setting and for nontrivial values of $(\sigma, K)$, the myopic IDS strategy cannot achieve the optimal constant sample complexity, and instead scales linearly in $K$.

**Second example.** Another example is a bandit environment containing a chain of two magic actions $\{1, m\}$, where the index of the first magic action (1) is known. Action 1 reveals the index $m$, and pulling arm $m$ subsequently identifies the best arm with certainty. In this scenario, IDS is myopic and typically neglects arm 1 because of its inability to plan more than 1-step ahead in the future. However, depending on the total number of arms and reward variances, IDS may still select arm 1 if doing so significantly reduces the set of candidate best arms faster than pulling other arms (e.g., if the variance is significantly large). The following theorem illustrates the sub-optimality of IDS.

**Theorem B.20.** *Consider a bandit environment with a chain of 2 magic actions. The reward of the regular arms is $\mathcal{N}(\mu_a, 1)$ with $\mu_a \sim \mathcal{U}([0,1]), a \neq 1, m$. For $K \geq 7$ there exists $\delta_0 \in (0, 1/2)$ such that for any $\delta \leq \delta_0$, we have that IDS is not sample optimal in the fixed confidence setting.*

*Proof of thm. B.20.* Let $Y_{t,a}$ be the random reward observed upon selecting arm $a_t = a$ . We use that $g_t(a) = I_t(A^\star; Y_{t,a}) = \text{KL}\left(\frac{\mathbb{P}(A^\star, Y_{t,a}|\mathcal{D}_t)}{\mathbb{P}(A^\star|\mathcal{D}_t)\mathbb{P}(Y_{t,a}|\mathcal{D}_t)}\right)$, with $\mathcal{D}_1$ containing an empty observation. The proof relies on showing that action $a_1$ is not chosen during the first two rounds for large values of $K$.

In the proofs, for brevity, we write $\mathbb{P}_t(\cdot) = \mathbb{P}(\cdot|\mathcal{D}_t)$. Observe the following lemmas.

**Lemma B.21.** *Let $Y_{t,a}$ be the random reward observed upon selecting arm $a_t = a$ and let $S_{t,a} = \mathbf{1}\{a_t = a \text{ is magic}\}$. Under the assumption that the agent knows with absolute certainty that $a$ is magic after observing $Y_{t,a}$, we have that $I_t(A^\star; Y_{t,a}) = I_t(A^\star; Y_{t,a}, S_{t,a})$.*

*Proof.* Note that

$$I_t(A^\star; Y_{t,a}, S_{t,a}) = H_t(Y_{t,a}, S_{t,a}) - H_t(Y_{t,a}, S_t | A^\star).$$

Note that by assumption we have that $H_t(S_{t,a} | Y_{t,a}) = 0$. Then, the first term can also be rewritten as

$$H_t(Y_{t,a}, S_t) = H_t(S_{t,a} | Y_{t,a}) + H_t(Y_{t,a}) = H_t(Y_t, a).$$

Similarly, we also have $H_t(Y_{t,a}, S_{t,a} | A^\star) = H_t(S_{t,a} | Y_{t,a}, A^\star) + H_t(Y_{t,a} | A^\star) = H_t(Y_{t,a} | A^\star)$. Henceforth

$$I_t(A^\star; Y_{t,a}, S_{t,a}) = H_t(Y_{t,a}) - H_t(Y_{t,a} | A^\star) = I_t(A^\star; Y_{t,a}).$$

$\square$

Using the decomposition from the previous lemma we can rewrite the mutual information between $A^\star$ and $Y_{t,a}$ as

$$I_t(A^\star; Y_{t,a}) = I_t(A^\star; Y_{t,a}, S_{t,a}) = I_t(A^\star; Y_{t,a} | S_{t,a}) + I_t(A^\star; S_{t,a}).$$

**Lemma B.22.** *Let $\mathcal{E}_t = \{(a_1, \ldots, a_{t-1})$ are not magic actions$\}$, with $\mathcal{E}_1 = \emptyset$. Under $a_t = 1$ we have that $I_t(A^\star; Y_{t,1} | \mathcal{E}_t, a_t = 1) = \log\left(\frac{K - |\mathcal{A}_t| - 1}{K - |\mathcal{A}_t| - 2}\right)$ where $\mathcal{A}_t = \{a | \exists i < t : a_t = a\}$ is the unique number of actions chosen in $t \in \{1, \ldots, t - 1\}$.*

*Proof.* We use that $\mathbb{P}_t(S_{t,1} = 1 | a_t = 1) = 1$. Hence, for arm 1 we have

$$\begin{aligned} I_t(A^\star; S_{t,1} | \mathcal{E}_t, a_t = 1) &= H_t(S_{t,1} | \mathcal{E}_t, a_t = 1) - H_t(S_{t,1} | A^\star, \mathcal{E}_t, a_t = 1), \\ &= 0 - 0 = 0. \end{aligned}$$

Then, we have

$$\begin{aligned} I_t(A^\star; Y_{t,1} | S_{t,1}, \mathcal{E}_t, a_t = 1) &= I_t(A^\star; Y_{t,1} | S_{t,1} = 1, \mathcal{E}_t, a_t = 1), \\ &= \mathrm{KL}\left(\mathbb{P}_t(A^\star, Y_{t,1} | a_t = 1, \mathcal{E}_t) || \mathbb{P}_t(A^\star | a_t = 1, \mathcal{E}_t)\mathbb{P}_t(Y_{t,1} | a_t = 1, \mathcal{E}_t)\right), \\ &= \mathrm{KL}\left(\mathbb{P}_t(Y_{t,1} | A^\star, a_t = 1, \mathcal{E}_t) || \mathbb{P}_t(Y_{t,1} | a_t = 1, \mathcal{E}_t)\right), \\ &= \log\left(\frac{1/(K - |\mathcal{A}_t| - 2)}{1/(K - |\mathcal{A}_t| - 1)}\right), \end{aligned}$$

where we used that under $\mathcal{E}_t$ exactly $\mathcal{A}_t$ regular arms have been pulled and recognised as regular; the still-unrevealed set of candidates for the second magic arm has therefore size $K - |\mathcal{A}_t| - 1$ (since arm 1 is known to be magic). Thus the result follows from applying the previous lemma. $\square$

**Lemma B.23.** *For any un-pulled arm $a \neq 1$ at time $t$ we have that $I_t(A^\star; Y_{t,a} | \mathcal{E}_t, a_t = a) \geq \frac{1}{K - |\mathcal{A}_t| - 1} \log(K - |\mathcal{A}_t| - 2)$.*

*Proof.* To compute the mutual information we use that $I_t(A^\star; Y_{t,a} | \mathcal{E}_t) = I_t(A^\star; Y_{t,a}, S_{t,a} | \mathcal{E}_t) = I_t(A^\star; Y_{t,a} | S_{t,a}, \mathcal{E}_t) + I_t(A^\star; S_{t,a} | \mathcal{E}_t) \geq I_t(A^\star; Y_{t,a} | S_{t,a}, \mathcal{E}_t)..$ We start by computing the first term of this expression, and finding a non-trivial lower bound.

Note that for $a \neq 1$ we have

$$\begin{aligned} I_t(A^\star; Y_{t,a} | S_{t,a}, \mathcal{E}_t, a_t = a) &= \mathbb{P}_t(S_{t,a} = 0 | \mathcal{E}_t, a_t = a) I_t(A^\star; Y_{t,a} | S_{t,a} = 0, \mathcal{E}_t, a_t = a) \\ &\quad + \mathbb{P}_t(S_{t,a} = 1 | \mathcal{E}_t, a_t = a) I_t(A^\star; Y_{t,a} | S_{t,a} = 1, \mathcal{E}_t, a_t = a), \\ &\geq \frac{1}{K - |\mathcal{A}_t| - 1} I_t(A^\star; Y_{t,a} | S_{t,a} = 1, \mathcal{E}_t, a_t = a), \end{aligned}$$

where we used that under $\mathcal{E}_t$, we have a uniform prior over the remaining $K - |\mathcal{A}_t| - 1$ un-pulled arms, and the agent knows that arm 1 is magic.

If $a \neq 1$ and $S_{t,a} = 1$, then $a$ is the second magic arm. Therefore we have $\mathbb{P}_t(Y_{t,a} | A^\star, S_{t,a} = 1, \mathcal{E}_t, a_t = a) = 1$. Hence $I_t(A^\star; Y_{t,a} | S_{t,a} = 1, \mathcal{E}_t) = \log(K - |\mathcal{A}_t| - 2)$ since $Y_{t,a}$ can only take values uniformly over $K - |\mathcal{A}_t| - 2$ arms under the event $\{S_{t,a} = 1, \mathcal{E}_t, a_t = a\}$.

$\square$

**Lemma B.24.** *Assume $a_1 = j$ is a regular arm, pulled at the first timestep. Then $I_2(A^\star; Y_{2,j}|a_1 = j) \leq \frac{1}{2} \ln(1 + \frac{1}{12\sigma^2})$.*

*Proof.* First, note that

$$I_2(A^\star; Y_{2,j} \mid a_1 = j) \leq I_2(\mu_j; Y_{2,j} \mid a_1 = j) = H_2(Y_{2,j}|a_1 = j) - \frac{1}{2}\ln(2\pi e\sigma^2)$$

Then, since $\mathrm{Var}_2(Y_{2,j}|a_1 = j) = \mathrm{Var}_2(\mu_j|a_1 = j) + \sigma^2 \leq 1/12 + \sigma^2$. Therefore $H_2(Y_{2,j}|a_1 = j) \leq \frac{1}{2}\ln(2\pi e(1/12 + \sigma^2))$. Hence $I_2(A^\star; Y_{2,j}|a_1 = j) \leq \frac{1}{2}\ln(1 + \frac{1}{12\sigma^2})$. $\qquad\square$

Hence, one can verify that for $K \geq 6$ the first magic arm will never be chosen at the first timestep. Similarly, at the second timestep the first magic arm will not be chosen if $K \geq 7$.

Consider the fixed-confidence setting with some confidence level $\delta < 1/2$. Let $\mathcal{A}_1 = \{$second magic arm sampled at $t = 1\}$ and $\mathcal{A}_2 = \{$second magic arm sampled at $t = 2\}$. Then, the sample complexity of IDS satisfies $\mathbb{E}[\tau_{IDS}|\mathcal{A}_1^c, \mathcal{A}_2^c] \geq 3$ for $\delta$ sufficiently small (since the sample complexity scales as $\log(1/(2.4\delta))$).

We also have that at the first timestep the decision is uniform over $\{2, \ldots, K\}$. Lastly, if the first sampled arm is not magic, then it's a regular arm, and by the previous lemmas the information gain of such arm will be smaller than the information gain of another un-pulled arm. In fact the inequality

$$\frac{\log(x - 3)}{x - 2} > \frac{1}{2}\ln(1 + \frac{1}{12})$$

it satisfied over $x \in \{5, \ldots, 121\}$. Since it is sub-optimal to sample again the same regular arm, since the information gain on all the other arms remains the same, we have that the decision at the second timestep is again uniform over the remaining unchosen arms. Therefore

$$\begin{aligned}
\mathbb{E}[\tau_{IDS}] &= \mathbb{E}[\tau_{IDS}|\mathcal{A}_1]\mathbb{P}(\mathcal{A}_1) + \mathbb{E}[\tau_{IDS}|\mathcal{A}_1^c]\mathbb{P}(\mathcal{A}_1^c), \\
&= \frac{1}{K - 1} + \frac{K - 2}{K - 1}\mathbb{E}[\tau_{IDS}|\mathcal{A}_1^c], \\
&= \frac{1}{K - 1} + \frac{K - 2}{K - 1}\left(\mathbb{E}[\tau_{IDS}|\mathcal{A}_1^c, \mathcal{A}_2]\frac{1}{K - 2} + \mathbb{E}[\tau_{IDS}|\mathcal{A}_1^c, \mathcal{A}_2^c]\frac{K - 3}{K - 2}\right), \\
&\geq \frac{1}{K - 1} + \frac{K - 2}{K - 1}\left(2\frac{1}{K - 2} + 3\frac{K - 3}{K - 2}\right), \\
&= \frac{3}{K - 1} + 3\frac{K - 3}{K - 1},
\end{aligned}$$

which is larger than 2 for $K > 4$. Since there is a policy with sample complexity 2, we have that IDS cannot be sample optimal for $K \in \{7, \ldots, 121\}$.

Similarly, for large values of $K > 121$, resampling the same regular arm at the second timestep leads IDS to a sample complexity larger than 2. And therefore cannot be sample optimal.

$\qquad\square$

## B.4 Sample Complexity Bounds for MAB Problems with Fixed Minimum Gap

We now derive a sample complexity lower bound for a MAB problem where the minimum gap is known and the rewards are normally distributed.

Consider a MAB problem wit $K$ arms $\{1, \ldots, K\}$. To each arm $a$ is associated a reward distribution $\nu_a = \mathcal{N}(\mu_a, \sigma^2)$ that is simply a Gaussian distribution. Let $a^\star(\mu) = \arg\max_a \mu_a$, and define the gap in arm $a$ to be $\Delta_a(\mu) = \mu_{a^\star(\mu)} - \mu_a$. In the following, without loss of generality, we assume that $a^\star(\mu) = 1$.

We define the minimum gap to be $\Delta_{\min}(\mu) = \min_{a \neq a^\star(\mu)} \Delta_a(\mu)$. Assume now to know that $\Delta_{\min} \geq \Delta_0 > 0$.

Then, for any $\delta$-correct algorithm, guaranteeing that at some stopping time $\tau$ the estimated optimal arm $\hat{a}_\tau$ is $\delta$-correct, i.e., $\mathbb{P}_\mu(\hat{a}_\tau \neq a^\star(\mu)) \leq \delta$, we have the following result.

**Theorem B.25.** *Consider a model $\mu$ satisfying $\Delta_{\min} \geq \Delta_0 > 0$. Then, for any $\delta$-probably correct method* Alg*, with $\delta \in (0, 1/2)$, we have that the optimal sample complexity is bounded as*

$$\frac{1}{\max\left(\Delta_0^2, \frac{1}{\sum_{a \neq 1} 1/\Delta_a^2}\right)} \leq \inf_{\tau : \text{Alg } is \, \delta\text{-correct}} \frac{\mathbb{E}_\mu[\tau]}{2\sigma^2 \text{kl}(1 - \delta, \delta)} \leq 2 \sum_a \frac{1}{(\Delta_a + \Delta_0)^2},$$

*with $\Delta_1 = 0$ and $\text{kl}(x, y) = x \log(x/y) + (1 - x) \log((1 - x)/(1 - y))$. In particular, the solution $\omega_a \propto 1/(\Delta_a + \Delta_0)^2$ (up to a normalization constant) achieves the upper bound.*

*Proof.* **Step 1: Log-likelihood ratio.** The initial part of the proof is rather standard, and follows the same argument used in the Best Arm Identification and Best Policy Identification literature (Garivier & Kaufmann, 2016; Russo & Vannella, 2025).

Define the set of models

$$\mathcal{S} = \left\{\mu' \in \mathbb{R}^K : \Delta_{\min}(\mu') \geq \Delta_0\right\},$$

and the set of alternative models

$$\text{Alt}(\mu) = \left\{\mu' \in \mathcal{S} : \arg\max_a \mu'_a \neq 1\right\}.$$

Take the expected log-likelihood ratio between $\mu$ and $\mu' \in \text{Alt}(\mu)$ of the data observed up to $\tau$ $\Lambda_\tau = \log \frac{d\mathbb{P}_\mu(A_1, R_1, ..., A_\tau, R_\tau)}{d\mathbb{P}_{\mu'}(A_1, R_1, ..., A_\tau, R_\tau)}$, where $A_t$ is the action taken in round $t$, and $R_t$ is the reward observed upon selecting $A_t$. Then, we can write

$$\Lambda_t = \sum_a \sum_{n=1}^{t} \mathbf{1}_{\{A_n = a\}} \log \frac{f_a(R_n)}{f'_a(R_n)}$$

where $f_a, f'_a$, are, respectively, the reward density for action $a$ in the two models $\mu, \mu'$ with respect to the Lebesgue measure. Letting $N_a(t)$ denote the number of times action $a$ has been selected up to round $t$, by an application of Wald's lemma the expected log-likelihood ratio can be shown to be

$$\mathbb{E}_\mu[\Lambda_\tau] = \sum_a \mathbb{E}_\mu[N_a(\tau)] \text{KL}(\mu_a, \mu'_a)$$

where $\text{KL}(\mu_a, \mu'_a)$ is the KL divergence between two Gaussian distributions $\mathcal{N}(\mu_a, \sigma)$ and $\mathcal{N}(\mu'_a, \sigma)$ (note that we have $\sigma_1$ instead of $\sigma$ for $a = 1$).

We also know from the information processing inequality (Kaufmann et al., 2016) that $\mathbb{E}_\mu[\Lambda_\tau] \geq \sup_{\mathcal{E} \in \mathcal{M}_\tau} \text{kl}(\mathbb{P}_\mu(\mathcal{E}), \mathbb{P}_{\mu'}(\mathcal{E}))$, where $\mathcal{M}_t = \sigma(A_1, R_1, \ldots, A_t, R_t)$. We use the fact that the algorithm is $\delta$-correct: by choosing $\mathcal{E} = \{\hat{a}_\tau = a^\star\}$ we obtain that $\mathbb{E}_\mu[\Lambda_\tau] \geq \text{kl}(1 - \delta, \delta)$, since $\mathbb{P}_\mu(\mathcal{E}) \geq 1 - \delta$ and $\mathbb{P}_{\mu'}(\mathcal{E}) = 1 - \mathbb{P}_{\mu'}(\hat{a}_\tau \neq a^\star) \leq 1 - \mathbb{P}_{\mu'}(\hat{a}_\tau = \arg\max_a \mu'_a) \leq \delta$ (we also used the monotonicity properties of the Bernoulli KL divergence). Hence

$$\sum_a \mathbb{E}_\mu[N_a(\tau)] \text{KL}(\mu_a, \mu'_a) \geq \text{kl}(1 - \delta, \delta).$$

Letting $\omega_a = \mathbb{E}_\mu[N_a(\tau)] / \mathbb{E}_\mu[\tau]$, we have that

$$\mathbb{E}_\mu[\tau] \sum_a \omega_a \text{KL}(\mu_a, \mu'_a) \geq \text{kl}(1 - \delta, \delta).$$

Lastly, optimizing over $\mu' \in \text{Alt}(\mu)$ and $\omega \in \Delta(K)$ yields the bound:

$$\mathbb{E}_\mu[\tau] \geq T^\star(\mu) \text{kl}(1 - \delta, \delta),$$

where $T^\star(\mu)$ is defined as

$$(T^\star(\mu))^{-1} = \sup_{\omega \in \Delta(K)} \inf_{\mu' \in \text{Alt}(\mu)} \sum_a \omega_a \text{KL}(\mu_a, \mu'_a).$$

**Step 2: Optimization over the set of alternative models.** We now face the problem of optimizing over the set of alternative models.

Defining $\mathrm{Alt}_a = \{\mu' \in \mathbb{R}^K : \mu'_a - \mu'_b \geq \Delta_0 \ \forall b \neq a\}$, the set of alternative models can be decomposed as

$$\mathrm{Alt}(\mu) = \left\{\mu' \in \mathbb{R}^K : \arg\max_a \mu'_a \neq 1, \ \Delta_{\min}(\mu') \geq \Delta_0\right\},$$
$$= \cup_{a \neq 1} \mathrm{Alt}_a.$$

Hence, the optimization problem over the alternative models becomes

$$\inf_{\mu' \in \mathrm{Alt}(\mu)} \sum_a \omega_a \mathrm{KL}(\mu_a, \mu'_a) = \min_{\bar{a} \neq 1} \inf_{\mu' \in \mathrm{Alt}_{\bar{a}}} \sum_a \omega_a \frac{(\mu_a - \mu'_a)^2}{2\sigma^2}.$$

The inner infimum over $\mu'$ can then be written as

$$P^\star_{\bar{a}}(\omega) := \inf_{\mu' \in \mathbb{R}^K} \ \sum_a \omega_a \frac{(\mu_a - \mu'_a)^2}{2\sigma^2}. \tag{25}$$
$$\text{s.t.} \quad \mu'_{\bar{a}} - \mu'_b \geq \Delta_0 \quad \forall b \neq \bar{a}.$$

While the problem is clearly convex, it does not yield an immediate closed form solution.

To that aim, we try to derive a lower bound and an upper bound of the value of this minimization problem.

**Step 3: Upper bound on $P^\star_{\bar{a}}$.** Note that an upper bound on $\min_{\bar{a} \neq 1} P^\star_{\bar{a}}(\omega)$ can be found by finding a feasible solution $\mu'$. Consider then the solution $\mu'_1 = \mu_1 - \Delta$, $\mu'_{\bar{a}} = \mu_1$ and $\mu'_b = \mu_b$ for all other arms. Clearly We have that $\mu'_{\bar{a}} - \mu'_b \geq \Delta_0$ for all $b \neq \bar{a}$. Hence, we obtain

$$\min_{\bar{a} \neq 1} P^\star_{\bar{a}}(\omega) \leq \omega_1 \frac{\Delta_0^2}{2\sigma^2} + \min_{\bar{a} \neq 1} \omega_{\bar{a}} \frac{\Delta_{\bar{a}}^2}{2\sigma^2}.$$

At this point, one can easily note that if $\frac{\Delta_0^2}{2\sigma^2} \geq \frac{1}{2\sigma^2 \sum_{a \neq 1} \frac{1}{\Delta_a^2}}$, then $\sup_{\omega \in \Delta(K)} \min_{\bar{a} \neq 1} P^\star_{\bar{a}}(\omega) \leq \frac{\Delta_0^2}{2\sigma^2}$. This corresponds to the case where all the mass is given to $\omega_1 = 1$. Otherwise, the solution is to set $\omega_1 = 0$ and $\omega_a = \frac{1/\Delta_a^2}{\sum_b 1/\Delta_b^2}$ for $a \neq 1$.

Hence, we conclude that

$$(T^\star(\mu))^{-1} = \sup_{\omega \in \Delta(K)} \min_{\bar{a} \neq 1} P^\star_{\bar{a}}(\omega) \leq \frac{1}{2\sigma^2} \max\left(\Delta_0^2, \frac{1}{\sum_{a \neq 1} 1/\Delta_a^2}\right).$$

**Step 4: Lower bound on $P^\star_{\bar{a}}$.** For the lower bound, note that we can relax the constraint to only consider $\mu'_{\bar{a}} - \mu'_1 \geq \Delta_0$. This relaxation enlarges the feasible set, and thus the infimum of this new problem lower bounds $P^\star_{\bar{a}}(\omega)$.

By doing so, since the other arms are not constrained, by convexity of the KL divergence at the infimum we have $\mu'_b = \mu_b$ for all $b \notin \{1, \bar{a}\}$. Therefore

$$P^\star_{\bar{a}}(\omega) \geq \inf_{\mu' : \mu'_{\bar{a}} - \mu'_1 \geq \Delta_0} \sum_a \omega_a \frac{(\mu_a - \mu'_a)^2}{2\sigma^2} = \inf_{\mu' : \mu'_{\bar{a}} - \mu'_1 \geq \Delta_0} \omega_1 \frac{(\mu_1 - \mu'_1)^2}{2\sigma^2} + \omega_{\bar{a}} \frac{(\mu_{\bar{a}} - \mu'_{\bar{a}})^2}{2\sigma^2}.$$

Solving the KKT conditions we find the equivalent conditions $\mu'_{\bar{a}} = \mu'_1 + \Delta_0$ and

$$\omega_1(\mu_1 - \mu'_1) + \omega_{\bar{a}}(\mu_{\bar{a}} - \mu'_1 - \Delta_0) = 0 \Rightarrow \mu'_1 = \frac{\omega_1 \mu_1 + \omega_{\bar{a}} \mu_{\bar{a}} - \omega_{\bar{a}} \Delta_0}{\omega_1 + \omega_{\bar{a}}}.$$

Therefore

$$\mu'_{\bar{a}} = \frac{\omega_1 \mu_1 + \omega_{\bar{a}} \mu_{\bar{a}} - \omega_{\bar{a}} \Delta_0}{\omega_1 + \omega_{\bar{a}}} + \Delta_0 = \frac{\omega_1 \mu_1 + \omega_{\bar{a}} \mu_{\bar{a}} + \omega_1 \Delta_0}{\omega_1 + \omega_{\bar{a}}}.$$

Plugging these solutions back in the value of the problem, we obtain

$$P^\star_{\bar{a}}(\omega) \geq \frac{\omega_1 \omega_{\bar{a}}^2}{(\omega_1 + \omega_{\bar{a}})^2} \frac{(\mu_1 - \mu_{\bar{a}} + \Delta_0)^2}{2\sigma^2} + \frac{\omega_{\bar{a}} \omega_1^2}{(\omega_1 + \omega_{\bar{a}})^2} \frac{(\mu_{\bar{a}} - \mu_1 - \Delta_0)^2}{2\sigma^2},$$
$$= \frac{\omega_1 \omega_{\bar{a}}}{\omega_1 + \omega_{\bar{a}}} \frac{(\mu_1 - \mu_{\bar{a}} + \Delta_0)^2}{2\sigma^2},$$
$$= \frac{\omega_1 \omega_{\bar{a}}}{\omega_1 + \omega_{\bar{a}}} \frac{(\Delta_{\bar{a}} + \Delta_0)^2}{2\sigma^2}.$$

Let $\theta_a = \Delta_a + \Delta_0$, with $\theta_1 = \Delta_0$. We plug in a feasible solution $\omega_a = \frac{1/\theta_a^2}{\sum_b 1/\theta_b^2}$, yielding

$$
\begin{aligned}
(T^\star(\mu))^{-1} = \sup_{\omega \in \Delta(K)} \min_{\bar{a} \neq 1} P_{\bar{a}}^\star(\omega) &\geq \min_{\bar{a} \neq 1} \frac{1/(\theta_1 \theta_{\bar{a}})^2}{\sum_b 1/\theta_b^2 (1/\theta_1^2 + 1/\theta_{\bar{a}}^2)} \frac{\theta_{\bar{a}}^2}{2\sigma^2}, \\
&= \min_{\bar{a} \neq 1} \frac{1}{\sum_b 1/\theta_b^2 (1 + \theta_1^2/\theta_{\bar{a}}^2)} \frac{1}{2\sigma^2}, \\
&= \frac{1}{2\sigma^2 \sum_b 1/\theta_b^2} \min_{\bar{a} \neq 1} \frac{1}{1 + \theta_1^2/\theta_{\bar{a}}^2}, \\
&\geq \frac{1}{2\sigma^2 \sum_b 1/\theta_b^2} \frac{1}{1 + \theta_1^2/\Delta_0^2}, \\
&= \frac{1}{4\sigma^2 \sum_b 1/\theta_b^2}.
\end{aligned}
$$

$\square$

## B.5 Sample Complexity Lower Bound for the Magic Action MAB Problem

We now consider a special class of models that embeds information about the optimal arm in the mean reward of some of the arms. Let $\phi : \mathbb{R} \to \mathbb{R}$ be a strictly decreasing function over $\{2, \ldots, K\}$[5].

Particularly, we make the following assumptions:

1. We consider mean rewards $\mu$ satisfying $\mu_1 = \phi(\arg\max_{a \neq 1} \mu_a)$, and $\mu^\star = \max_a \mu_a > \phi(2)$. Arm 1 is called "magic action", and with this assumption we are guaranteed that the magic arm is not optimal, since

$$
\mu_1 \frac{1}{\max_a \mu_a} = \phi(\arg\max_{a \neq 1} \mu_a) \frac{1}{\max_a \mu_a} \leq \phi(2) \frac{1}{\max_a \mu_a} < 1 \Rightarrow \max_a \mu_a > \mu_1.
$$

2. The rewards are normally distributed, with a fixed known standard deviation $\sigma_1$ for the magic arm, and fixed standard deviation $\sigma$ for all the other arms.

Hence, define the set of models

$$
\mathcal{S} = \left\{ \mu \in \mathbb{R}^K : \mu_1 = \phi(\arg\max_{a \neq 1} \mu_a), \max_a \mu_a > \phi(2) \right\},
$$

and the set of alternative models

$$
\mathrm{Alt}(\mu) = \left\{ \mu' \in \mathcal{S} : \arg\max_a \mu_a' \neq a^\star \right\},
$$

where $a^\star = \arg\max_a \mu_a$.

Then, for any $\delta$-correct algorithm, guaranteeing that at some stopping time $\tau$ the estimated optimal arm $\hat{a}_\tau$ is $\delta$-correct, i.e., $\mathbb{P}_\mu(\hat{a}_\tau \neq a^\star) \leq \delta$, we have the following result.

**Theorem B.26.** *For any $\delta$-correct algorithm, the sample complexity lower bound on the magic action problem is*

$$
\mathbb{E}_\mu[\tau] \geq T^\star(\mu)\mathrm{kl}(1 - \delta, \delta), \tag{26}
$$

*where $\mathrm{kl}(x, y) = x \log(x/y) + (1 - x) \log((1 - x)/(1 - y))$ and $T^\star(\mu)$ is the characteristic time of $\mu$, defined as*

$$
(T^\star(\mu))^{-1} = \max_{\omega \in \Delta(K)} \min_{a \neq 1, a^\star} \omega_1 \frac{(\phi(a^\star) - \phi(a))^2}{2\sigma_1^2} + \sum_{b \in \mathcal{K}_a(\omega)} \omega_b \frac{(\mu_b - m(\omega; \mathcal{K}_a(\omega)))^2}{2\sigma^2}, \tag{27}
$$

*where $m(\omega; \mathcal{C}) = \frac{\sum_{a \in \mathcal{C}} \omega_a \mu_a}{\sum_{a \in \mathcal{C}} \omega_a}$ and the set $\mathcal{K}_a(\omega)$ is defined as*

$$
\mathcal{K}_a(\omega) = \{a\} \cup \{b \in \{2, \ldots, K\} : \mu_b \geq m(\omega; \mathcal{C}_b \cup \{a\}) \text{ and } \mu_b \geq \phi(2)\} .
$$

*with $\mathcal{C}_x = \{b \in \{2, \ldots, K\} : \mu_b \geq \mu_x\}$ for $x \in [K]$.*

---

[5]One could also consider strictly increasing functions.

*Proof.* **Step 1: Log-likelihood ratio.** The initial part of the proof is rather standard, and follows the same argument used in the Best Arm Identification and Best Policy Identification literature (Garivier & Kaufmann, 2016).

Take the expected log-likelihood ratio between $\mu$ and $\mu' \in \mathrm{Alt}(\mu)$ of the data observed up to $\tau$ $\Lambda_\tau = \log \frac{\mathrm{d}\mathbb{P}_\mu(A_1, R_1, \ldots, A_\tau, R_\tau)}{\mathrm{d}\mathbb{P}_{\mu'}(A_1, R_1, \ldots, A_\tau, R_\tau)}$, where $A_t$ is the action taken in round $t$, and $R_t$ is the reward observed upon selecting $A_t$. Then, we can write

$$\Lambda_t = \sum_a \sum_{n=1}^{t} \mathbf{1}_{\{A_n = a\}} \log \frac{f_a(R_n)}{f_a'(R_n)}$$

where $f_a, f_a'$, are, respectively, the reward density for action $a$ in the two models $\mu, \mu'$ with respect to the Lebesgue measure. Letting $N_a(t)$ denote the number of times action $a$ has been selected up to round $t$, by an application of Wald's lemma the expected log-likelihood ratio can be shown to be

$$\mathbb{E}_\mu[\Lambda_\tau] = \sum_a \mathbb{E}_\mu[N_a(\tau)] \mathrm{KL}(\mu_a, \mu_a')$$

where $\mathrm{KL}(\mu_a, \mu_a')$ is the KL divergence between two Gaussian distributions $\mathcal{N}(\mu_a, \sigma)$ and $\mathcal{N}(\mu_a', \sigma)$ (note that we have $\sigma_1$ instead of $\sigma$ for $a = 1$).

We also know from the information processing inequality (Kaufmann et al., 2016) that $\mathbb{E}_\mu[\Lambda_\tau] \geq \sup_{\mathcal{E} \in \mathcal{M}_\tau} \mathrm{kl}(\mathbb{P}_\mu(\mathcal{E}), \mathbb{P}_{\mu'}(\mathcal{E}))$, where $\mathcal{M}_t = \sigma(A_1, R_1, \ldots, A_t, R_t)$. We use the fact that the algorithm is $\delta$-correct: by choosing $\mathcal{E} = \{\hat{a}_\tau = a^\star\}$ we obtain that $\mathbb{E}_\mu[\Lambda_\tau] \geq \mathrm{kl}(1 - \delta, \delta)$, since $\mathbb{P}_\mu(\mathcal{E}) \geq 1 - \delta$ and $\mathbb{P}_{\mu'}(\mathcal{E}) = 1 - \mathbb{P}_{\mu'}(\hat{a}_\tau \neq a^\star) \leq 1 - \mathbb{P}_{\mu'}(\hat{a}_\tau = \arg\max_a \mu_a') \leq \delta$ (we also used the monotonicity properties of the Bernoulli KL divergence). Hence

$$\sum_a \mathbb{E}_\mu[N_a(\tau)] \mathrm{KL}(\mu_a, \mu_a') \geq \mathrm{kl}(1 - \delta, \delta).$$

Letting $\omega_a = \mathbb{E}_\mu[N_a(\tau)]/\mathbb{E}_\mu[\tau]$, we have that

$$\mathbb{E}_\mu[\tau] \sum_a \omega_a \mathrm{KL}(\mu_a, \mu_a') \geq \mathrm{kl}(1 - \delta, \delta).$$

Lastly, optimizing over $\mu' \in \mathrm{Alt}(\mu)$ and $\omega \in \Delta(K)$ yields the bound:

$$\mathbb{E}_\mu[\tau] \geq T^\star(\mu) \mathrm{kl}(1 - \delta, \delta),$$

where $T^\star(\mu)$ is defined as

$$(T^\star(\mu))^{-1} = \sup_{\omega \in \Delta(K)} \inf_{\mu' \in \mathrm{Alt}(\mu)} \sum_a \omega_a \mathrm{KL}(\mu_a, \mu_a').$$

**Step 2: Optimization over the set of alternative models.** We now face the problem of optimizing over the set of alternative models. First, we observe that $\mathcal{S} = \cup_{a \neq a^\star} \{\mu : \mu_1 = \phi(a), \mu_a > \phi(2)\}$. Therefore, we can write

$$\mathrm{Alt}(\mu) = \cup_{a \notin \{1, a^\star\}} \{\mu' : \mu_1' = \phi(a), \mu_a' > \max(\phi(2), \mu_b') \ \forall b \neq a\}.$$

Hence, for a fixed $a \notin \{1, a^\star\}$, the inner infimum becomes

$$\begin{aligned}
\inf_{\mu' \in \mathbb{R}^K} \quad & \omega_1 \frac{(\phi(a^\star) - \phi(a))^2}{2\sigma_1^2} + \sum_{a \neq 1} \omega_a \frac{(\mu_a - \mu_a')^2}{2\sigma^2} \\
\text{s.t.} \quad & \mu_a' \geq \max(\phi(2), \mu_b') \quad \forall b, \\
& \mu_1' = \phi(a).
\end{aligned} \tag{28}$$

To solve it, we construct the following Lagrangian

$$\ell(\mu', \theta) = \omega_1 \frac{(\phi(a^\star) - \phi(a))^2}{2\sigma_1^2} + \sum_{b \neq 1} \omega_b \frac{(\mu_b - \mu_b')^2}{2\sigma^2} + \sum_b \theta_b (\max(\phi(2), \mu_b') - \mu_a'),$$

where $\theta \in \mathbb{R}_+^K$ is the multiplier vector. From the KKT conditions we already know that $\theta_1 = 0, \theta_a = 0$ and $\theta_b = 0$ if $\mu_b' \leq \phi(2)$, with $b \in \{2, \dots, K\}$. In particular, we also know that either we have $\mu_b' = \mu_a'$ or $\mu_b' = \mu_b$. Therefore, for $\mu_b \leq \phi(2)$ the solution is $\mu_b' = \mu_b$, while for $\mu_b > \phi(2)$ the solution depends also on $\omega$.

To fix the ideas, let $\mathcal{K}$ be the set of arms for which $\mu_b' = \mu_a'$ at the optimal solution. Such set must necessarily include arm $a$. Then, note that

$$\frac{\partial \ell}{\partial \mu_a'} = \omega_a \frac{\mu_a' - \mu_a}{\sigma^2} - \sum_{b \in [K]} \theta_b = 0.$$

and

$$\frac{\partial \ell}{\partial \mu_b'} = \omega_b \frac{\mu_b' - \mu_b}{\sigma^2} + \theta_b = 0 \quad \text{for } b \neq (1, a).$$

Then, using the observations derived above, we conclude that

$$\mu_a' = \frac{\sum_{b \in \mathcal{K}} \omega_b \mu_b}{\sum_{b \in \mathcal{K}} \omega_b},$$

with $\mu_b' = \mu_a'$ if $b \in \mathcal{K}$, and $\mu_b' = \mu_b$ otherwise. However, how do we compute such set $\mathcal{K}$?

First, $\mathcal{K}$ includes arm $a$. However, in general we have $\mathcal{K} \neq \{a\}$ : if that were not true we would have $\mu_a' = \mu_a$ and $\mu_b' = \mu_b$ for the other arms – but if any $\mu_b$ is greater than $\mu_a$, then $a$ is not optimal, which is a contradiction. Therefore, also arm $a^\star$ is included in $\mathcal{K}$, since any convex combination of $\{\mu_a\}$ is necessarily smaller than $\mu_{a^\star}$. We apply this argument repeatedly for every arm $b$ to obtain $\mathcal{K}$.

Hence, for some set $\mathcal{C} \subseteq [K]$ define the average reward

$$m(\omega; \mathcal{C}) = \frac{\sum_{a \in \mathcal{C}} \omega_a \mu_a}{\sum_{a \in \mathcal{C}} \omega_a},$$

and the set $\mathcal{C}_x = \{a\} \cup \{b \in \{2, \dots, K\} : \mu_b \geq \mu_x\}$ for $x \in [K]$. Then,

$$\mathcal{K} := \mathcal{K}(\omega) = \{a\} \cup \{b \in \{2, \dots, K\} : \mu_b \geq m(\omega; \mathcal{C}_b) \text{ and } \mu_b \geq \phi(2)\}.$$

In other words, $\mathcal{K}$ is the set of *confusing arms* for which the mean reward in the alternative model changes. An arm $b$ is *confusing* if the average reward $m$, taking into account $b$, is smaller than $\mu_b$. If this holds for $b$, then it must also hold all the arms $b'$ such that $\mu_{b'} \geq \mu_b$. $\qquad\square$

As a corollary, we have the following upper bound on $T^\star(\mu)$.

**Corollary B.27.** *We have that*

$$T^\star(\mu) \leq \min_{\omega \in \Delta(K)} \max_{a \neq 1, a^\star} \frac{2\sigma_1^2}{\omega_1(\phi(a^\star) - \phi(a))^2}.$$

*In particular, for $\phi(x) = 1/x$ and $a^\star < K$ we have*

$$T^\star(\mu) \leq 2\sigma^2(a^\star(a^\star + 1))^2,$$

*while for $a^\star = K$ we get $T^\star(\mu) \leq 2\sigma^2(a^\star(a^\star - 1))^2$.*

*Proof.* Let $f_1(a) = \frac{(\phi(a^\star) - \phi(a))^2}{2\sigma_1^2}$. For every weight vector $\omega \in \Delta(K)$ and every $a \neq 1, a^\star$, the quantity

$$g_a(\omega) = \omega_1 f_1(a) + \sum_{b \in \mathcal{K}_a} \omega_b \frac{(\mu_b - m(\omega; \mathcal{K}_a))^2}{2\sigma^2}$$

satisfies $g_a(\omega) \geq \omega_1 f_1(a)$ because the variance term is non–negative. Hence

$$(T^\star(\mu))^{-1} = \max_{\omega \in \Delta(K)} \min_{a \neq 1, a^\star} g_a(\omega) \geq \max_{\omega \in \Delta(K)} \omega_1 \min_{a \neq 1, a^\star} f_1(a).$$

Since $\omega_1 \leq 1$, the right–hand side is lower bounded by $\omega_1 = 1$, giving

$$(T^\star(\mu))^{-1} \geq \min_{a \neq 1, a^\star} f_1(a) = \frac{1}{2\sigma_1^2} \min_{a \neq 1, a^\star} (\phi(a^\star) - \phi(a))^2.$$

Taking reciprocals yields

$$T^\star(\mu) \leq \frac{2\sigma_1^2}{\min\limits_{a \neq 1,a^\star} (\phi(a^\star) - \phi(a))^2} = \min_{\omega \in \Delta(K)} \max_{a \neq 1,a^\star} \frac{2\sigma_1^2}{\omega_1 (\phi(a^\star) - \phi(a))^2},$$

because the minimisation over $\omega$ clearly selects $\omega_1 = 1$. (This justifies the form stated in the corollary.)

**Specialising to** $\phi(x) = 1/x$. With $\phi(x) = 1/x$ the difference $\phi(a^\star) - \phi(a) = \frac{1}{a^\star} - \frac{1}{a}$ is positive for all $a > a^\star$ and negative otherwise; its smallest non-zero magnitude is obtained for the *closest* index to $a^\star$:

- If $a^\star < K$, that index is $a^\star + 1$, giving

$$\min_{a \neq 1,a^\star} (\phi(a^\star) - \phi(a))^2 = \left(\frac{1}{a^\star} - \frac{1}{a^\star + 1}\right)^2 = \frac{1}{\left[a^\star(a^\star + 1)\right]^2}.$$

- If $a^\star = K$, the closest index is $K - 1$, leading to

$$\min_{a \neq 1,a^\star} (\phi(a^\star) - \phi(a))^2 = \left(\frac{1}{K-1} - \frac{1}{K}\right)^2 = \frac{1}{\left[a^\star(a^\star - 1)\right]^2}.$$

Plugging each expression in the general upper bound above concludes the proof. $\qquad\square$

Finally, to get a better intuition of the main result, we can look at the 3-arms case: it is optimal to only sample the magic arm iff $|\phi(a^\star) - \phi(a)| > \frac{\sigma_1(\mu_{a^\star} - \mu_a)}{2\sigma}$.

**Lemma B.28.** *With $K = 3$ we have that $\omega_1 = 1$ if and only if*

$$|\phi(a^\star) - \phi(a)| > \frac{\sigma_1(\mu_{a^\star} - \mu_a)}{2\sigma},$$

*and $\omega_1 = 0$ if the reverse inequality holds.*

*Proof.* With 3 arms, from the proof of the theorem we know that $\mathcal{K}_a(\omega) = \{a, a^\star\}$ for all $\omega$. Letting $m(\omega) = \frac{\omega_a \mu_a + \omega_{a^\star} \mu_{a^\star}}{\omega_a + \omega_{a^\star}}$, we obtain

$$(T^\star(\mu))^{-1} = \max_{\omega \in \Delta(3)} \omega_1 \frac{(\phi(a^\star) - \phi(a))^2}{2\sigma_1^2} + \frac{\omega_a(\mu_a - m(\omega))^2 + \omega_{a^\star}(\mu_{a^\star} - m(\omega))^2}{2\sigma^2}.$$

Clearly the solution is $\omega_1 = 1$ as long as

$$\frac{(\phi(a^\star) - \phi(a))^2}{2\sigma_1^2} > \max_{\omega:\omega_a + \omega_{a^\star} = 1} \frac{\omega_a(\mu_a - m(\omega))^2 + \omega_{a^\star}(\mu_{a^\star} - m(\omega))^2}{2\sigma^2}.$$

To see why this is the case, let $f_1 = \frac{(\phi(a^\star) - \phi(a))^2}{2\sigma_1^2}$, $f_2(\omega_a, \omega_{a^\star}) = \frac{\omega_a(\mu_a - m(\omega))^2}{2\sigma^2}$ and $f_3(\omega_a, \omega_{a^\star}) = \frac{\omega_{a^\star}(\mu_{a^\star} - m(\omega))^2}{2\sigma^2}$. Then, we can write

$$\omega_1 f_1 + \omega_a f_2(\omega_a, \omega_{a^\star}) + \omega_{a^\star} f_3(\omega_a, \omega_{a^\star}) = \omega_1 f_1 + (1 - \omega_1)\left[\frac{\omega_a f_2}{1 - \omega_1} + \frac{\omega_{a^\star} f_3}{1 - \omega_1}\right].$$

Being a convex combination, this last term can be upper bounded as

$$\omega_1 f_1 + \omega_a f_2(\omega_a, \omega_{a^\star}) + \omega_{a^\star} f_3(\omega_a, \omega_{a^\star}) \leq \max\left(f_1, \frac{\omega_a f_2}{1 - \omega_1} + \frac{\omega_{a^\star} f_3}{1 - \omega_1}\right).$$

Now, note that also the term inside the bracket is a convex combination. Therfore, let $\omega_a = (1 - \omega_1)\alpha$ and $\omega_{a^\star} = (1 - \omega_1)(1 - \alpha)$ for some $\alpha \in [0, 1]$. We have that

$$m(\omega) = \frac{(1 - \omega_1)\alpha\mu_a + (1 - \omega_1)(1 - \alpha)\mu_{a^\star}}{1 - \omega_1} = \alpha\mu_a + (1 - \alpha)\mu_{a^\star}.$$

Hence, we obtain that

$$
\begin{aligned}
\frac{\omega_a(\mu_a - m(\omega))^2 + \omega_{a^\star}(\mu_{a^\star} - m(\omega))^2}{2(1-\omega_1)\sigma^2} &= \frac{\omega_a f_2 + \omega_{a^\star} f_3}{1-\omega_1}, \\
&= \frac{\alpha(1-\alpha)^2(\mu_a - \mu_{a^\star})^2 + (1-\alpha)\alpha^2(\mu_{a^\star} - \mu_a)^2}{2\sigma^2}, \\
&= \alpha(1-\alpha)\frac{(1-\alpha)(\mu_a - \mu_{a^\star})^2 + \alpha(\mu_{a^\star} - \mu_a)^2}{2\sigma^2}, \\
&= \alpha(1-\alpha)\frac{(\mu_a - \mu_{a^\star})^2}{2\sigma^2}.
\end{aligned}
$$

Since this last term is maximized for $\alpha = 1/2$, we obtain

$$
\omega_1 f_1 + \omega_a f_2(\omega_a, \omega_{a^\star}) + \omega_{a^\star} f_3(\omega_a, \omega_{a^\star}) \le \max\left(f_1, \frac{(\mu_a - \mu_{a^\star})^2}{8\sigma^2}\right).
$$

Since $f_1$ is attained for $\omega_1 = 1$, we have that as long as $f_1 > \frac{(\mu_a - \mu_{a^\star})^2}{8\sigma^2}$, then the solution is $\omega_1 = 1$.

On the other hand, if $\frac{(\mu_a - \mu_{a^\star})^2}{8\sigma^2} > f_1$, then we can set $\omega_a = (1-\omega_1)/2$ and $\omega_{a^\star} = (1-\omega_1)/2$, leading to

$$
\omega_1 f_1 + \omega_a f_2(\omega_a, \omega_{a^\star}) + \omega_{a^\star} f_3(\omega_a, \omega_{a^\star}) = \omega_1 f_1 + (1-\omega_1)\frac{(\mu_a - \mu_{a^\star})^2}{8\sigma^2},
$$

which is maximized at $\omega_1 = 0$. $\qquad\square$

## B.6 Sample Complexity Bound for the Multiple Magic Actions MAB Problem

We now extend our analysis to the case where multiple magic actions can be present in the environment. In contrast to the single magic action setting, here a *chain* of magic actions sequentially reveals information about the location of the optimal action. Without loss of generality, assume that the first $n$ arms (with indices $1, \ldots, n$) are the magic actions, and the remaining $K - n$ arms are non–magic. The chain structure is such that pulling magic arm $j$ (with $1 \le j < n$) yields information about only the location of the next magic arm $j + 1$, while pulling the final magic action (arm $n$) reveals the identity of the optimal action. As before, we assume that the magic actions are informational only and are never optimal.

To formalize the model, let $\phi : \{1, \ldots, n\} \to \mathbb{R}$ be a strictly decreasing function. We assume that the magic actions have fixed means given by

$$
\mu_j = \begin{cases} \phi(j+1), & \text{if } j = 1, \ldots, n-1, \\ \phi\left(\arg\max_{a \notin \{1, \ldots, n\}} \mu_a\right), & \text{if } j = n. \end{cases}
$$

and that the non–magic arms satisfy

$$
\mu^\star = \max_{a \notin \{1, \ldots, n\}} \mu_a > \phi(n).
$$

Thus, the optimal arm lies among the non–magic actions. Considering the noiseless case where the rewards of all actions are fixed and the case where we can identify if an action is magic once revealed, we have the following result.

**Theorem B.29.** *Consider noiseless magic bandit problem with $K$ arms and $n$ magic actions. The optimal sample complexity is upper bounded as*

$$
\inf_{\text{Alg}} \mathbb{E}_{\text{Alg}}[\tau] \le \min\left(n, \sum_{j=1}^{K-n}\left(\prod_{i=j+1}^{K-n}\frac{i}{n-1+i}\right)\left(1 + \frac{n-1}{n-1+j}\min\left(\frac{n-2}{2}, \frac{j(n-1+j)}{j+1}\right)\right)\right).
$$

*Proof.* In the proof we derive a sample complexity bound for a policy based on some insights. We use the assumption that upon observing a reward from a magic arm, the learner can almost surely identify that the pulled arm is a magic arm.

Let us define the state $(m, r, l)$, where $m$ denotes the number of remaining unrevealed magic actions ($m_0 = n - 1$), $r$ denotes the number of remaining unrevealed non-magic actions ($r_0 = K - n$), and $l$ is the binary indicator with value 1 if we have revealed any hidden magic action and 0 otherwise.

Before any observation the learner has no information about which $n - 1$ indices among $\{2, \ldots, K\}$ form the chain of intermediate magic arms. Hence, one can argue that at the first timestep is optimal to sample uniformly at random an action in $\{2, \ldots, K\}$.

Upon observing a magic action, and thus we are in state $(m, r, 1)$, we consider the following candidate policies: (1) start from the revealed action and follow the chain, or (2) keep sampling unrevealed actions uniformly at random until all non-magic actions are revealed. As previously discussed, starting the chain from the initial magic action would be suboptimal and we do not consider it.

Upon drawing a hidden magic arm, let its chain index be $j \in \{2, \ldots, n\}$ (which is uniformly distributed). The remaining cost to complete the chain is $n - j$, and hence its expected value is

$$\mathbb{E}[n - j] = \frac{n - 2}{2}.$$

Therefore, the total expected cost for strategy (1) is

$$T_1 = \frac{n - 2}{2}.$$

We can additionally compute the expected cost for strategy (2) as follows: if the last non-magic action is revealed at step $i$, then among the first $i - 1$ draws there are exactly $r - 1$ non-magic arms. Since there are $\binom{m+r}{r}$ ways to place all $r$ non-magic arms $m + r$ slots, we have

$$
\begin{aligned}
T_2 &= \mathbb{E}[\text{Draws until all non-magic revealed}] \\
&= \sum_{i=r}^{m+r} i \cdot \mathbb{P}[\text{Last non-magic revealed at step } i] \\
&= \sum_{i=r}^{m+r} i \cdot \frac{\binom{i-1}{r-1}}{\binom{m+r}{r}} \\
&= \frac{r! \cdot m!}{(m+r)!} \sum_{i=r}^{m+r} i \binom{i-1}{r-1} \\
&= \frac{r! \cdot m!}{(m+r)!} \sum_{i=r}^{m+r} \frac{i!}{(r-1)!(i-r)!} \\
&= \frac{r! \cdot m!}{(m+r)!} \sum_{i=r}^{m+r} r \binom{i}{r} \\
&= \frac{r \cdot r! \cdot m!}{(m+r)!} \binom{m+r+1}{r+1} \\
&= \frac{r \cdot r! \cdot m!}{(m+r)!} \cdot \frac{(m+r+1) \cdot (m+r)!}{(r+1) \cdot r! \cdot m!} \\
&= \frac{r(m+r+1)}{r+1}
\end{aligned}
$$

Finally, we define a policy in $(m, r, 1)$ as the one choosing between strategy 1 and strategy 2, depending on which one achieves the minimum cost. Hence, the complexity of this policy is

$$V(m, r, 1) = \min \left( \frac{n - 2}{2}, \frac{r(m+r+1)}{r+1} \right).$$

Now, before finding a magic arm, consider a policy that uniformly samples between the non-revealed arms. Therefore, in $(m, r, 0)$ we can achieve a complexity of $1 + \frac{m}{m+r} V(m-1, r, 1) + \frac{r}{m+r} V(m, r-1, 0)$. Since we can always achieve a sample complexity of $n$, we can find a policy with the following

complexity:

$$V(m, r, 0) = \min\left(n, 1 + \frac{m}{m+r}V(m-1, r, 1) + \frac{r}{m+r}V(m, r-1, 0)\right)$$

$$= \min\left(n, 1 + \frac{m}{m+r}\min\left(\frac{n-2}{2}, \frac{r(m+r)}{r+1}\right) + \frac{r}{m+r}V(m, r-1, 0)\right)$$

Given we always start with $n-1$ hidden magic actions we can define a recursion in terms of just the variable $r$ as follows:

$$V(r) = 1 + \frac{n-1}{n-1+r}T(r) + \frac{r}{n-1+r}V(r-1),$$

where $T(r) = \min\left(\frac{n-2}{2}, \frac{r(n-1+r)}{r+1}\right)$. Letting $A(r) = \frac{r}{n-1+r}$ and $B(r) = 1 + \frac{n-1}{n-1+r}T(r)$, we can write

$$V(r) = B(r) + A(r)V(r-1),$$

Clearly $V(0) = 0$ since if all non-magic actions are revealed, then we know the optimal action deterministically. Unrolling the recursion we get

$$V(1) = B(1),$$
$$V(2) = B(2) + A(2)B(1),$$
$$V(3) = B(3) + A(3)B(2) + A(3)A(2)B(1),$$
$$...$$

$$V(r) = \sum_{j=1}^{r}\left(\prod_{i=j+1}^{r}A(i)\right)B(j).$$

Substituting back in our expression, we get

$$V(r) = \sum_{j=1}^{r}\left(\prod_{i=j+1}^{r}\frac{i}{n-1+i}\right)\left(1 + \frac{n-1}{n-1+j}T(j)\right).$$

Thus starting at $r = K - n$ we get the following expression:

$$\min\left(n, \sum_{j=1}^{K-n}\left(\prod_{i=j+1}^{K-n}\frac{i}{n-1+i}\right)\left(1 + \frac{n-1}{n-1+j}\min\left(\frac{n-2}{2}, \frac{j(n-1+j)}{j+1}\right)\right)\right),$$

which is also an upper bound on the optimal sample complexity.

□

To get a better intuition of the result, we also have the following corollary, which shows that we should expect a scaling linear in $n$ for small values of $n$ (for large values the complexity tends instead to "flatten").

**Corollary B.30.** *Let $T$ be the scaling in thm. B.29. We have that*

$$\min(n, (K-n)/2) \lesssim T \lesssim C\min(n, K/2).$$

*Proof.* First, observe the scaling

$$\left(1 + \frac{n-1}{n-1+j}\min\left(\frac{n-2}{2}, \frac{j(n-1+j)}{j+1}\right)\right) = O(n/2).$$

At this point, note that

$$\prod_{i=j+1}^{K-n} \frac{i}{n-1+i} = \prod_{i=j+1}^{K-n} \left(1 + \frac{n-1}{i}\right)^{-1}.$$

Using that $\frac{x}{1+x} \le \log(1+x) \le x$, we have

$$\log \prod_{i=j+1}^{K-n} \frac{i}{n-1+i} = \sum_{i=j+1}^{K-n} -\log\left(1 + \frac{n-1}{i}\right) \ge -(n-1) \sum_{i=j+1}^{K-n} \frac{1}{i}.$$

and

$$\log \prod_{i=j+1}^{K-n} \frac{i}{n-1+i} = \sum_{i=j+1}^{K-n} -\log\left(1 + \frac{n-1}{i}\right) \le -(n-1) \sum_{i=j+1}^{K-n} \frac{1}{n-1+i}.$$

Define $H_n = \sum_{i=1}^{n} 1/i$ to be the $n$-th Harmonic number, we also have

$$\sum_{i=j+1}^{K-n} \frac{1}{i} = H_{K-n} - H_j.$$

Therefore

$$-(n-1)(H_{K-n} - H_j) \le \log \prod_{i=j+1}^{K-n} \frac{i}{n-1+i} \le -(n-1)(H_{K-1} - H_{n+j-1})$$

Using that $H_\ell \sim \log(\ell) + \gamma + O(1/\ell)$, where $\gamma$ is the Euler–Mascheroni constant, we get

$$\left(\frac{j}{K-n}\right)^{n-1} \lesssim \prod_{i=j+1}^{K-n} \frac{i}{n-1+i} \lesssim \left(\frac{n+j-1}{K-1}\right)^{n-1}.$$

Therefore, we can bound $\sum_{j=1}^{K-n} \left(\frac{n+j-1}{K-1}\right)^{n-1}$ using an integral bound

$$\sum_{j=1}^{K-n} \left(\frac{n+j-1}{K-1}\right)^{n-1} \le \int_0^{K-n} \left(\frac{n+x}{K-1}\right)^{n-1} dx \le \frac{e(K-1)}{n}.$$

From which follows that the original expression can be upper bounded by an expression scaling as $O(\min(n, (K-1)/2))$.

Similarly, using that $\sum_{j=1}^{K-n} \left(\frac{j}{K-n}\right)^{n-1} \ge (K-n)/n$, we have that the lower bound scales as $\min(n, (K-n)/2)$. $\qquad\square$

## C   ALGORITHMS

In this section we present some of the algorithms more in detail. These includes: **ICPE** with fixed horizon, $I$-DPT and $I$-IDS.

Recall that in **ICPE** we treat trajectories of data $\mathcal{D}_t = (x_1, a_1, \ldots, x_t)$ as sequences to be given as input to sequential models, such as Transformers.

We define the input at timestep $t$ to be passed to a transformer as $s_t = (\mathcal{D}_t, \varnothing_{t:N})$, with $\varnothing_{t:N}$ indicating a null sequence of tokens for the remaining steps up to some pre-defined horizon $N$, with $s_1 = (x_1, \varnothing_{1:N})$.

To be more precise, letting $(x_t^\varnothing, a_t^\varnothing)$ denote, respectively, the null elements in the state and action at timestep $t$, we have $\varnothing_{t:t+k} = \{x_t^\varnothing, a_{t+1}^\varnothing, x_{t+1}^\varnothing, \cdots, a_{t+k-1}^\varnothing, x_{t+k}^\varnothing\}$.

The limit $N$ is a practical upper bound on the horizon that limits the dimensionality of the state, which is introduced for implementing the algorithm.

---

**Algorithm 3 ICPE** (In-Context Pure Explorer)

---

1: **Input:** Tasks distribution $\mathcal{P}$; confidence $\delta$; horizon $N$; initial $\lambda$ and hyper-parameter $T_\phi, T_\theta$.
      // Training phase
2: Initialize buffer $\mathcal{B}$, networks $Q_\theta, I_\phi$ and set $\bar\theta \leftarrow \theta, \bar\phi \leftarrow \phi$.
3: **while** Training is not over **do**
4:      Sample environment $M \sim \mathcal{P}$ with hypothesis $H^\star$, observe $x_1 \sim \rho$ and set $t \leftarrow 1$.
5:      **repeat**
6:          Execute action $a_t = \arg\max_a Q_\theta(\mathcal{D}_t, a)$ in $M$ and observe $x_{t+1}$.
7:          Add partial trajectory $(\mathcal{D}_t, \mathcal{D}_{t+1}, H^\star)$ to $\mathcal{B}$ and set $t \leftarrow t + 1$.
8:      **until** $a_{t-1} = a_{\text{stop}}$ or $t > N$.
9:      In the fixed confidence, update $\lambda$ according to eq. (11).
10:     Sample batch $B \sim \mathcal{B}$ and update $\theta, \phi$ using $\mathcal{L}_{\inf}(B; \phi)$ (eq. (7)) and $\mathcal{L}_{\text{policy}}(B; \theta)$ (eq. (8) or eq. (9)).
11:     Every $T_\phi$ steps set $\bar\phi \leftarrow \phi$ (similarly, every $T_\theta$ steps set $\bar\theta \leftarrow \theta$).
12: **end while**

---

      // Inference phase
13: Sample unknown environment $M \sim \mathcal{P}$.
14: Collect a trajectory $\mathcal{D}_N$ (or $\mathcal{D}_\tau$ in fixed confidence) according to a policy $\pi_t(\mathcal{D}_t) = \arg\max_a Q_\theta(\mathcal{D}_t, a)$, until $t = N$ (or $a_t = a_{\text{stop}}$).
15: **Return** $\hat{H}_N = \arg\max_H I_\phi(H|\mathcal{D}_N)$ (or $\hat{H}_\tau = \arg\max_H I_\phi(H|\mathcal{D}_\tau)$ in the fixed confidence)

---

### C.1   ICPE WITH FIXED CONFIDENCE

Optimizing the dual formulation

$$\min_{\lambda \geq 0} \max_{I, \pi} V_\lambda(\pi, I)$$

can be viewed as a multi-timescale stochastic optimization problem: the slowest timescale updates the variable $\lambda$, an intermediate timescale optimizes over $I$, and the fastest refines the policy $\pi$.

**MDP-like formulation.**   As shown in the theory, we can use the MDP formalism to define an RL problem: we define a reward $r$ that penalizes the agent at all timesteps, that is $r_t = -1$, while at the stopping-time we have $r_\tau = -1 + \lambda I(H^\star|\mathcal{D}_\tau)$. Hence, a trajectory's return can be written as

$$G_\tau = \sum_{t=1}^{\tau} r_t = -\tau + \lambda I(H^\star|\mathcal{D}_\tau).$$

Accordingly, one can define the $Q$-value of $(\pi, I, \lambda)$ in a pair $(\mathcal{D}_\tau, a)$.

**Optimization over $\phi$.**   We treat each optimization separately, employing a descent-ascent scheme. The distribution $I$ is modeled using a sequential architecture parameterized by $\phi$, denoted by $I_\phi$. Fixing $(\pi, \lambda)$, the inner maximization in eq. (4) corresponds to

$$\max_\phi \mathbb{E}^\pi[\mathbf{1}_{\{\hat{H}_\tau = H^\star\}}], \quad \text{with } \hat{H}_\tau = \arg\max_H I_\phi(H|\mathcal{D}_\tau).$$

We train $\phi$ via cross-entropy loss:

$$-\sum_{H'} \mathbf{1}_{\{\hat{H}_\tau = H^\star\}} \log I_\phi(H'|\mathcal{D}_\tau) = -\log I_\phi(H^\star|\mathcal{D}_\tau),$$

averaged across trajectories in a batch.

**Optimization over $\pi$.** The policy $\pi$ is defined as the greedy policy with respect to learned $Q$-values. Therefore, standard RL techniques can learn the $Q$-function that maximizes the value in eq. (4) given $(\lambda, I)$. Denoting this function by $Q_\theta$, it is parameterized using a sequential architecture with parameters $\theta$.

We train $Q_\theta$ using DQN (Mnih et al., 2015; Van Hasselt et al., 2016), employing a replay buffer $\mathcal{B}$ and a target network $Q_{\bar{\theta}}$ parameterized by $\bar{\theta}$. To maintain timescale separation, we introduce an additional inference target network $I_{\bar{\phi}}$, parameterized by $\bar{\phi}$, which provides stable training feedback for $\theta$. When $(I, \lambda)$ are fixed, optimizing $\pi$ reduces to maximizing:

$$-\tau + \lambda \log I_\phi(H^\star|\mathcal{D}_\tau).$$

Hence, we define the reward at the transition $z = (\mathcal{D}_n, a_N, \mathcal{D}_{n+1}, d, H^\star)$ (with the convention that $\mathcal{D}_{n+1} \leftarrow \mathcal{D}_n$ if $a = a_{\text{stop}}$) as:

$$r_\lambda(z) := -1 + d\lambda \log I_{\bar{\phi}}(H^\star|\mathcal{D}_{n+1}),$$

where $d = \mathbf{1}\{z \text{ is terminal}\}$ ($z$ is terminal if the transition corresponds to the last timestep in a horizon, or $a = a_{\text{stop}}$). Furthermore, for a transition $z$ we define $z_{\text{stop}} := z|_{(a, \mathcal{D}_{n+1}) \leftarrow (a_{\text{stop}}, \mathcal{D}_n)}$ as the same transition $z$ with $a \leftarrow a_{\text{stop}}$ and $\mathcal{D}'_{n+1} \leftarrow \mathcal{D}_n$.

There is one thing to note: the logarithm in the reward is justified since the original problem can be equivalently written as:

$$\min_{\lambda \geq 0} \max_{I, \pi} -\mathbb{E}^\pi[\tau] + \lambda \left[ \log \left( \mathbb{P}^\pi(\hat{H}_\tau = H^\star) \right) - \log(1 - \delta) \right],$$

after noting that we can apply the logarithm to the constraint in eq. (4), before considering the dual. Thus the optimal solutions $(I, \pi)$ remain the same.

Then, using classical TD-learning (Sutton & Barto, 2018), the training target for a transition $z$ is defined as:

$$y_\lambda(z) = r_\lambda(z) + (1 - d)\gamma \max_{a'} Q_{\bar{\theta}}(\mathcal{D}_{n+1}, a'),$$

where $\gamma \in (0, 1]$ is the discount factor.

As discussed earlier, we have a dedicated stopping action $a_{\text{stop}}$, whose value depends solely on history. Thus, its $Q$-value is updated retrospectively at any state $s$ using an additional loss:

$$\left( r_\lambda(z_{\text{stop}}) - Q_\theta(s, a_{\text{stop}}) \right)^2.$$

Therefore, the overall loss that we consider for $\theta$ for a single transition $z$ can be written as

$$\mathbf{1}_{\{a \neq a_{\text{stop}}\}} \left( y_\lambda(z) - Q_\theta(s, a) \right)^2 + \left( r_\lambda(z_{\text{stop}}) - Q_\theta(s, a_{\text{stop}}) \right)^2,$$

where $\mathbf{1}_{\{a \neq a_{\text{stop}}\}}$ avoids double accounting for the stopping action.

To update parameters $(\theta, \phi)$, we sample a batch $B \sim \mathcal{B}$ from the replay buffer and apply gradient updates as specified in the main text. Lastly, target networks are periodically updated.

**Optimization over $\lambda$.** We update $\lambda$ by assessing the confidence of $I_\phi$ at the stopping time according to eq. (11), maintaining a slow ascent-descent optimization schedule for sufficiently small learning rates.

**Cost implementation.** Lastly, in practice, we optimize a reward $r_\lambda(z) = -c + dI_{\bar\phi}(H^\star|\mathcal{D}_{n+1})$, by setting $c = 1/\lambda$, and noting that for a fixed $\lambda$ the RL optimization remains the same. The reason why we do so is due to the fact that with this expression we do not have the product $\lambda I(H^\star|\mathcal{D}_{n+1})$, which makes the descent-ascent process more difficult.

We also use the following cost update

$$c_{t+1} = c_t - \beta(1 - \delta - \mathbf{1}_{\{H^\star = \arg\max_H I(H|\mathcal{D}_\tau)\}}).$$

To see why the cost can be updated in this way, define the parametrization $\lambda = e^{-x}$. Then the optimization problem becomes

$$\min_x \max_I \min_\pi -\mathbb{E}\pi[\tau] + e^{-x}\left[\mathbb{P}^\pi\left(H^\star = \hat{H}_\tau\right) - 1 + \delta\right],$$

Letting $\rho = \mathbb{P}^\pi\left(H^\star = \hat{H}_\tau\right) - 1 + \delta$, the gradient update for $x$ with a learning rate $\beta$ simply is

$$x_{t+1} = x_t - \beta e^{-x_t}\rho,$$

implying that

$$-\log(\lambda_{t+1}) = -\log(\lambda_t) - \beta\lambda_t\rho.$$

Defining $c_t = 1/\lambda_t$, we have that

$$\log(c_{t+1}) = \log(c_t) - (\beta\rho/c_t) \Rightarrow c_{t+1} = c_t e^{\beta\rho/c_t}.$$

Using then the approximation $e^x \approx 1 + x$, we find $c_{t+1} = c_t + \beta\rho = c_t - \beta(1 - \delta - \mathbf{1}_{\{H^\star = \arg\max_H I(H|\mathcal{D}_\tau)\}})$.

**Training vs Deployment.** Thus far, our discussion of `ICPE` has focused on the training phase. After training completes, the learned policy $\pi$ and inference network $I$ can be deployed directly: during deployment, $\pi$ both collects data and determines when to stop—either by triggering its stopping action or upon reaching the horizon $N$.

## C.2 ICPE WITH FIXED BUDGET

In the fixed budget setting (problem in eq. (2)) the MDP terminates at timestep $N$, and we set the reward to be $r_t = 0$ for $t < N$ and $r_N = I(H^\star|\mathcal{D}_N)$, with $\hat{H}_N = \arg\max_H I(H|\mathcal{D}_N)$ being the inferred hypothesis. Accordingly, one can define the value of $(\pi, I)$ using $Q$ functions as beofre.

**Practical implementation.** The practical implementation for the fixed horizon follows closely that of the fixed confidence setting, and we refer the reader to that section for most of the details. In this case the reward in a transition $z = (\mathcal{D}_n, a, \mathcal{D}'_{n+1}, d, H^\star)$ is defined as as:

$$r_\lambda(z) \coloneqq d\log I_{\bar\phi}(H^\star|\mathcal{D}_{n+1}), \tag{29}$$

where $d = \mathbf{1}\{z \text{ is terminal}\}$. Note that we can use the logarithm, since solving the original problem is also equivalent to solving But note that the original problem is also equivalent to solving

$$\max_I \max_\pi \log\left(\mathbb{P}^\pi\left(H^\star = \hat{H}_N\right)\right), \tag{30}$$

due to monotonicity of the logarithm.

The $Q$-values can be learned using classical TD-learning techniques (Sutton & Barto, 2018): to that aim, for a transition $z$, we define the target:

$$y_\lambda(z) = r_\lambda(z) + (1 - d)\max_{a'} Q_{\bar\theta}(\mathcal{D}_{n+1}, a'). \tag{31}$$

Then, the gradient updates are the same as for the fixed confidence setting.

## C.3 OTHER ALGORITHMS

In this section we describe Track and Stop (TaS) (Garivier & Kaufmann, 2016), and some variants such as $I$-IDS, $I$-DPT and the explore then commit variant of `ICPE`.

### C.3.1 TRACK AND STOP

Track and Stop (TaS, (Garivier & Kaufmann, 2016)) is an asymptotically optimal as $\delta \to 0$ for MAB problems. For simplicity, we consider a Gaussian MAB problem with $K$ actions, where the reward of each action is normally distributed according to $\mathcal{N}(\mu_a, \sigma^2)$, and let $\mu = (\mu_a)_{a \in [K]}$ denote the model. The TaS algorithm consists of: (1) the model estimation procedure and recommender rule; (2) the sampling rule, dictating which action to select at each timestep; (3) the stopping rule, defining when enough evidence has been collected to identify the best action with sufficient confidence, and therefore to stop the algorithm.

**Estimation Procedure and Recommender Rule**  The algorithm maintains a maximum likelihood estimate $\hat{\mu}_a(t)$ of the average reward for each arm based on the observations up to time $t$. Then, the recommender rule is defined as $\hat{a}_t = \arg\max_a \hat{\mu}_a(t)$.

**Sampling Rule.**  The sampling rule is based on the observation that any $\delta$-correct algorithm, that is an algorithm satisfying $\mathbb{P}(\hat{a}_\tau = a^\star) \geq 1 - \delta$, with $a^\star = \arg\max_a \mu_a$, satisfies the following sample complexity

$$\mathbb{E}[\tau] \geq T^\star(\mu)\mathrm{kl}(1 - \delta, \delta),$$

where $\mathrm{kl}(x, y) = x \log(x/y) + (1 - x) \log((1 - x)/(1 - y))$ and

$$(T^\star(\mu))^{-1} = \sup_{\omega \in \Delta(K)} \min_{a \neq a^\star} \frac{\omega_{a^\star} \omega_a}{\omega_a + \omega_{a^\star}} \frac{\Delta_a^2}{2\sigma^2},$$

with $\Delta_a = \mu_{a^\star} - \max_{a \neq a^\star} \mu_a$. Interestingly, to design an algorithm with minimal sample complexity, we can look at the solution $\omega^\star = \arg\inf_{\omega \in \Delta(K)} T(\omega; \mu)$, with $(T(\omega))^{-1} = \min_{a \neq a^\star} \frac{\omega_{a^\star} \omega_a}{\omega_a + \omega_{a^\star}} \frac{\Delta_a^2}{2\sigma^2}$.

The solution $\omega^\star$ provides the best proportion of draws, that is, an algorithm selecting an action $a$ with probability $\omega_a^\star$ matches the lower bound and is therefore optimal with respect to $T^\star$. Therefore, an idea is to ensure that $N_t/t$ tracks $\omega^\star$, where $N_t$ is the visitation vector $N(t) := [N_1(t) \quad \dots \quad N_K(t)]^\top$.

However, the average rewards $(\mu_a)_a$ are initially unknown. A commonly employed idea (Garivier & Kaufmann, 2016; Kaufmann et al., 2016) is to track an estimated optimal allocation $\omega^\star(t) = \arg\inf_{\omega \in \Delta(K)} T(\omega; \hat{\mu}(t))$ using the current estimate of the model $\hat{\mu}(t)$.

However, we still need to ensure that $\hat{\mu}(t) \to \mu$. To that aim, we employ a D-tracking rule (Garivier & Kaufmann, 2016), whcih guarantees that arms are sampled at a rate of $\sqrt{t}$. If there is an action $a$ with $N_a(t) \leq \sqrt{t} - K/2$ then we choose $a_t = a$. Otherwise, choose the action $a_t = \arg\min_a N_a(t) - t\omega_a^\star(t)$.

**Stopping rule.**  The stopping rule determines when enough evidence has been collected to determine the optimal action with a prescribed confidence level. The problem of determining when to stop can be framed as a statistical hypothesis testing problem (Chernoff, 1959), where we are testing between $K$ different hypotheses $(\mathcal{H}_a : (\mu_a > \max_{b \neq b} \mu_a))_a$.

We consider the following statistic $L(t) = tT(N(t)/t; \hat{\mu}(t))^{-1}$, which is a Generalized Likelihood Ratio Test (GLRT), similarly as in (Garivier & Kaufmann, 2016). Comparing with the lower bound, one needs to stop as soon as $L(t) \geq \mathrm{kl}(\delta, 1 - \delta) \sim \ln(1/\delta)$. However, to account for the random fluctuations, a more natural threshold is $\beta(t, \delta) = \ln((1 + \ln(t))/\delta)$, thus we use $L(t) \geq \beta(t, \delta)$ for stochastic MAB problems. We also refer the reader to (Kaufmann & Koolen, 2021) for more details.

**Why computing the sampling rule can be difficult.**  To derive the sampling rule, one usually first derives the characteristic time $T^\star(\mu)$. Above, we discussed the case where the underlying distributions are Gaussians, but in the more general case $T^\star$ can be written as

$$T^\star(\mu)^{-1} = \sup_{w \in \Delta_K} \inf_{\lambda \in \mathrm{Alt}(\mu)} \sum_{a=1}^{K} \omega_a \mathrm{KL}(P_{\mu_a}, P_{\lambda_a}),$$

where $\mathrm{Alt}(\mu)$ is the set of alternatives under which the identity of the best arm changes, $P_\mu$ is the distribution of rewards under $\mu$ (sim. for $\lambda$). Even though the objective is linear in $\omega$ for any fixed

$\lambda$, the inner feasible set $\text{Alt}(\mu)$ can be a nonconvex set ("make some competitor optimal"), and the map $\lambda \mapsto \text{KL}(P_{\mu_a}, P_{\lambda_a})$ is typically nonlinear and model–dependent. Even if these distributions are known, no closed form is available in general.

When arms belong to a one–parameter exponential family, and the problem has no structure, the optimal $\omega$ can be simply computed by applying (for example) the bisection method to a function whose evaluations requires the resolution of $K$ smooth scalar equations, thus linearly scaling in the number of arms. Since this optimization problem is usually solved at each timestep (or every $T$ timesteps), the complexity scales in the horizon $N$ and the number of arms $K$ as $NK$.

For general distributions, the situation worsens, and may be intractable without additional modeling assumptions. Lastly, note that the supremum over $\omega$ is a convex program in principle. First–order methods such as Frank–Wolfe can be applied to find an approximate solution. However, any tractable implementation presumes structural knowledge (e.g., an exponential–family model, smoothness) to guarantee a number of necessary properties.

### C.3.2 $I$-IDS

---

**Algorithm 4** $I$-IDS

---

1: **Input:** Pre-trained inference network $I_\phi$; prior means and variances $\mu_a, \sigma_a^2$ for all $a \in \mathcal{A}$; target error threshold $\delta$
2: **Initialize:** $f_a(x) = \mathcal{N}(x \mid \mu_a, \sigma_a^2)$ for each $a$
3: **for** $t = 1, 2, \ldots$ **do**
4:     **if** $\max_a I_\phi(a \mid \mathcal{D}_{t-1}) \geq 1 - \delta$ **then**
5:         **return** $\arg\max_a I_\phi(a \mid \mathcal{D}_{t-1})$
6:     **end if**
7:     **for** each arm $a \in \mathcal{A}$ **do**
8:         Approximate information gain:

$$g_t(a) = H\left(I_\phi(\cdot \mid \mathcal{D}_{t-1})\right) - \mathbb{E}_{r \sim p(r \mid a, \mathcal{D}_{t-1})}\left[H\left(I_\phi(\cdot \mid \mathcal{D}_{t-1}, a, r)\right)\right]$$

9:     **end for**
10:    Select action $a_t = \arg\max_a g_t(a)$
11:    Observe reward $r_t$
12:    Update dataset $\mathcal{D}_t = \mathcal{D}_{t-1} \cup \{(a_t, r_t)\}$
13: **end for**

---

We implement a variant of Information Directed Sampling (IDS) (Russo & Van Roy, 2018), where we use the inference network $I_\phi$ learned during ICPE training as a posterior over optimal arms. This approach enables IDS to exploit latent structure in the environment without explicitly modeling it via a probabilistic model; instead, the learned $I$-network implicitly captures such structure.

By using the same inference network in both ICPE and $I$-IDS, we directly compare the exploration policy learned by ICPE to the IDS heuristic applied on top of the same posterior distribution. While computing the expected information gain in IDS requires intractable integrals, we approximate them using a Monte Carlo grid of 30 candidate reward values per action. The full pseudocode for $I$-IDS is given in Algorithm 4.

### C.3.3 IN-CONTEXT EXPLORE-THEN-COMMIT

We implement an ICPE variant for regret minimization via an *explore-then-commit* framework. This method reuses the exploration policy and inference network learned during fixed-confidence training. The agent interacts with the environment using the learned exploration policy until it selects the stopping action. At that point, it commits to the arm predicted to be optimal by the $I$-network and repeatedly pulls that arm for the remainder of the episode. The full pseudo-code is provided in Algorithm 5.

---

**Algorithm 5** In-Context Explore-then-Commit

---

1: **Input:** Environment $M \sim \mathcal{P}(\mathcal{M})$; pre-trained critic network $Q_\theta$; pre-trained inference network $I_\phi$
2: Initialize *stopped* $\leftarrow$ False
3: Observe initial state $s_1 \sim \rho$
4: **for** $t = 1$ to $N$ **do**
5:     **if** *stopped* = False **and** $a_{\text{stop}} \neq \arg\max_a Q_\theta(s_t, a)$ **then**
6:         Execute $a_t = \arg\max_a Q_\theta(s_t, a)$ and observe $s_{t+1}$
7:     **else if** *stopped* = False **and** $a_{\text{stop}} = \arg\max_a Q_\theta(s_t, a)$ **then**
8:         Set *stopped* $\leftarrow$ True
9:         Execute $a_t = \arg\max_a I_\phi(s_t)$ and observe $s_{t+1}$
10:     **else**
11:         Execute $a_t = \arg\max_a I_\phi(s_t)$ and observe $s_{t+1}$
12:     **end if**
13: **end for**

---

### C.3.4    $I$-DPT

We implement a variant of DPT (Lee et al., 2023) using the inference network. The idea is to act greedily with respect to the posterior distribution $I$ at inference time.

First, we train $I$ using `ICPE`. Then, at deployment we act with respect to $I$: in round $t$ we selection action $a_t = \arg\max_H I(H|D_t)$. Upon observing $x_{t+1}$, we update $D_{t+1}$ and stop as soon as $\arg\max_H I(H|D_t) \geq 1 - \delta$.

### C.4    TRANSFORMER ARCHITECTURE

Here we briefly describe the architecture of the inference network $I$ and of the network $Q$.

Both networks are implemented using a Transformer architecture. For the inference network, it is designed to predict a hypothesis $H$ given a sequence of observations. Let the input tensor be denoted by $X \in \mathbb{R}^{B \times H \times m}$, where:

- $B$ is the batch size,
- $H$ is the sequence length (horizon), and
- $m = (d + |\mathcal{A}|)$, where $d$ is the dimensionality of each observation $x_t$.

The inference network operates as follows:

1. **Embedding Layer**: Each observation vector $m_t = (x_t, a_t)$ is first embedded into a higher-dimensional space of size $d_e$ using a linear transformation followed by a GELU activation: $h_t = \text{GELU}(W_{\text{embed}} m_t + b_{\text{embed}})$,    $h_t \in \mathbb{R}^{d_e}$.

2. **Transformer Layers**: The embedded sequence $h \in \mathbb{R}^{B \times H \times d_e}$ is then passed through multiple Transformer layers (specifically, a GPT-2 model configuration). The Transformer computes self-attention over the embedded input to model dependencies among observations:

$$h' = \text{Transformer}(h), \quad h' \in \mathbb{R}^{B \times H \times d_e}.$$

3. **Output Layer**: The final hidden state corresponding to the last element of the sequence $(h'_{:,-1,:})$ is fed into a linear output layer that projects it to logits representing the hypotheses:

$$o = W_{\text{out}} h'_{:,-1,:} + b_{\text{out}}, \quad o \in \mathbb{R}^{B \times |\mathcal{H}|}.$$

4. **Probability Estimation**: The output logits are transformed into log-probabilities via a log-softmax operation along the last dimension

$$\log p(H|X) = \text{log\_softmax}(o).$$

For $Q$, we use the same architecture, but do not take a log-softmax at the final step.

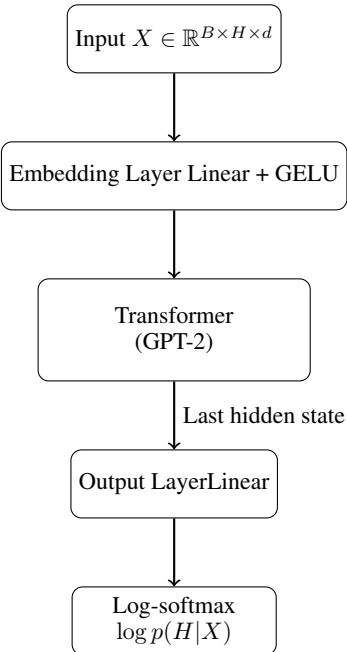

Figure 6: Model architecture for the inference network $I$ (similarly for $Q$).

## D  EXPERIMENTS

This section provides additional experimental results, along with detailed training and evaluation protocols to ensure clarity and reproducibility. All experiments were conducted using four NVIDIA A100 GPUs.

For more informations about the hyper-parameters, we also refer the reader to the `README.md` file in the code, as well as the training configurations in the `configs/experiments` folder.

**Libraries used in the experiments.**   We set up our experiments using Python 3.10.12 (Van Rossum & Drake Jr, 1995) (For more information, please refer to the following link `http://www.python.org`), and made use of the following libraries: NumPy (Harris et al., 2020), SciPy (Virtanen et al., 2020), PyTorch (Paszke et al., 2019), Seaborn (Waskom et al., 2017), Pandas (McKinney et al., 2010), Matplotlib (Hunter, 2007), CVXPY (Diamond & Boyd, 2016), Wandb (Biewald, 2020), Gurobi (Gurobi Optimization, LLC, 2024). Changes, and new code, are published under the MIT license. To run the code, please, read the attached README file for instructions.

**Hierarchical bootstrapping.**   For each experiment, we compute confidence intervals using hierarchical bootstrapping. Our data is organized at three levels: seed, environment, and trajectory. For each training seed we evaluate multiple environments, and for each environment we roll out multiple trajectories. Hierarchical bootstrapping allows us to correctly account for this nested structure when estimating uncertainty. This approach captures variability across seeds, environments, and trajectories, yielding a more reliable characterization of performance compared to classical bootstrapping.

To fix the ideas, consider the following random-effects model

$$Y_{a,b,c} = \mu + \alpha_a + \beta_{a,b,} + \gamma_{a,b,c}, \quad \alpha_a \sim \mathcal{N}(0, \sigma_{\text{seed}}^2), \ \beta_{a,b} \sim \mathcal{N}(0, \sigma_{\text{env}}^2), \ \gamma_{a,b,c} \sim \mathcal{N}(0, \sigma_{\text{traj}}^2),$$

and let $\bar{Y} = \frac{1}{mKN} \sum_{a=1}^{m} \sum_{b=1}^{k} \sum_{c=1}^{N} (Y_{a,b,c})$ with $m$ seeds, $K$ environments per seed and $N$ trajectories per environment.

The variance of $\bar{Y}$ then is $\mathrm{Var}(\bar{Y}) = \frac{\sigma_{\mathrm{seed}}^2}{m} + \frac{\sigma_{\mathrm{env}}^2}{mK} + \frac{\sigma_{\mathrm{traj}}^2}{mKN}$, which hierarchical bootstrapping accounts for. Instead, a naive bootstrap over the $mKN$ trajectories targets $\mathrm{Var}_{\mathrm{naive}}(\bar{Y}) \approx \frac{\mathrm{Var}(Y_{a,b,c})}{mKN} = \frac{\sigma_{\mathrm{seed}}^2 + \sigma_{\mathrm{env}}^2 + \sigma_{\mathrm{traj}}^2}{mKN}$, which effectively reduces the contribution at the seed-level by a factor $KN$.

## D.1 BANDIT PROBLEMS

Here, we provide the implementation and evaluation details for the bandit experiments reported in Section 4.1, covering deterministic, stochastic, and structured settings. Note that for this setting the observations are simply the observed rewards, i.e., $x_t = r_t$.

**Model Architecture and Optimization.** For all bandit tasks, `ICPE` uses a Transformer with 3 layers, 2 attention heads, hidden dimension 256, GELU activations, and dropout of 0.1 applied to attention, embeddings, and residuals (see also app. C.4 for a description of the architecture). Layer normalization uses $\epsilon = 10^{-5}$. Inputs are one-hot action-reward pairs with positional encodings. Models are trained using Adam with learning rates between $1 \times 10^{-4}$ and $1 \times 10^{-6}$, and batch sizes from 128 to 1024 depending on task complexity.

### D.1.1 DETERMINISTIC BANDITS WITH FIXED BUDGET

Each environment consists of $K$ arms, where $K \in \{4, 6, 8, \ldots, 20\}$. Mean rewards for each arm are sampled uniformly from $[0, 1]$, and rewards are deterministic (i.e., zero variance). Agents interact with the environment for exactly $K$ steps and are then required to predict the optimal arm. Success is measured by the probability of correctly identifying the best arm. We also compute the average number of unique arms selected during training episodes as a proxy for exploration diversity.

`ICPE` is compared against three baselines in the deterministic setting: *Uniform Sampling*, which selects arms uniformly at random; *DQN*, a deep Q-network trained directly on environmental rewards (Mnih et al., 2013); and *I-DPT*, which performs posterior sampling using `ICPE`'s $I$-network (Lee et al., 2023). All methods were evaluated over five seeds, with 900 environments per seed. 95% confidence intervals were computed with hierarchical bootstrapping.

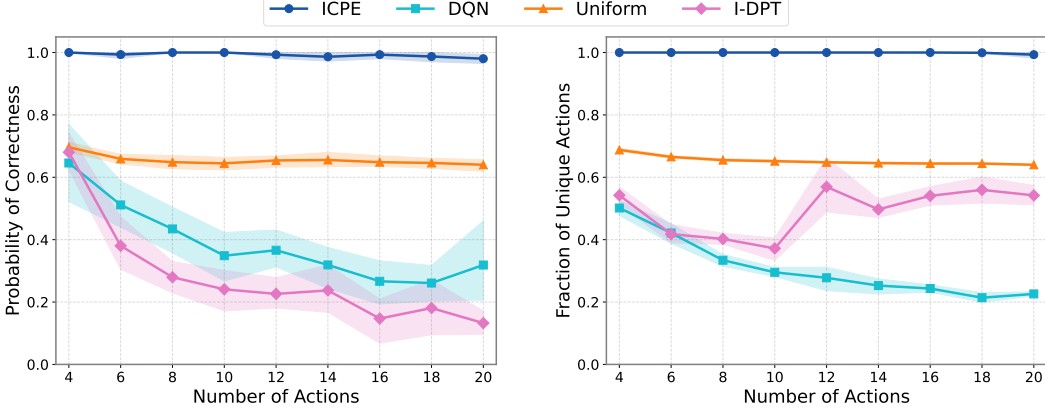

Figure 7: Deterministic bandits: (left) probability of correctly identifying the best action vs. $K$; (right) average fraction of unique actions selected during exploration vs. $K$.

### D.1.2 STOCHASTIC BANDITS PROBLEMS

In the stochastic Gaussian bandit setting, we evaluate `ICPE` on best-arm identification tasks with $K \in \{4, 6, 8, \ldots, 14\}$. Arm means are sampled uniformly from $[0, 0.4K]$, with a guaranteed minimum gap of $1/K$ between the top two arms. All arms have a fixed reward standard deviation of 0.5. The target confidence level is set to $\delta = 0.1$.

We compare `ICPE` against several widely used baselines: *Top-Two Probability Sampling (TTPS)* (Jourdan et al., 2022), *Track-and-Stop (TaS)* (Garivier & Kaufmann, 2016), *Uniform Sampling*, and *I-DPT*.

For *I-DPT*, stopping occurs when the predicted optimal arm has estimated confidence at least $1 - \delta$. For *TTPS* and *TaS*, we apply the classical stopping rule based on the characteristic time $T^{\star}(N_t/t; \hat{\mu}_t)$ (explained in app. C.3.1):

$$t \cdot T^{\star}(N_t/t; \hat{\mu}_t) \geq \log\left(\frac{1 + \log t}{\delta}\right).$$

Each method is evaluated over three seeds, with 300 environments, and 15 trajectories per environment. 95% confidence intervals were computed with hierarchical bootstrapping.

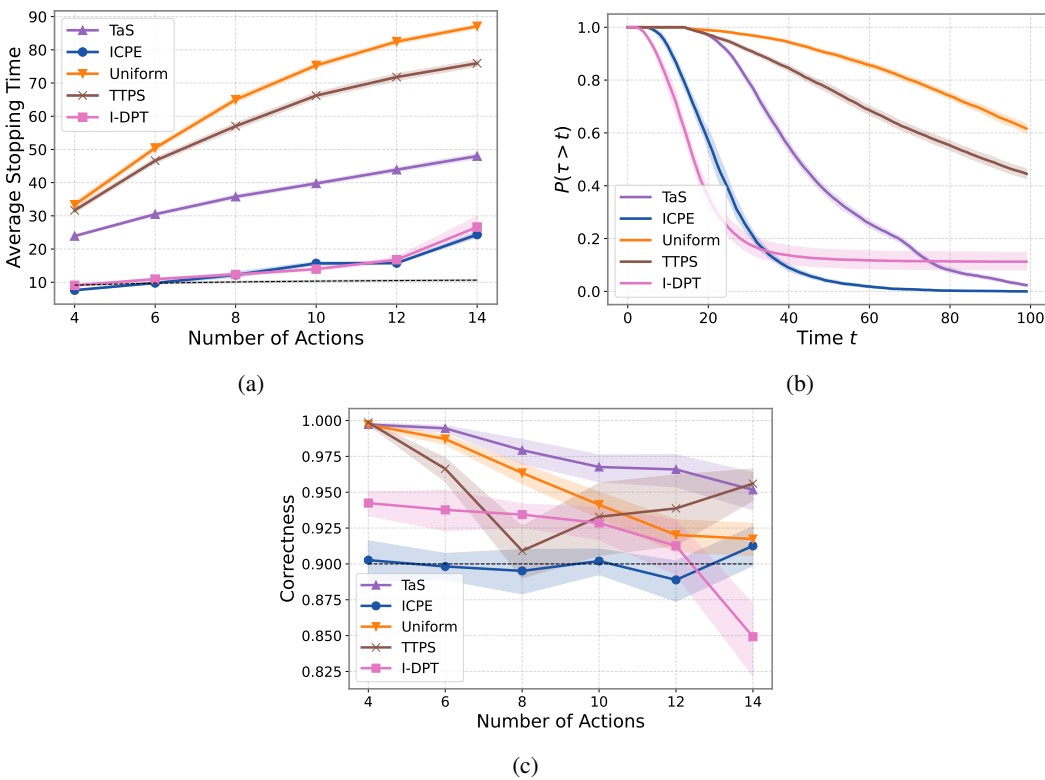

Figure 8: Results for stochastic MABs with fixed confidence $\delta = 0.1$ and $N = 100$: (a) average stopping time $\tau$; (b) survival function of $\tau$; (c) probability of correctness $\mathbb{P}^{\pi}(H^{\star} = \hat{H}_{\tau})$.

**Does `ICPE` learn randomized policies?** An intriguing question is whether `ICPE` is capable of learning randomized policies. Intuitively, one might expect randomized methods, such as actor-critic algorithms, to perform better. However, we observe that this is not the case for `ICPE`. Crucially, the inherent randomness of the environment, when passed as input to the transformer architecture, already serves as a source of stochasticity. Thus, although `ICPE` employs a deterministic mapping (via DQN) from observed trajectories, these trajectories themselves constitute random variables, rendering the policy's output effectively stochastic. To illustrate this, we examine an `ICPE` policy trained with fixed confidence ($\delta = 0.1$) in a setting with $K = 14$ actions (see the two rightmost plots in fig. 9). By analyzing 100 trajectories from this environment and computing an averaged policy, we clearly observe how trajectory randomness influences the policy's outputs. Specifically, exploration intensity peaks around the middle of the horizon and diminishes as the confidence level increases.

**Does `ICPE` resembles Track and Stop?** In fig. 9 (left figure) we compare an `ICPE` policy trained in the fixed confidence setting ($\delta = 0.1$) with an almost optimal version of TaS, that can be easily computed without solving any optimization problem. Let $\hat{\Delta}_t(a) = \hat{\mu}_{\hat{a}_t}(t) - \max_{a \neq \hat{a}_t} \hat{\mu}_a$, where $\hat{\mu}_a(t)$ is the empirical reward of arm $a$ in round $t$ and $\hat{a}_t = \arg\max_a \hat{\mu}_a(t)$ is the estimated optimal arm. Then, the approximate TaS policy is defined as

$$\pi_t(a) = \frac{1/\hat{\Delta}_a(t)}{\sum_b 1/\hat{\Delta}_b(t)},$$

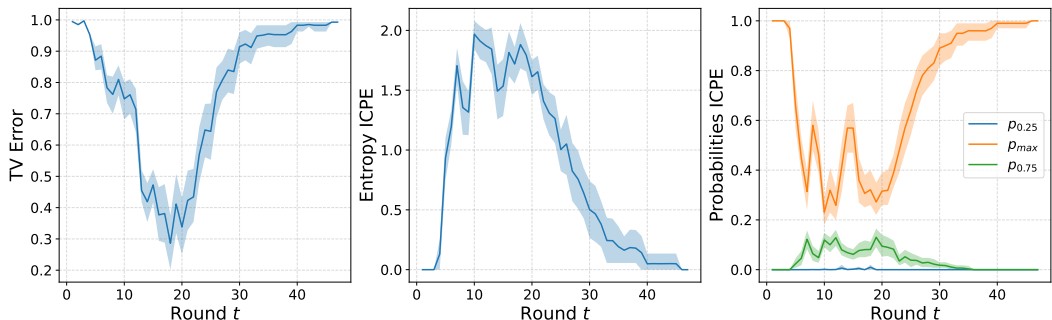

Figure 9: Statistics of `ICPE` with fixed confidence on 100 trajectories from a single environment, with $K = 14$. From left to right: Total variation error between the average `ICPE` policy and the approximate Track and Stop policy; entropy of the average policy of `ICPE`; probabilities of the average `ICPE` policy, with $p_{max}$ representing the maximum probability and $p_\alpha$ the $\alpha$-quantile.

with $\hat{\Delta}_{\hat{a}_t}(t) = \min_{a \neq \hat{a}_t} \hat{\Delta}_a(t)$. In the figure we sampled 100 trajectories from a single environment with $K = 14$, and computed an average `ICPE` policy. Then, we compared this policy to the approximate TaS policy, and computed the total variation. We can see that the two policies are not always similar. We believe this is due to the fact that `ICPE` is exploiting prior information on the environment, including the minimum gap assumption, and the fact that the average rewards are bounded in $[0, 0.4K]$.

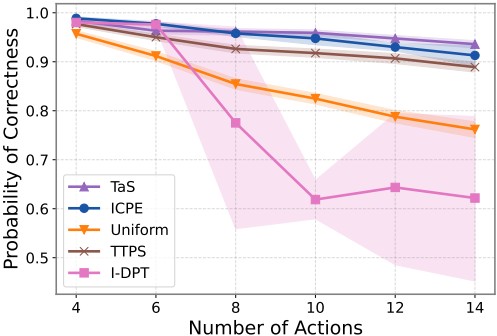

Figure 10: Correctness $\mathbb{P}^\pi(H^\star = \hat{H}_N)$ for stochastic MABs with fixed budget $N = 30$.

**Is `ICPE` robust to distribution shift?** As an in-context learning method, ICPE is designed to be meta-trained on the same family of tasks on which it will be deployed. That said, understanding robustness to changes in the environment distribution is important for assessing practicality. Therefore, we trained ICPE in the stochastic fixed-confidence bandit setting described above, where environments are sampled from a uniform distribution over Gaussian bandits with a minimum gap. At test time, we then evaluated the same trained model on bandit instances drawn from *shifted* environment distributions. We constructed these shifts by sampling reward means from a symmetric Dirichlet distribution with parameter $\alpha$ chosen so that

$$\text{KL}(\text{Dir}(\alpha, \dots, \alpha) \,\|\, \text{Dir}(1, \dots, 1)) = \text{target KL},$$

thereby controlling the divergence from the uniform training distribution. Intuitively, varying the target KL controls how concentrated generated samples are with respect to the simplex. ICPE's correctness and average stopping time across a range of KL values and number of actions is reported in tab. 2. Across all experiments, we observe that both correctness and stopping time remain remarkably stable, with only minor fluctuations within the reported confidence intervals. This suggests that ICPE is not excessively sensitive to moderate shifts in the environment distribution around the training family.

| KL Divergence From Uniform | Number of Actions = 4 | | Number of Actions = 6 | |
|---|---|---|---|---|
| | Correctness | Avg. Stop Time | Correctness | Avg. Stop Time |
| 0.00 | $0.91 \pm 0.01$ | $7.76 \pm 0.20$ | $0.91 \pm 0.01$ | $9.78 \pm 0.22$ |
| 0.25 | $0.91 \pm 0.01$ | $7.59 \pm 0.19$ | $0.91 \pm 0.01$ | $9.97 \pm 0.26$ |
| 0.50 | $0.90 \pm 0.01$ | $7.65 \pm 0.21$ | $0.91 \pm 0.01$ | $9.79 \pm 0.26$ |
| 1.00 | $0.90 \pm 0.01$ | $7.68 \pm 0.20$ | $0.90 \pm 0.01$ | $9.89 \pm 0.28$ |
| 2.00 | $0.89 \pm 0.01$ | $7.63 \pm 0.21$ | $0.90 \pm 0.01$ | $9.86 \pm 0.28$ |
| 4.00 | $0.89 \pm 0.01$ | $7.73 \pm 0.22$ | $0.90 \pm 0.01$ | $10.07 \pm 0.28$ |

| KL Divergence From Uniform | Number of Actions = 8 | | Number of Actions = 10 | |
|---|---|---|---|---|
| | Correctness | Avg. Stop Time | Correctness | Avg. Stop Time |
| 0.00 | $0.90 \pm 0.01$ | $11.37 \pm 0.22$ | $0.91 \pm 0.01$ | $15.41 \pm 0.37$ |
| 0.25 | $0.89 \pm 0.01$ | $11.45 \pm 0.26$ | $0.92 \pm 0.01$ | $15.13 \pm 0.37$ |
| 0.50 | $0.89 \pm 0.01$ | $11.54 \pm 0.26$ | $0.91 \pm 0.01$ | $15.35 \pm 0.40$ |
| 1.00 | $0.90 \pm 0.01$ | $11.33 \pm 0.24$ | $0.90 \pm 0.01$ | $15.33 \pm 0.42$ |
| 2.00 | $0.89 \pm 0.01$ | $11.41 \pm 0.28$ | $0.91 \pm 0.01$ | $15.54 \pm 0.41$ |
| 4.00 | $0.88 \pm 0.01$ | $11.47 \pm 0.28$ | $0.91 \pm 0.01$ | $15.22 \pm 0.40$ |

| KL Divergence From Uniform | Number of Actions = 12 | | Number of Actions = 14 | |
|---|---|---|---|---|
| | Correctness | Avg. Stop Time | Correctness | Avg. Stop Time |
| 0.00 | $0.91 \pm 0.01$ | $18.86 \pm 0.51$ | $0.91 \pm 0.01$ | $22.23 \pm 0.72$ |
| 0.25 | $0.91 \pm 0.02$ | $18.28 \pm 0.52$ | $0.92 \pm 0.01$ | $22.63 \pm 0.71$ |
| 0.50 | $0.91 \pm 0.01$ | $18.55 \pm 0.53$ | $0.91 \pm 0.01$ | $22.18 \pm 0.75$ |
| 1.00 | $0.90 \pm 0.01$ | $18.78 \pm 0.52$ | $0.91 \pm 0.01$ | $22.36 \pm 0.72$ |
| 2.00 | $0.91 \pm 0.01$ | $19.00 \pm 0.60$ | $0.91 \pm 0.01$ | $22.57 \pm 0.75$ |
| 4.00 | $0.91 \pm 0.01$ | $18.52 \pm 0.54$ | $0.91 \pm 0.01$ | $22.97 \pm 0.75$ |

Table 2: ICPE performance on stochastic fixed-confidence ($\delta = 0.1, N = 100$) bandits when test environments drift from the uniform training distribution by a prescribed KL distance (symmetric Dirichlet). Reported values are $95\%$ confidence intervals computed via hierarchical bootstrapping.

### D.1.3 BANDIT PROBLEMS WITH HIDDEN INFORMATION

**Magic Action Environments**   We evaluate `ICPE` in bandit environments where certain actions reveal information about the identity of the optimal arm, testing its ability to uncover and exploit latent structure under the fixed-confidence setting.

Each environment contains $K = 5$ arms. Non-magic arms have mean rewards sampled uniformly from $[1, 5]$, while the mean reward of the designated *magic action* (always arm 1) is defined as $\mu_m = \phi(\arg\max_{a \neq a_m} \mu_a)$ with $\phi(i) = i/K$. The magic action is not the optimal arm, but it encodes information about which of the other arms is. To control the informativeness of this signal, we vary the standard deviation of the magic arm $\sigma_m \in \{0.0, 0.1, \ldots, 1.0\}$, while fixing the standard deviation of all other arms to $\sigma = 1 - \sigma_m$.

`ICPE` is trained under the fixed-confidence setting with a target confidence level of 0.9. For each $\sigma_m$, we compare `ICPE`'s sample complexity to two baselines: (1) the average theoretical lower bound computed for the problem computed via averaging the result of Theorem B.26 over numerous environmental mean rewards, and (2) *I-IDS*, a pure-exploration information-directed sampling algorithm that uses `ICPE`'s $I$-network for posterior estimation. All methods are over 500 environments, with 10 trajectories per environment. 95% confidence intervals are computed using hierarchical bootstrapping with two levels.

Beyond the exploration efficiency analysis shown in Figure 4a, we also assess the correctness of each method's final prediction and its usage of the magic action. As shown in Figure 11a, both

**ICPE** and *I-IDS* consistently achieve the target accuracy of 0.9, validating their reliability under the fixed-confidence formulation.

Figure 11b plots the proportion of total actions that were allocated to the magic arm across different values of $\sigma_m$. While both methods adapt their reliance on the magic action as its informativeness degrades, *I-IDS* tends to abandon it earlier. This behavior suggests that **ICPE** is better able to retain and exploit structured latent information beyond what is captured by simple heuristics for expected information gain.

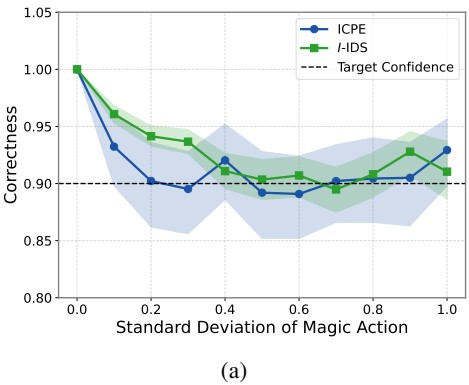
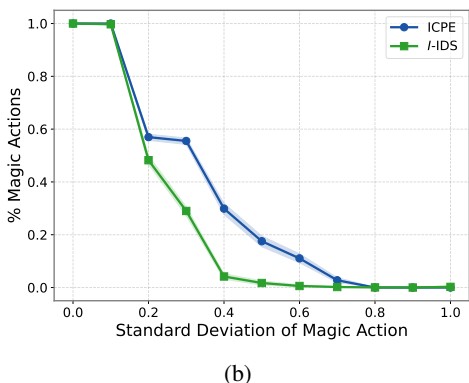

(a)                                                    (b)

Figure 11: (a) Final prediction accuracy across varying levels of noise in the magic action ($\sigma_m$). Both **ICPE** and *I-IDS* consistently achieve the target confidence threshold of 0.9. (b) Percentage of actions allocated to the magic arm as a function of $\sigma_m$. **ICPE** continues to exploit the magic action longer than *I-IDS*, suggesting more robust use of latent structure.

We also assess **ICPE**'s performance in a regret minimization setting. We define an *In-Context Explore-then-Commit* variant of **ICPE**, which explores until the $I$-network reaches confidence $1 - \delta$, then repeatedly selects the estimated optimal action. We compare this policy's cumulative regret to that of three standard algorithms: *UCB*, *Thompson Sampling*, and *IDS*, each initialized with Gaussian priors. For this evaluation, we fix $\sigma_m = 0.1$, $\sigma = 0.9$, and $\delta = 0.01$.

Implementation details for $I$-IDS and In-Context Explore-then-Commit are provided in Sections C.3.2 and C.3.3 respectively.

**Magic Chain Environments**    To assess **ICPE**'s ability to perform multi-step reasoning over latent structure, we evaluate it in environments where identifying the optimal arm requires sequentially uncovering a chain of informative actions. In these *magic chain* environments, each magic action reveals partial information about the next, culminating in identification of the best arm.

We use $K = 10$ arms and vary the number of magic actions $n \in \{1, 2, \ldots, 9\}$. Mean rewards for magic actions are defined recursively as:

$$\mu_{i_j} = \begin{cases} \phi(i_{j+1}), & \text{if } j = 1, \ldots, n-1, \\ \phi\Big(\arg\max_{a \notin \{i_1, \ldots, i_n\}} \mu_a\Big), & \text{if } j = n, \end{cases}$$

where $\phi(i) = i/K$, and the remaining arms have mean rewards sampled uniformly from $[1, 2]$. All rewards are deterministic (zero variance).

**ICPE** is trained under the fixed-confidence setting with $\delta = 0.99$. For each $n$, five models are trained across five seeds. We compare **ICPE**'s average stopping time to the theoretical optimum (computed via Theorem B.29) and to the *I-IDS* baseline equipped with access to the $I$-network. Each model is evaluated over 1000 test environments per seed. 95% confidence intervals are computed using hierarchical bootstrapping.

In interpreting the results from Figure 4b, we observe that for environments with one or two magic actions, **ICPE** reliably learns the optimal policy of following the magic chain to completion. In these cases, the agent is able to identify the optimal arm without ever directly sampling it, by exploiting the structured dependencies embedded in the reward signals of the magic actions. Figure 12 illustrates a

representative trajectory from the two-magic-arm setting, where the first magic action reveals the location of the second, which in turn identifies the optimal arm. The episode terminates without requiring the agent to explicitly sample the best arm itself.

Figure 12: Example trajectory in the 2-magic-arm environment. `ICPE` follows the magic chain: the first magic action reveals the second, which identifies the optimal arm.

For environments with more than two magic actions, however, `ICPE` learns a different strategy. As the length of the magic chain increases, the expected sample complexity of following the chain from the start becomes suboptimal. Instead, `ICPE` learns to randomly sample actions until it encounters one of the magic arms and then proceeds to follow the chain from that point onward. This behavior represents an efficient, learned compromise between exploration cost and informativeness. Figure 13 shows an example trajectory from the six-magic-arm setting, where the agent initiates random sampling until it lands on a magic action, then successfully follows the remaining chain to identify the optimal arm.

Figure 13: Example trajectory in the 6-magic-arm environment. Rather than starting from the first magic action, `ICPE` samples randomly until finding a magic action and then follows the chain to the optimal arm.

## D.2 SEMI-SYNTHETIC PIXEL SAMPLING

To evaluate `ICPE` in a setting that more closely resembles real-world decision-making tasks, we designed a semi-synthetic environment based on the MNIST dataset (LeCun et al., 1998), where the agent must adaptively reveal image regions to classify a digit while minimizing the number of pixels observed. This experiment serves as a proof-of-concept for using `ICPE` in perceptual tasks where observations are costly and information must be acquired efficiently.

**Environment Details.** Each MNIST image is partitioned into 36 non-overlapping $5 \times 5$ pixel regions, defining an action space of size $K = 36$. At each timestep, the agent selects a region to

reveal, progressively unmasking the image. The agent begins with a fully masked image and has a fixed budget of $N = 12$ steps to acquire information and make a prediction.

To prevent overfitting and encourage generalizable policies, we apply strong augmentations at each episode: random rotations ($\pm 30°$), translations (up to 2 pixels), Gaussian noise ($\mathcal{N}(0, 0.1)$), elastic deformations, and random contrast adjustments. These augmentations ensure that agents cannot memorize specific pixel layouts and must instead learn adaptive exploration strategies.

**Model Architecture and Optimization.** Due to the visual nature of the task, we use a convolutional encoder rather than a transformer. The `ICPE` critic network combines a CNN image encoder with a separate action-count encoder. The CNN consists of 3 convolutional blocks with 16 base channels, followed by max pooling and global average pooling. The action counts (which track how often each region has been sampled) are passed through a linear embedding layer with 32 output units, followed by ReLU activation and LayerNorm. The image and action embeddings are concatenated and processed through two residual MLP layers, producing $Q$-values over actions. The $I$-network shares the same architecture but outputs logits over 10 digit classes.

All models are implemented in PyTorch and trained with Adam using a learning rate of $1 \times 10^{-4}$. Training is performed over 500,000 episodes using 40 parallel environment instances. We use a batch size of 128, a replay buffer of size 100,000, and a discount factor $\gamma = 0.999$. The $Q$-network is updated using Polyak averaging with coefficient 0.01, and the $I$-network is updated every two steps using 30 bootstrap batches. To populate the buffer initially, we perform 10 batches of bootstrap updates before standard training begins. Gradients are clipped to a maximum norm of 2.

**Pretraining the Inference Network.** To provide stable reward signals and ensure consistency with baselines, we pretrain a separate CNN classifier to predict digit labels from fully revealed images. This classifier consists of three convolutional layers with max pooling, followed by two linear layers and a softmax head. The classifier is trained on the same augmented data used during `ICPE` training and is frozen during exploration learning. Its softmax confidence for the correct digit is used as the reward signal. This setup simulates realistic scenarios in which high-quality predictive models already exist for fully observed data (e.g., in clinical diagnosis).

**Evaluation.** We compare `ICPE` to two baselines: *Uniform Sampling*, which selects image regions uniformly at random at each timestep, and *Deep CMAB* (Collier & Llorens, 2018), a contextual bandit algorithm that uses a Bayesian neural network to model $p(r \mid x, a)$ and performs posterior sampling via dropout.

The Deep CMAB model uses a convolutional encoder to extract image features, which are concatenated with a learned embedding of the action count vector. The combined representation is passed through a multilayer perceptron with dropout applied to each hidden layer. At each decision point, the agent samples a dropout rate from a uniform distribution over $(0, 1)$ and uses the resulting forward pass as a sample from the posterior over rewards (Thompson sampling). The reward signal for each action is computed using the pretrained MNIST classifier: specifically, the increase in softmax probability for the correct digit class after a new region is revealed.

We train Deep CMAB for 100,000 episodes using Adam optimization. During training, the agent interacts with multiple MNIST instances in parallel, and updates its model based on the marginal improvement in confidence after each action. The model learns to maximize this incremental reward signal by associating particular visual contexts with the most informative actions.

For each trained model, we sample 1000 test environments and report on (1) the average final classification accuracy by the pretrained classifier at the end of trajectory, and (2) the average number of regions used before prediction. Confidence intervals are computed via bootstrapping.

**Adaptive Sampling Analysis.** To assess whether agents learn digit-specific exploration strategies, we analyze the distribution of selected image regions across digit classes. For each agent, we compute pairwise chi-squared tests between all digit pairs, testing whether the distributions of selected regions are statistically distinguishable.

To ensure sufficient support for the test, we only compare digit pairs that each have at least five trajectories and remove actions that appear in fewer than five total samples across the two classes. For

each qualifying digit pair, we construct a $2 \times \tilde{K}$ contingency table, where $\tilde{K}$ is the number of region indices that are meaningfully used by either digit. The rows correspond to digit classes, and each column counts how many samples from each class selected the corresponding region at least once.

We apply the chi-squared test of independence to each contingency table. A low p-value indicates that the region selection distributions for the two digits are significantly different, suggesting digit-specific adaptation. By comparing the number and strength of significant differences across agents, we evaluate the extent to which each method tailors its exploration policy to the structure of the input class.

We visualize the resulting pairwise p-values in Figure 14 using a heatmap. Each cell shows the chi-squared test p-value between a pair of digits. Lower values (blue cells) indicate greater divergence in sampling behavior, and thus more adaptive and digit-specific strategies.

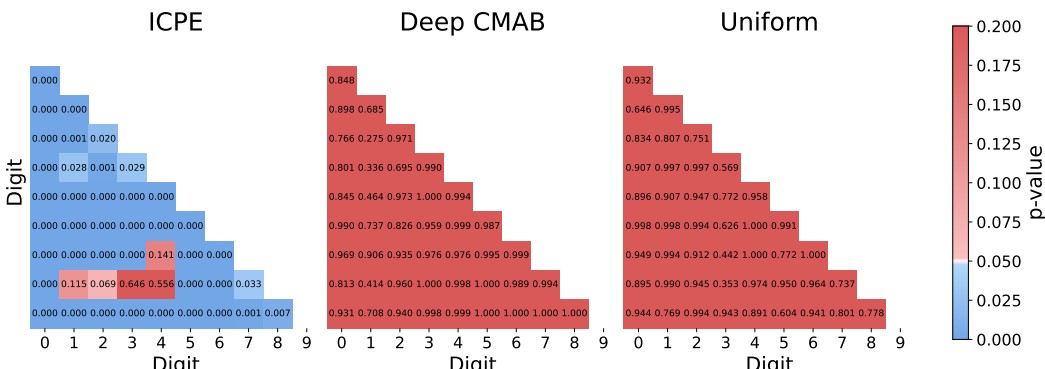

Figure 14: Pairwise chi-squared test p-values for region selection distributions across digit classes. Lower values indicate more statistically distinct exploration behaviors.

For further intuition into the sampling process, Figure 15 shows a representative example of the `ICPE` pipeline progressively revealing image regions and correctly classifying the digit '2'. This highlights the interplay between exploration and inference as the agent strategically uncovers informative regions to guide its decision.

To illustrate the impact of input corruption, Figure 16 presents an example where `ICPE` fails to correctly classify the digit. Although the agent successfully reveals the central body of the digit, the applied augmentations distort the image to the extent that the digit becomes visually ambiguous. In this case, the agent incorrectly predicts an '8' when the true label is a '9', underscoring the challenge introduced by realistic image corruptions in this setting.

## D.3 MDP PROBLEMS: MAGIC ROOM

The Magic Room is a sequential decision-making environment structured as a $K \times K$ grid-shaped room containing four doors, each positioned at the midpoint of one of the four walls (top, bottom, left, right). At the beginning of each episode, exactly one of these doors is randomly chosen to be the correct door ($H^\star$), unknown to the agent.

The agent's goal is to identify and pass through the correct door. Each episode lasts for a maximum of $N = K^2$ time steps, during which the agent navigates the grid, observes clues, and attempts to determine the correct door. Two binary clues, each randomly assigned a location within the sub-grid $[1, 1] \times [K - 1, K - 1]$, are placed in the room at the start of each episode. Each clue has a binary value, randomly set to either $-1$ or $1$. Collecting both clues provides sufficient information to unambiguously determine the correct door, given that the agent has learned the mapping from clue configurations to door identity.

At each time step $t$, the agent observes the state vector:

$$x_t = (z_t, y_t, c_{1,t}, c_{2,t}, r_t),$$

where:

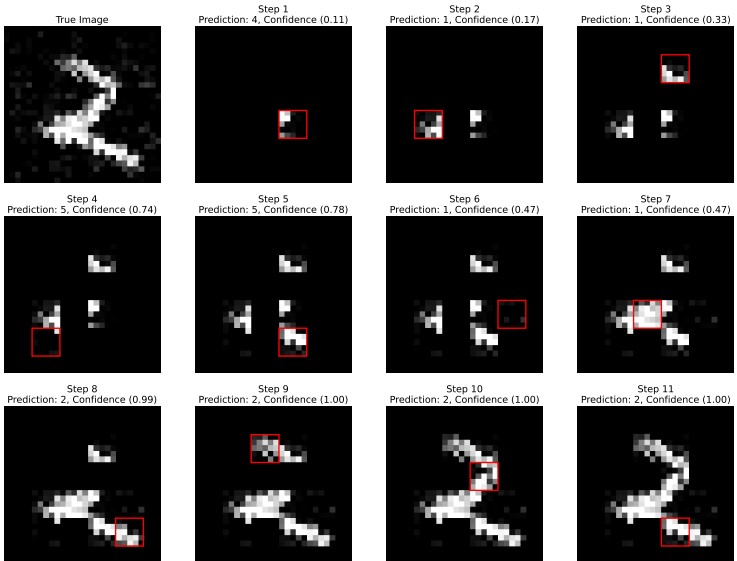

Figure 15: Illustrative example of the `ICPE` agent revealing regions of an MNIST digit and correctly classifying it as a '2'. The sequence shows the intermediate revealed image and predicted label at each timestep.

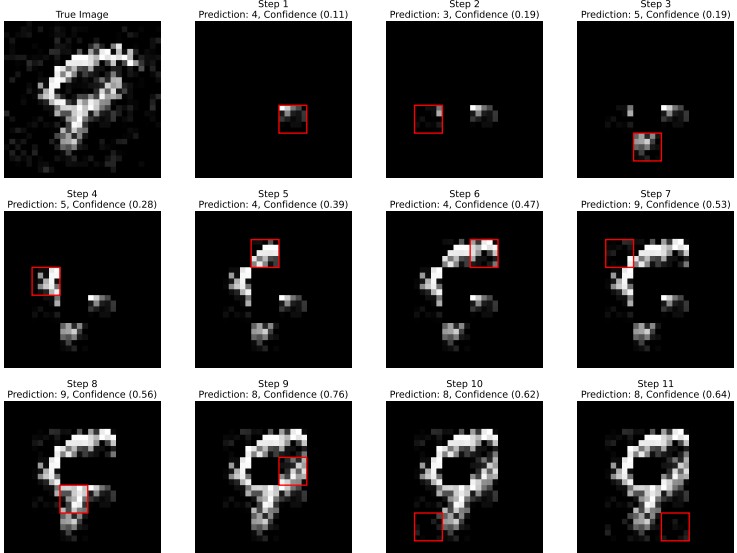

Figure 16: Example of an incorrect classification due to aggressive data augmentations. Although the agent reveals the central region of the digit, the distortions cause it to misclassify a '9' as an '8'.

- $(z_t, y_t)$ are the agent's current coordinates on the grid.

- $c_{i,t} \in \{-1, 0, 1\}$ indicates the status of clue $i$: it equals $0$ if clue $i$ has not yet been observed by the agent, or it equals either $-1$ or $1$ if the clue has been observed.

- $r_t \in \{0, 1\}$ represents the reward received at time $t$. Specifically, upon passing through a door:

  – If the chosen door is the correct one, the agent receives a reward of $1$ with probability $\frac{1}{4}$, and a reward of $0$ otherwise.

  – If the chosen door is incorrect, the agent always receives a reward of $0$.

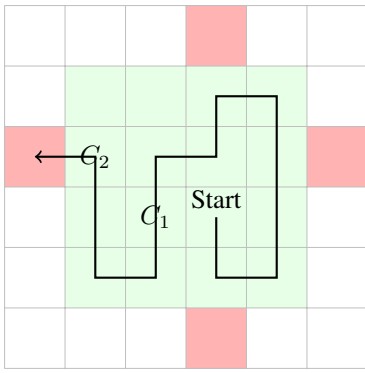

Figure 17: Magic room: example of trajectory of the
icpe agent.

An episode terminates when the agent chooses to pass through any of the four doors, irrespective of correctness, or when the horizon $N = K^2$ steps is reached. Upon termination, the agent is required to explicitly select which door it believes to be the correct one.

| Method | Average Correctness | | Average Stopping Time | |
|---|---|---|---|---|
| | $K = 6$ | $K = 8$ | $K = 6$ | $K = 8$ |
| **ICPE** | 0.953 (0.940, 0.968) | 0.948 (0.941, 0.954) | 13.721 (13.298, 14.165) | 27.704 (27.296, 28.086) |

Table 3: Magic Room: correctness and stopping times (mean and 95% CI) for $K = 6$ and $K = 8$.

This setup provides two distinct strategies for the agent:

1. **Luck-based strategy**: The agent directly attempts to pass through a door, observing the reward to determine correctness. A positive reward conclusively indicates the correct door; a zero reward provides no additional information.

2. **Inference-based strategy**: The agent efficiently navigates the room, locates both clues to deduce the identity of the correct door, and subsequently exits through that door.

Thus, optimal behavior requires an effective exploration of the room to finish as quickly as possible.

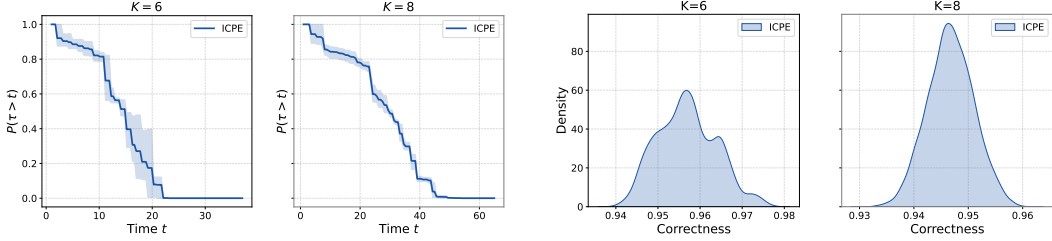

Figure 18: Magic room environment. Left: survival function $\mathbb{P}(\tau > t)$ for $K = 6$ and $K = 8$. Right: density of the correctness for $K = 6$ and $K = 8$.

We trained **ICPE** on 3 seeds, using the fixed confidence setting (disabling the stopping action) using $\delta = 0.05$ and evaluated the policies on 4500 episodes for $K = 6$ and $K = 8$. In tab. 3 are shown the statistics of the average correctness and of the stopping time.

In fig. 17 we can see a sample trajectory taken by **ICPE**. Starting from the middle of the room, **ICPE** follows a path that allows to find the clues $C_1, C_2$ in the green area. As soon as the second clue is found, it goes through the closest door.

In fig. 18, we present the survival functions of the stopping time $\tau$ for environments with grid sizes $K = 6, 8$, alongside the corresponding correctness densities. Lastly, fig. 19 illustrates the relationship

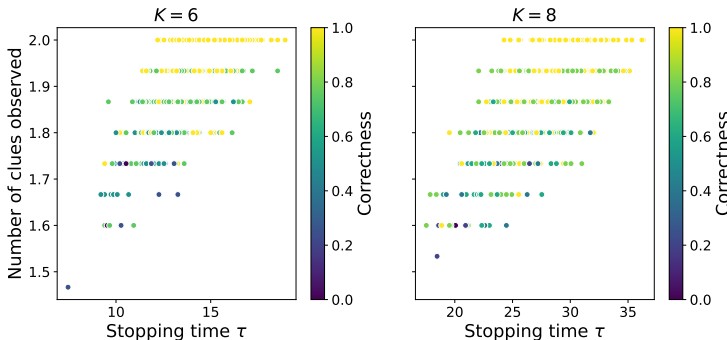

Figure 19: Magic room environment. Relationship among agent correctness, the number of clues observed, and the stopping time.

among agent correctness, the number of clues observed, and the stopping time. Specifically, smaller stopping times correlate with fewer observed clues, leading to lower correctness. Conversely, when the agent observes both clues, it consistently selects the correct door, demonstrating that it has effectively learned the association between the clues and the correct hypothesis.

### D.4    EXPLORATION ON FEEDBACK GRAPHS

In the standard bandits setting we studied in Section 4.1, the learner observes the reward of the selected action, while in full-information settings, all rewards are revealed. Feedback graphs generalize this spectrum by specifying, via a directed graph $G$ which additional rewards are observed when a particular action is chosen. Each node corresponds to an action, and an edge from $u$ to $v$ means that playing $u$ may reveal feedback about $v$.

While feedback graphs have been widely studied for regret minimization (Mannor & Shamir, 2011), their use in pure exploration remains relatively underexplored (Russo et al., 2025). We study them here as a challenging and structured testbed for in-context exploration. Unlike unstructured bandits, these environments contain latent relational structure and stochastic feedback dependencies that must be inferred and exploited to explore efficiently.

Formally, we define a feedback graph as an adjacency matrix $G \in [0, 1]^{K \times K}$, where $G_{u,v}$ denotes the probability that playing action $u$ reveals the reward of action $v$. The learner observes a feedback vector $r \in \mathbb{R}^K$, where each coordinate is revealed independently with probability $G_{u,v}$:

$$r_v \sim \begin{cases} \mathcal{N}(\mu_v, \sigma^2), & \text{with probability } G_{u,v}, \\ \text{no observation}, & \text{otherwise}. \end{cases}$$

This setting allows us to test whether `ICPE` can learn to uncover and leverage latent graph structure to guide exploration. As in the bandits setting, we have a finite number of actions $\mathcal{A} = \{1, \ldots, K\}$, corresponding to the actions (or vertices) in a feedback graph $G$. The learner's goal is to identify the best action, where $H^\star = \arg\max_a \mu_a$. At each time step $t$, the observation is the partially observed reward vector $x_t = r_t$.

We evaluate performance on best-arm identification tasks across three representative feedback graph families:

- **Loopy Star Graph** (Figure 20): A star-shaped graph with self-loops, parameterized by $(p, q, r)$. The central node observes itself with probability $q$, one neighboring node with probability $p$, and all others with probability $r$. When $p$ is small, it may be suboptimal to pull the central node, requiring the agent to adapt its strategy accordingly.

- **Ring Graph** (Figure 21): A cyclic graph where each node observes its right neighbor with probability $p$ and its left neighbor with probability $1 - p$. Effective exploration requires reasoning about which neighbors provide more informative feedback.

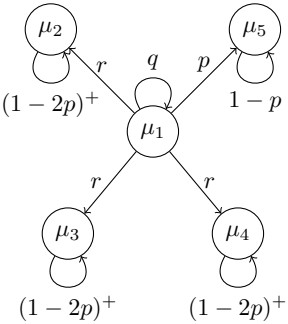
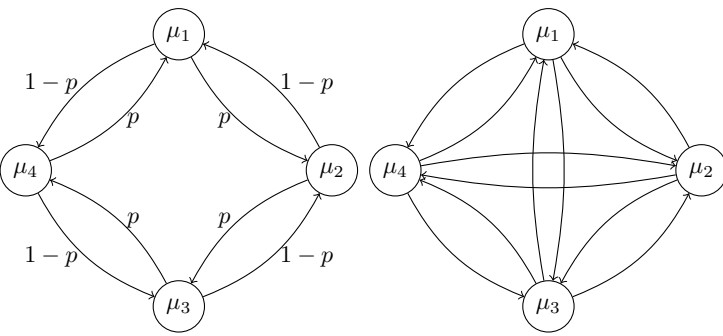

Figure 20: Loopy star graph.    Figure 21: Ring graph.    Figure 22: Loopless clique graph.

- **Loopless Clique Graph** (Figure 22): A fully connected graph with no self-loops. Edge probabilities are defined as:

$$G_{u,v} = \begin{cases} 0 & \text{if } u = v, \\ \frac{p}{u} & \text{if } v \neq u \text{ and } v \text{ is odd}, \\ 1 - \frac{p}{u} & \text{otherwise.} \end{cases}$$

Here, informativeness varies systematically with action index, requiring the learner to infer which actions are most useful.

These environments offer a diverse testbed for evaluating whether `ICPE` can uncover and exploit complex feedback structures without direct access to the underlying graph.

**Fixed-Horizon.**    For each graph family, mean rewards were sampled uniformly from $[0, 1]$ with fixed variance $0.2$, using hyperparameters: $(p, q, r) = (0.25, 0.3, 0.35)$ for the loopy star graph, $p = 0.3$ for the ring, and $p = 0.5$ for the loopless clique. We considered both small ($K = 5$, $H = 25$) and large ($K = 10$, $H = 50$) environments.

`ICPE` was compared to three baselines: Uniform Sampling, EXP3.G (Rouyer et al., 2022), and Tas-FG (Russo et al., 2025). All methods performed maximum likelihood inference at the end of the trajectory. Table 4 reports the average probability of correctly identifying the best arm.

| Algorithm | Loopy Star | | Loopless Clique | | Ring | |
|---|---|---|---|---|---|---|
| | **Small** | **Large** | **Small** | **Large** | **Small** | **Large** |
| **ICPE** | $\mathbf{0.88 \pm 0.01}$ | $0.59 \pm 0.02$ | $\mathbf{0.95 \pm 0.01}$ | $0.79 \pm 0.04$ | $\mathbf{0.79 \pm 0.01}$ | $0.51 \pm 0.03$ |
| TasFG | $0.82 \pm 0.01$ | $\mathbf{0.73 \pm 0.02}$ | $0.84 \pm 0.01$ | $\mathbf{0.83 \pm 0.01}$ | $0.70 \pm 0.02$ | $0.56 \pm 0.02$ |
| EXP3.G | $0.66 \pm 0.02$ | $0.40 \pm 0.01$ | $0.84 \pm 0.01$ | $0.78 \pm 0.02$ | $0.77 \pm 0.02$ | $0.52 \pm 0.02$ |
| Uniform | $0.73 \pm 0.02$ | $0.60 \pm 0.02$ | $0.86 \pm 0.01$ | $0.79 \pm 0.02$ | $0.78 \pm 0.02$ | $\mathbf{0.62 \pm 0.02}$ |

Table 4: Probability of correctly identifying the best arm. Small environments: $K = 5$, $H = 25$; Large: $K = 10$, $H = 50$. Results reported as mean ± 95% CI.

`ICPE` outperforms all baselines in small environments across all graph families, highlighting its ability to learn efficient strategies from experience. Performance slightly degrades in larger environments, likely due to difficulty in credit assignment over long horizons. Still, `ICPE` remains competitive, validating its capacity to generalize across graph-structured settings.

**Fixed-Confidence.**    We next tested `ICPE` in a fixed-confidence setting, using the same graph families but setting the optimal arm's mean to $1$ and all others to $0.5$ to facilitate faster convergence. `ICPE` was trained for $K = 4, 6, \ldots, 14$ with a target error rate of $\delta = 0.1$. We compared it to Uniform Sampling, EXP3.G, and Tas-FG using a shared stopping rule from (Russo et al., 2025).

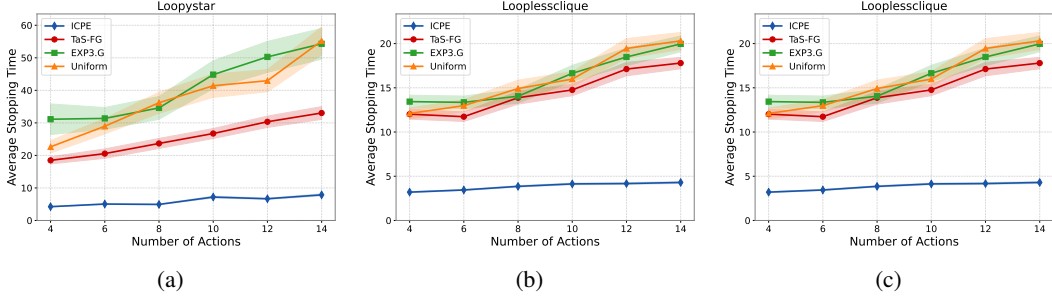

Figure 23: Sample complexity comparison under the fixed-confidence setting for: (a) Loopy Star, (b) Loopless Clique, and (c) Ring graphs.

As shown in Figure 23, `ICPE` consistently achieves significantly lower sample complexity than all baselines. This suggests that `ICPE` is able to meta-learn the underlying structure of the feedback graphs and leverage this knowledge to explore more efficiently than *uninformed* strategies. These results align with expectations: when environments share latent structure, learning to explore from experience offers a substantial advantage over fixed heuristics that cannot adapt across tasks.

### D.5 META-LEARNING BINARY SEARCH

To test `ICPE`'s ability to recover classical exploration algorithms, we evaluate whether it can autonomously meta-learn binary search.

We define an action space of $\mathcal{A} = \{1, \ldots, K\}$, where $K$ is the upper bound on the possible location of the hidden target $H^\star \sim \mathcal{A}$. Pulling an arm above or below $H^\star$ yields a observation $x_t = -1$ or $x_t = +1$, respectively—providing directional feedback.

We train `ICPE` under the fixed-confidence setting for $K = 2^3, \ldots, 2^8$, using 150,000 in-context episodes and a target error rate of $\delta = 0.01$. Evaluation was conducted on 100 held-out tasks per setting. We report the minimum accuracy, mean stopping time, and worst-case stopping time, and compare against the theoretical binary search bound $O(\log_2 K)$.

| Number of Actions ($K$) | Minimum Accuracy | Mean Stopping Time | Max Stopping Time | $\log_2 K$ |
|---|---|---|---|---|
| 8 | 1.00 | $2.13 \pm 0.12$ | 3 | 3 |
| 16 | 1.00 | $2.93 \pm 0.12$ | 4 | 4 |
| 32 | 1.00 | $3.71 \pm 0.15$ | 5 | 5 |
| 64 | 1.00 | $4.50 \pm 0.21$ | 6 | 6 |
| 128 | 1.00 | $5.49 \pm 0.23$ | 7 | 7 |
| 256 | 1.00 | $6.61 \pm 0.26$ | 8 | 8 |

Table 5: `ICPE` performance on the binary search task as the number of actions $K$ increases.

As shown in Table 5, `ICPE` consistently achieves perfect accuracy with worst-case stopping times that match the optimal $\log_2(K)$ rate, demonstrating that it has successfully rediscovered binary search purely from experience. While simple, this task illustrates `ICPE`'s broader potential to learn efficient search strategies in domains where no hand-designed algorithm is available.