# OpenReview forum: "In-Context Learning for Pure Exploration"
_ICLR.cc/2026/Conference — ICLR 2026 Poster_

### Official Review · Reviewer_g3VE · 2025-10-27

**Soundness:** 3
**Presentation:** 2
**Contribution:** 2
**Rating:** 4
**Confidence:** 4

**Summary:**

This paper introduces In-Context Pure Exploration (ICPE), a meta-learning approach for active sequential hypothesis testing that connects the pure exploration theory in bandits to sequential modeling using transformers. The key idea is to adaptively collect data from an environment M to identify a ground-truth hypothesis H* with minimal samples in either fixed-confidence or fixed-budget (N samples) settings. They have extensive theoretical analysis, starting with Proposition 3.1 that establishes that the optimal inference rule is $I^{\star}(z) = arg \max_H P(H^{\star}=H|D_t=z)$, making posterior computation central. For policy optimization, the paper derives reward functions that emerge from the dual formulation - in fixed-budget, rewards equal the maximum posterior value at termination, while fixed-confidence uses a Lagrangian formulation with rewards $r_t, \lambda = -1 + \lambda\cdot1_{stop}·\max_H P(H^{\star}=H|D_t)$. Note that this is heavily influenced by the Best-Arm Identification (BAI) setting in MAB. This is a special case where hypotheses coincide with actions (H=A) and $H^{\star}=\arg \max_a E[\text{reward}|a]$. They also discuss how ICPE generalizes beyond standard BAI by handling history-dependent observation kernels $P_t(·|D_t,a)$ rather than memoryless $P(·|H^{\star},a)$, and learning the inference rule rather than assuming known likelihood functions. The paper proves that optimal policies satisfy Bellman equations with Q-functions incorporating posterior distributions, establishing the MDP formulation's validity. ICPE uses two Transformer networks: $I_{\phi}$ approximating posteriors via cross-entropy loss, and $Q_{\theta}$ learning the exploration policy via DQN with information-theoretic rewards. This architecture handles the credit assignment problem in sequential exploration. The approach unifies several exploration settings - it recovers classical algorithms like binary search, exploits structured dependencies in "magic action" bandits where certain arms reveal information about others, and generalizes to non-bandit settings like image classification, where actions (pixel regions) differ from hypotheses (digit classes). Experiments demonstrate that ICPE matches theoretical lower bounds in structured bandits (Theorem B.20) while maintaining competitive performance in standard settings, validating both theoretical predictions and practical effectiveness.

**Strengths:**

1. The paper provides a rigorous theoretical justification showing how the optimal inference rule and policy come from existing bandit theory. The problem formulation, along with the reward function, is principled rather than heuristic. The connection between posterior maximization and Q-learning is well established.
2. The evaluation spans diverse domains from classical bandits to structured environments with hidden information, demonstrating ICPE's versatility. The magic action experiments particularly showcase the method's ability to discover and exploit non-obvious information structures.
3. ICPE learns both components from data, handling history-dependent observations and unknown likelihood functions, making it more applicable to (some) real-world scenarios.

**Weaknesses:**

1. My main concern is that it is not clear to me how a single approach works for both best confidence and best budget setting and is simultaneously optimal. Is this a best-of-both-worlds approach? Or is there a tradeoff? This requires more discussion.
2. The writing needs more improvement. In the discussion of Algorithm 1, the stop action was suddenly introduced in lines 258-262. It is not clear to me why it is necessary. Also, the stop action is not directly motivated from theory, and seems to be heuristic. Each theorem requires a dedicated discussion section to bring out the novelty in the proofs. Again, the novelty of the theoretical proofs is not clear to me. I have not checked the appendix.
3. While the paper proves optimality under certain conditions, it doesn't provide finite-sample complexity bounds for ICPE or formal guarantees on the meta-learning convergence, leaving uncertainty about when the approach will succeed. This deviates from the traditional BAI analysis under the bandit framework

**Questions:**

1. Can a single model trained on the trajectory-level data handle both settings at test time? How does performance degrade when using a fixed-budget trained model on fixed-confidence tasks (or vice versa)?
2. Given the theoretical nature of the paper, the above question should also be answered from a theoretical standpoint. For example, see https://papers.nips.cc/paper_files/paper/2012/file/8b0d268963dd0cfb808aac48a549829f-Paper.pdf.
3. Why can't the stopping decision emerge from the value function without an explicit action ($a_{stop}$)? How does forcing stop as an action affect the exploration strategy compared to value-based stopping?
4. How many training tasks are needed for $\epsilon$-optimal performance? The paper provides no sample complexity for the meta-learning phase.
5. More rigorous real-world experiments should be done.

If my questions and concerns are addressed, I am willing to raise my score.

**Details Of Ethics Concerns:**

Not applicable.

---

> ### Author Response · Authors · 2025-11-21
> **Response to Reviewer g3VE Part 1**
>
> We would like to  thank the reviewer for their thoughtful evaluation and recognition of the theoretical and empirical contributions of ICPE.
>
> Below, we address the questions of the reviewer.
>
> > - it is not clear to me how a single approach works for both best confidence and best budget setting and is simultaneously optimal. [...]
> > - Can a single model trained on the trajectory-level data handle both settings at test time? [...]
>
> We acknowledge the confusion regarding the distinction between fixed-confidence and fixed-horizon algorithms. We do not simultaneously train a single model for both settings, and we do not claim that one trained model is simultaneously optimal for both fixed-confidence and fixed-budget BAI.
>
> What we show in the paper is that, in both settings, one can use a similar  methodology to train a sampling rule  $\pi$:
>
> - In the fixed budget setting, theorem 3.2 defines an optimal policy $\pi^\star$ using a $Q$-function with terminal reward $r_N({\cal D}_N)=\max_H \mathbb{P}(H^\star=H|{\cal D}_N)$, and zero reward at intermediate steps. This leads to the DQN loss in eq. (8).
>
> - In the fixed confidence setting,  theorem 3.3 defines a different optimal policy. This optimal policy is defined with respect to the reward function in eq. (5), an explicit stopping action $a_{\rm stop}$ and a dual variable $\lambda$. This gives the loss in eqs. (9)-(10).
>
> Algorithm 1 abstracts the main ideas in a unified way, but in practice we train separate models for the fixed-budget and fixed-confidence regimes, each with its own reward and $Q$-function. We will clarify this point in the revised version of the paper.
>
> > - In the discussion of Algorithm 1, the stop action was suddenly introduced in lines 258-262. It is not clear to me why it is necessary. Also, the stop action is not directly motivated from theory, and seems to be heuristic.  [...]
> > - Why can't the stopping decision emerge from the value function without an explicit action ($a_{stop}$)? How does forcing stop as an action affect the exploration strategy compared to value-based stopping?
>
> We thank the reviewer for raising this important point. The stopping action is introduced in lines 179-182 (page 4), and later it is used in lines 258-262, as pointed out by the reviewer.
>
> In our implementation, the stopping decision is modeled as an additional action $a_{\text{stop}}$ in the action set. At each time step  the policy selects
>
> $$
> a_t \in \arg\max_{a \in \mathcal{A} \cup \\{a_{\text{stop}}\\}} Q_\theta(D_t,a),
> $$
>
> and if $a_t = a_{\text{stop}}$  the trajectory terminates  and we output the final hypothesis $\hat H_t = \arg\max_{H} I_\phi(H| D_t)$.
>
> Importantly, adding a stopping action $a_{\text{stop}}$ is not a heuristic, and we show that it  follows directly from our fixed-confidence formulation. Specifically, we provide the full derivation in Appendix B.1.4 (`"Fixed confidence setting: dual problem formulation"). We start from the constrained problem
>
> $$
> \inf_{\pi,\tau,I} \mathbb{E}^\pi[\tau]\quad\text{s.t.}\quad \mathbb{P}^\pi(H^\star = I_\tau(\mathcal{D}_\tau)) \ge 1-\delta,
> $$
>
> derive its Lagrangian dual, and show that for any stopping time $\tau$ adapted to $(\sigma(\mathcal{D}_t))_t$ there exists a policy $\bar{\pi}$ on
>
> an extended action space $\bar{\mathcal{A}} = \mathcal{A} \cup \\{ a_{\mathrm{stop}} \\}$, with stopping rule $\bar{\tau} = \inf\\{ t : a_t = a_{\mathrm{stop}} \\}$, such that $V_\lambda(\pi,\tau,I) = V_\lambda(\bar{\pi},I)$ (Lemma B.6 in Appendix B). Thus, encoding $\tau$ as the hitting time of $a_{\mathrm{stop}}$ is equivalent: it is the same as optimizing over all admissible stopping rules.
>
> Regarding the question of "value-based stopping":  in this dual formulation, the stopping decision already *emerges*  from the value function, in the sense that the agent stops exactly when $Q_\theta(D_t,a_{\text{stop}})$ becomes larger than the $Q$-values of all the other actions.  This is also something that we show in appendix  B.1.4. This stopping action does not restrict the exploration strategy: by the equivalence result above, any admissible pair $(\pi,\tau)$ in the original problem can be realized by some policy over $\mathcal{A}\cup\\{a_{\text{stop}}\\}$, and we empirically find that learning $Q_\theta(\cdot,a_{\text{stop}})$ jointly with the other actions leads to a more stable training procedure in the fixed-confidence setting.

---

> ### Author Response · Authors · 2025-11-21
> **Response to Reviewer g3VE Part 2**
>
> > - Each theorem requires a dedicated discussion section to bring out the novelty in the proofs. Again, the novelty of the theoretical proofs is not clear to me. I have not checked the appendix.
>
> We appreciate this comment and we will clarify the role and novelty of our theoretical results. The purpose of Section 3.1  (and Appendix B)  is to show that active sequential hypothesis testing, in both the fixed-budget and fixed-confidence regimes, with possible history-dependent observation kernels, allows one to learn  the sampling/inference rules using an RL formulation from which ICPE is derived.
>
> Concretely, Proposition 3.1 (Appendix B.1.2)  identifies the optimal inference rule as the MAP estimator based on the posterior $P(H^\star  = H | D_t)$,  so that learning $I_\phi$  amounts to learning this posterior. Theorem 3.2 and Proposition B.3 show that, in the fixed-budget setting, an optimal sampling policy is obtained by a greedy policy with respect to a $Q$-function whose terminal reward is the maximum posterior mass $r_N(D_N) =\max_H P(H^\star = H| D_N)$.  For the fixed-confidence setting, Propositions B.5–B.11  first recast the constrained problem via a Lagrangian dual, then prove that *any*  admissible stopping rule $\tau$  can be represented as hitting time of an absorbing stop action $a_{\text{stop}}$,  and finally show that the resulting dual problem is solved by a greedy policy on a $Q$-function  with reward in eq. (5), and, importantly, that it achieves the right level of correctness. This establishes that the $\delta$-aware  stopping rule and the exploration strategy can both be learned on the extended action space $\mathcal  A \cup \\{a_{\text{stop}}\\}$.
>
> We will ensure to revise the manuscript to add short discussion paragraphs after Proposition 3.1, Theorem 3.2,  and Theorem 3.3  explicitly stating these takeaways and how they motivate the ICPE architecture and its reward design.
>
> > - While the paper proves optimality under certain conditions, it doesn't provide finite-sample complexity bounds for ICPE or formal guarantees on the meta-learning convergence, leaving uncertainty about when the approach will succeed. [...]
> > - How many training tasks are needed for $\epsilon$-optimal performance? The paper provides no sample complexity for the meta-learning phase.
>
> We would like to emphasize that the current manuscript already carries out a substantial amount of theoretical work: we fully characterize Bayes-optimal sampling and stopping rules in both fixed-budget and fixed-confidence regimes (Section 3 and Appendix B), and we use these characterizations to derive the RL objective optimized by ICPE.
>
> At the same time, we agree that a finite-sample characterization of the meta-learning phase is important, and we have already made substantial progress toward a result that bounds the number of training tasks required for ICPE to achieve $\epsilon$-optimal performance. In the rebuttal period, we will upload a new version of the manuscript including this new theoretical result and its proof.
>
> > More rigorous real-world experiments should be done.
>
> We agree with the reviewer that evaluating ICPE on real-world applications is an important direction of future work,  but we see it as a complementary  result to the main goal of this paper. Our main focus here is to introduce and analyze ICPE as a general framework that can address a broad family of pure exploration problems, and we validate ICPE on synthetic and structured benchmarks. The current manuscript already combines substantial theory (see Section 3 and Appendix B) with experiments across multiple problem classes, including BAI and active search problems. We view real-world experiments as a natural and promising direction of future work, and we will ensure to clarify this positioning and briefly discuss potential applications in the revised version.
>
> We sincerely thank the reviewer again for their feedback. We have clarified the distinction between fixed-budget and fixed-confidence training, formally motivated the stop action from our dual fixed-confidence formulation, and highlighted the role and novelty of our theoretical results. We have also outlined our ongoing work on finite-sample guarantees for the meta-learning phase, which we will upload shortly. We hope that these clarifications address your concerns, and in that case we would be very grateful if you would consider increasing your score.

---

> > ### Author Response · Authors · 2025-11-30
> > **Update on  the foraml guarantees on meta-training**
> >
> > In response to the concern of the reviewer  about finite-sample theoretical guarantees for the meta-training phase, we have added new theoretical results in the paper (see updated manuscript). In Appendix Section B.2  we provide finite-sample complexity bounds for ICPE in the meta-training phase, and formal convergence guarantees to an $\epsilon$-optimal policy under appropriate assumptions.
> >
> >  The new analysis culminates in two theorems:
> > - **Theorem B.14** (line 1948) shows how single-step Bellman residuals propagate during training and bounds the suboptimality of the policy at training epoch $k$ in terms of these residuals. The result is controlled by concentrability coefficients that quantify how much errors can be amplified along the induced trajectory distribution.
> > - **Theorem B.15** (line 1968) provides a finite-sample bound on these residuals, decomposing the error into an _approximation_ term (how well the function class can represent the exact Bellman update) and an _estimation_ term that depends on the dimension of the function space and the batch size.
> >
> > We also refer to these results in the main text at lines 300-306.
> >
> >
> > We thank again the reviewer for the discussion and thoughtful feedback. We hope our explanations have clarified the concerns of the reviewer.

---

### Official Review · Reviewer_HFZw · 2025-10-28

**Soundness:** 4
**Presentation:** 3
**Contribution:** 3
**Rating:** 8
**Confidence:** 3

**Summary:**

This paper studies Active Sequential Hypothesis Testing (ASHT) in Bayesian formulation, where each environment is drawn from a prior $M \sim \mathcal{P}$. This problem is also known as pure exploration where actions index hypotheses. There are two settings (i) with a fixed confidence $\delta$ (i.e., stop as soon as the predicted hypothesis is correct w.p. at least  1- $\delta$) and (ii) a fixed budget (use $N$ samples to predict the correct hypothesis).
In both settings, the paper introduces In-Context Pure Explorer (ICPE), a Transformer-based
architecture meta-trained on a family of tasks to jointly learn a sampling  policy  $\pi$ that collects data $\mathcal{D}$ and an inference rule $I(\mathcal{D})$ to identify an environment-specific ground-truth hypothesis $H^*$.
The training phase is implemented using the formulation of Q-learning.  The inference phase consists of the $a  \mathrm{argmax} Q_\theta (D_t,a)$ and the final hypothesis $\hat{H}_N= \mathrm{argmax}_H I_{\phi}(H \mid D_N)$.

**Strengths:**

- When the generative mechanism $P_t$  is not history-dependent, this coincides with Best Arm Identification of multi-armed bandits. The paper  proposes to go beyond  classical ASHT by allowing environment-specific, history-dependent observation kernels $x_{t+1} \sim $P_t (\cdot \mid D_t, a_t)$. The model is very realistic and may be useful in practical cases in real-world settings.



- Policy optimality for both settings using RL is shown.
In the fixed budget setting, the reward is assigned to $r_N(D_N)=\max_H P(H^*=H \mid D_N)$ at the last time step and $0$ otherwise. By defining $V_T(D_T)=r_T(D_T)$, we can define $Q$-function and  the policy $\pi^t(D_t)= \mathrm{argmax}_a Q_t(D_t,a)$. Optimizing with
respect to this reward function yields an optimal solution (Theorem 3.2). Similar results can be shown in fixed confidence settings (Theorem 3.3).


- The literature review is thorough and provides an excellent summary connecting the work to other online learning problems. The connections with BAI and RL, in particular, are well-discussed.
The experiments are also thorough, including several widely used baselines from BAI.

**Weaknesses:**

- Description of the stopping action is not fully specified in the paper. I am assuming a_stop = \mathrm{argmax}_a Q_\theta (s_t,a).
- In the fixed-budget setting experiment, the probability of correctness is evaluated by varying the number of actions. In this test, ICPE’s performance does not consistently surpass the other baselines. What is the performance sensitivity to the budget size N?

**Questions:**

- Is the stopping condition in fixed confidence setting related to existing principles in the BAI literature or completely independent?
-  (Regarding the fixed-budget setting) Could the authors elaborate on ICPE's performance sensitivity to the budget N, given that it did not outperform all baselines when varying the number of actions?

---

> ### Author Response · Authors · 2025-11-20
> **Response to Reviewer HFZw**
>
> We sincerely thank the reviewer for their positive evaluation of our work and for highlighting both the generality of our ASHT formulation and the strength of our theoretical and empirical contributions. We also appreciate the reviewer’s thoughtful concerns and questions, which we address in detail below.
>
> > Description of the stopping action is not fully specified in the paper. I am assuming $a_{\text{stop}} = \arg\max_a Q_\theta(s_t, a)$.
>
> We appreciate this comment, as it allows us to better explain this important detail. The stopping action is introduced in lines 179–180 and 258–262 in the manuscript. In our implementation, the stopping action is modeled as an additional action $a_{\text{stop}}$ in the action set. At each time step, the policy selects
>
> $$
> a_t = \arg\max_{a \in \mathcal{A} \cup \\{a_{\text{stop}}\\}} Q_\theta(D_t, a),
> $$
>
> and if $a_t = a_{\text{stop}}$ the trajectory terminates and we output the final hypothesis $\hat H_t = \arg\max_H I_\phi(H \mid D_t)$. With this action, the learner can decide when to stop during training and testing, enabling it to learn efficient exploration policies. In the revised manuscript, we will briefly restate in the algorithmic description that the stopping rule is implemented as a learned terminal action in the $Q$-network, whose value is trained using the reward functions in Eqs. (5)–(6). This directly connects the stopping action to the fixed-confidence objective via Theorem 3.3.
>
> > In the fixed-budget setting experiment, the probability of correctness is evaluated by varying the number of actions. In this test, ICPE’s performance does not consistently surpass the other baselines. What is the performance sensitivity to the budget size $N$?
> > Could the authors elaborate on ICPE's performance sensitivity to the budget $N$, given that it did not outperform all baselines when varying the number of actions?
>
> We thank the reviewer for raising this point and for drawing attention to the fixed-budget Gaussian bandit experiment (Figure 10). In this setting we consider the standard, unstructured BAI problem with Gaussian rewards and compare ICPE against Track-and-Stop (TaS) as a strong classical baseline. Empirically, ICPE performs on par with TaS as we vary the number of actions $K$ for a fixed budget $N$. We view this as expected rather than concerning, since (i) the problem is unstructured, and (ii) in the fixed-budget regime the reward signal is concentrated at the final time step of longer trajectories, which makes it harder to finely optimize the sampling rule than in the fixed-confidence setting, where many shorter trajectories provide more frequent terminal rewards. With improved training procedures, ICPE could in principle outperform TaS even here, but establishing such gains is only complementary to our current focus and we leave this as an interesting direction for future work.
>
> Regarding sensitivity to $N$ in the fixed-budget setting, we did not explicitly run studies that vary the budget while holding the set of actions and hypotheses fixed. We view such experiments as marginal to our main focus. However, we understand that a more systematic sensitivity analysis with respect to $N$ in the fixed-budget case could be informative, and we will consider adding a concise version of such an experiment in the appendix.
>
> > Is the stopping condition in fixed confidence setting related to existing principles in the BAI literature or completely independent?
>
> We appreciate this question and will clarify the connection to classical fixed-confidence BAI. Our stopping rule is not driven by the usual BAI constructions, which often employ a generalized likelihood ratio (GLR)  combined with a carefully chosen threshold function. We do not rely on these results because we want our method to apply as generally as possible beyond problem-specific analyses. Instead, we take a different route and formally derive the optimal stopping rule as a sequential decision problem. Concretely, we embed the stopping time $\tau$ as a stopping action $a_{\text{stop}}$ and study the dual objective in Eq. (4), which we then solve via reinforcement learning using the reward $r_{t,\lambda}$ in Eq. (5). Appendix B.1.5 and B.1.6 show that optimizing this RL objective yields a Bayes-optimal policy for the dual problem, and Theorem 3.3 then guarantees that for an appropriate $\lambda^\star$ the resulting policy is $\delta$-correct.
>
> We sincerely thank the reviewer again for their feedback and careful reading of our paper. We hope our clarifications have addressed your concerns, and in that case we would greatly appreciate you reconsidering your confidence and even your score upward. We remain committed to further improving our manuscript.

---

### Official Review · Reviewer_Ncbv · 2025-10-31

**Soundness:** 3
**Presentation:** 3
**Contribution:** 3
**Rating:** 6
**Confidence:** 3

**Summary:**

This paper studies the pure exploration problem in both fixed-confidence and fixed-budget settings. It introduces an in-context pure exploration approach. The approach meta-trains Transformers to map observation histories to query actions and a predicted hypothesis. Experiments on various multi-armed bandit problems, general search problems illustrate the advantage of the approach in both the averaged stopping time and correctness of the selected hypothesis.

**Strengths:**

1. Using a transformer-based approach to solve the pure exploration problem is novel and interesting.
2. The approach is theoretically sound. For both the fixed confidence and fixed budget settings, the paper defines an MDP structure with corresponding reward functions. The paper shows that achieving the pure exploration objective is equivalent to finding an optimal policy for the corresponding MDP problem.
3. Compared with existing approaches, the proposed in-context pure exploration does not pose structural assumptions in the underlying problem structure and can perform well without prior knowledge of the underlying problem.
4. Extensive experiments in different BAI problem settings show that the proposed approach achieves better stopping time and a guaranteed correctness level.

**Weaknesses:**

1. My main concern lies in the setup of the training set, as the experimental section does not disclose these details. In general, a core challenge in online learning is that data arrive sequentially. If the method requires an extensive training set, along with prior knowledge of $H^*$ on that training set, and assumes that for each chosen $a_t$ during the training process, the corresponding $x_{t+1}$ can be observed, this may significantly limit its applicability in real-world scenarios. I hope the authors can provide more discussion on this aspect.
2. In experiments, the authors use different metrics in different settings, e.g., Figure 2 presents $P(\tau>t)$ and Figure 4 presents cumulative regret. I think it would be better to compare algorithms in consistent metrics.
3. For deterministic bandits under a fixed-budget setting, it seems natural—even for traditional algorithms—to enumerate all possible actions and then select the one that performs best. What is the author trying to illustrate through this example?

**Questions:**

Please see the last part.

---

> ### Author Response · Authors · 2025-11-20
> **Response to Reviewer Ncbv Part 1**
>
> We sincerely thank the reviewer for their positive comments and thoughtful evaluation of our paper. We particularly appreciate your recognition of ICPE's novelty in using in-context learning to address prior misspecification in existing approaches, as well as your comments on the strength of our transformer backbone and the quality of our experimental evaluation. Below, we address your concerns and questions in detail.
>
> > My main concern lies in the setup of the training set, as the experimental section does not disclose these details. In general, a core challenge in online learning is that data arrive sequentially. If the method requires an extensive training set, along with prior knowledge of $H^\star$ on that training set, and assumes that for each chosen $a_t$ during the training process, the corresponding $x_t$ can be observed, this may significantly limit its applicability in real-world scenarios [...]
>
> We thank the reviewer for raising this point and are happy to clarify the training setup. As described in Algorithm 1 and Section 3.2, ICPE is *meta-trained* on a distribution of environments $M \sim \mathcal{P}$ using an online simulator. During training, we interact with these environments where at each step we choose an action $a_t$ and observe a single outcome $x_{t+1}$ from the environment’s generative mechanism. We do not assume access to counterfactual outcomes for unplayed actions or any stronger feedback.
>
> Access to the ground-truth hypothesis $H^\star$ is only required during training in order to train the inference network and to compute the posterior-based rewards to train the exploration policy. This is the same assumption made in most meta-RL and in-context exploration works, where one has a simulator with known task parameters during pretraining, and then deploys the learned policy on new tasks from the same family without access to those parameters [A,B].
>
> At inference time, ICPE operates in the standard online fashion, where when it interacts in a new environment $M \sim \mathcal{P}$, it receives data sequentially, selects actions using only its interaction history, and never observes $H^\star$. Decision making reduces to a forward pass of the trained transformer, so no large training sets or labels are required at deployment.
>
> We acknowledge that the need for an online simulator with known $H^\star$ at meta-training time can limit applicability in domains where such simulators cannot be constructed. We already discuss this in the Limitations section (Lines 934-937). However, it is important to note that ICPE could similarly be trained from purely logged data with offline RL techniques (e.g., IQL, CQL) when direct simulators are unavailable. A full treatment of this offline meta-training setting is beyond the scope of the current work and is an interesting direction for future research.
>
> [A] Supervised Pretraining Can Learn In-Context Reinforcement Learning
>
> [B] Pretraining Decision Transformers with Reward Prediction for In-Context Multi-task Structured Bandit Learning
>
> > In experiments, the authors use different metrics in different settings, e.g., Figure 2 presents $P(\tau > t)$ and Figure 4 presents cumulative regret. I think it would be better to compare algorithms in consistent metrics.
>
> We appreciate the reviewer’s comment regarding our choice of metrics. Our primary goal is pure exploration in the two standard regimes of sequential hypothesis testing: (i) in the fixed-budget setting we evaluate algorithms by the probability of correct identification at a fixed horizon, and (ii) in the fixed-confidence setting we evaluate by the expected stopping time needed to achieve a target confidence level. These are the metrics we use for all core comparisons between ICPE and baselines.
>
> The reviewer is correct that Figure 2 includes a survival curve $P(\tau > t)$ and Figure 4 reports cumulative regret, which go beyond these two main metrics. We included the survival curve to provide additional insight into the full distribution of stopping times across environments, and cumulative regret to illustrate that an explore-then-commit version of ICPE can also perform competitively in standard regret-minimization benchmarks. While these metrics do deviate from the standard, we believe these plots are largely informative in demonstrating both the robustness and applicability of our stopping mechanism. Regardless, we appreciate the reviewer's comment and will consider moving these plots to the appendix in the final manuscript.

---

> ### Author Response · Authors · 2025-11-20
> **Response to Reviewer Ncbv Part 2**
>
> > For deterministic bandits under a fixed-budget setting, it seems natural—even for traditional algorithms—to enumerate all possible actions and then select the one that performs best. What is the author trying to illustrate through this example?
>
> Thank you for raising this question. We agree that in this deterministic setting, an optimal strategy is to enumerate the arms and then commit to the best. Precisely because the optimal behavior is so simple, we use this environment as a sanity check to test whether generic methods actually learn this exploration pattern. Our results show that several standard baselines over-exploit early and fail to visit all arms, while ICPE reliably learns the full-enumeration strategy and achieves close-to-perfect identification accuracy. This highlights that even in “easy’’ cases, existing methods can struggle with pure exploration, whereas ICPE does not.
>
> We sincerely thank the reviewer for their feedback. We hope that our clarifications address your concerns, and we would be grateful if you would consider adjusting your score accordingly.

---

### Official Review · Reviewer_fitX · 2025-11-01

**Soundness:** 3
**Presentation:** 3
**Contribution:** 3
**Rating:** 6
**Confidence:** 3

**Summary:**

The pure exploration problem in sequential decision making is the task of identifying a correct hypothesis based on sequential testing. Applications included best arm identification in multi-armed bandits. This paper presents a method for using in context learning to solve pure exploration problems. Within a multi-task setting, this paper proposes to learn two transformer networks: one for action selection conditioned on history and one for posterior inference conditioned on history. The approach is evaluated on both synthetic multi-armed bandits and real-world inspired domains where it is shown that the approach has a higher probability of returning the correct hypothesis or can reach a set confidence threshold faster. Overall, this paper provides an interesting exploration of learning to learn for the pure exploration problem setting.

**Strengths:**

- The paper is written very well in my opinion. I am not deeply familiar with the pure exploration problem but I still feel like the paper was accessible and did a good job of conveying necessary background and its own technical contribution.
- The experiments have a good breadth of domains covered, including both toy and real-world inspired. Across these domains, the proposed method improves the key performance criteria compared to baselines.
- I find the approach of meta-learning both an inference function and sampling function to be quite interesting. I'm not sure if it entirely novel in the field of meta- or in-context RL but, if it is, it is a nice contribution.

**Weaknesses:**

- The empirical study only considers 3-5 seeds per baseline ran. This seems much too little for understanding the true spread of results. Confidence intervals are computed with hierachical bootstrapping but no explanation for why this method was chosen is given.
- The importance of the theoretical results in this paper is unclear.

**Questions:**

Let's discuss the weaknesses raised above.

---

> ### Author Response · Authors · 2025-11-20
> **Response to Reviewer fitX**
>
> We sincerely thank the reviewer for their positive evaluation of our work and for acknowledging the novel contribution of jointly meta-learning both an inference function and a sampling function for the hypothesis testing problem. We also appreciate the reviewer’s thoughtful concerns, which we address in detail below.
>
> >The empirical study only considers 3-5 seeds per baseline ran. This seems much too little for understanding the true spread of results. Confidence intervals are computed with hierachical bootstrapping but no explanation for why this method was chosen is given.
>
> We appreciate the reviewer’s comment, as it gives us an opportunity to clarify our evaluation protocol and the motivation for using hierarchical bootstrapping for computing confidence intervals. While 3-5 seeds may appear small, each seed corresponds to a single training module of ICPE, where furthermore, each seed is evaluated on 300-1000 independently sampled environments, and for each environment we roll out up to 4500 trajectories (depending on the specific experiment). For metrics such as prediction accuracy and stopping time, this design yields tens of thousands of unique data points per configuration, spanning three levels of hierarchy (seed $\rightarrow$ environment $\rightarrow$ trajectory). To correctly report the right confidence intervals, we use hierarchical bootstrapping as a method to characterize the confidence when the data is clustered at multiple levels. Naively treating the trajectories as independent would underestimate uncertainty, while in this case we can correctly characterize both the training variance, the prior variance and the variance of the trajectories. This results in a more informative and fine-grained characterization of performance. We will clarify this evaluation protocol in the manuscript.
>
> > The importance of the theoretical results in this paper is unclear.
>
> We appreciate this comment, as the theoretical results are quite important to our contribution. Our theory shows an alternative way to derive both exploration and inference rules for active testing problems. In much of the existing literature, solutions are heavily problem-specific. For example, in BAI the optimal sampling rule is typically engineered from an instance-dependent sample complexity lower bound, which changes from setting to setting. In contrast, our results show that learning a sampling rule can be cast as solving a well-defined RL problem whose reward is derived directly from the Bayesian ASHT objective. This perspective lets us obtain exploration and stopping rules for different problems (e.g., BAI, active search) within a single framework, rather than designing a new algorithm and analysis from scratch for each case.
>
> More specifically, we can clarify the meaning of our theoretical results here. The goal of Section 3.1 is to demonstrate that ICPE is not just an interesting meta-RL architecture, but is grounded in the Bayesian active sequential hypothesis testing formulation we study. Proposition 3.1 and Theorems 2.2-2.3 establish that in both the fixed-budget and fixed-confidence pure exploration regime, an optimal policy can be derived in terms of (i) a posterior-based inference rule and (ii) a data collection policy that maximizes a specific $Q$-function whose reward is the maximum posterior mass. This theoretical deduction of the optimal policy informs how the reward function is specified in the algorithm and directly motivates splitting ICPE into an inference network $I_\phi(H \mid D_t)$ (which approximates the posterior) and a $Q$-network $Q_\theta(D_t,a)$ (which learns the optimal acquisition policy and the $\delta$-aware stopping rule via the stop action $(a_{\text{stop}}$). In other words, the theory specifies what quantity should be optimized (posterior concentration for identification), how this induces a formal RL objective, and thus why ICPE will converge to an optimal policy, as opposed to just choosing a reward shaping scheme ad hoc. We will revise the final paper to make these connections more explicit by adding short takeaway summaries after each main theorem.
>
>
> We sincerely thank the reviewer for their feedback. We hope our clarifications have addressed your concerns, and in that case we would greatly appreciate you reconsidering your score upward. We remain committed to further improving our manuscript.

---

> > ### Comment · Reviewer_fitX · 2025-11-25
> >
> > Thanks for your clarifications! My concern about the number of seeds persists. While you are generating a lot of data points, the small number of seeds means that uncertainty due to the seed isn't integrated out well. Hypothetically, if performance could vary greatly by seed but varied little by environment or trajectory then your estimate of the mean uisng a few seeds will still be high variance even with infinite environment and trajectory samples per seed.

---

> ### Author Response · Authors · 2025-11-26
>
> We understand the reviewer’s concern and, in general, we agree with the comment. Let us first  offer a more quantitative explanation of why we chose hierarchical bootstrapping, and then conclude on why it is unlikely that the performance varies greatly across seeds.
>
> ---
> (_Example with a random effect model_). First, note that  a naive  bootstrap that ignores this hierarchical structure would underestimate uncertainty.
>
> To fix the ideas, consider the following random-effects model
> $$
> Y_{a,b,c} = \mu + \alpha_a + \beta_{a,b,} + \gamma_{a,b,c},\quad
> \alpha_a\sim(0,\sigma^2_{\text{seed}}),
> \beta_{a,b}\sim(0,\sigma^2_{\text{env}}),
> \gamma_{a,b,c}\sim(0,\sigma^2_{\text{traj}}),
> $$
> and let $\bar Y=\frac{1}{mKN}\sum_{a=1}^m\sum_{b=1}^k\sum_{c=1}^N (Y_{a,b,c})$ with $m$ seeds, $K$ environments per seed and $N$ trajectories per environment.
>
>
>
> The variance of $\bar Y$ then is
> $\text{Var}(\bar Y) = \frac{\sigma^2_{\text{seed}}}{m}+ \frac{\sigma^2_{\text{env}}}{mK} + \frac{\sigma^2_{\text{traj}}}{mKN}$.
> A naive bootstrap over the $mKN$ trajectories instead targets
> $Var_{naive}(\bar Y)
> \approx \frac{\text{ Var}(Y_{a,b,c})}{mKN}=\frac{\sigma^2_{\text{seed}}+\sigma^2_{\text{env}}+\sigma^2_{\text{traj}}}{mKN},$
> which effectively shrinks the seed-level contribution by a factor $KN$.
> This is precisely why we use a hierarchical  bootstrap, so that the  $\sigma^2_{\text{seed}}/m$ term is correctly reflected in our confidence intervals. Therefore, in general, if performance varied greatly across seeds but little by environment
> or trajectory, one would expect the resulting hierarchical confidence intervals to be wide.
>
>
> ---
>
> (_Our experiments_).
> Now, coming to our experiments, we observed that the standard deviation is relatively small in general, and the reported confidence intervals are
> quite narrow. Let's consider the possibility the sample variance across seeds is underestimated.
> Under a simple Gaussian model, with $m$ seeds the probability of observing a sample variance $S^2$ that is smaller, let's say $1/10$ (one order of magnitude smaller) of the true value $\sigma_{\text{seed}}^2$  is $P(S^2\leq \sigma_{\text{seed}}^2/10)=P((m-1)S^2/\sigma_{\text{seed}}^2 \leq (m-1)/10) = P(\chi_{m-1}^2\leq (m-1)/10)$. For $m=5$ this yields $P(\chi_{4}^2\leq 0.4)\approx 0.02$ (and $\approx 0.023$ for $m=3$). The probability of observing such a small variance is small under this assumption.
>
>
>   That said, we agree that 5 seeds cannot fully rule out rare catastrophic seeds. However, since we observed similar results across  the different problems we presented in the paper, it seems quite unlikely for this to be the case.   We  hope this explanation addressed your concern, and we will clarify this random-effects view in the final version of the paper to explain how we took into account the overall uncertainty.

---

> > ### Comment · Reviewer_fitX · 2025-11-26
> >
> > Thanks for the detailed explanation. Before addressing your comments specifically, let me add that re-running your experiments with an appropriate number of seeds could be an easy way to resolve this issue. I realize that isn't always very easy so I'll engage with your points below as an alternative path to resolving this issue.
> >
> > My concern with your analysis is that you are first assuming a Gaussian distribution to make your argument but we don't know that the distribution is Gaussian -- it easily could be some fat-tailed distribution. My second concern is that bootstrapping is known to potentially fail with small sample sizes. From a quick Google search, it seems that practitioners recommend 20-30 samples (seeds in this case). Admittedly, that recommendation is for standard percentile BCI and not hierarchical BCI.
> >
> > So I don't buy "Therefore, in general, if performance varied greatly across seeds but little by environment or trajectory, one would expect the resulting hierarchical confidence intervals to be wide" as the seed contribution may be under-estimated due to the small sample size.
> >
> > I'll add that I appreciate the attention to choosing appropriate confidence tools that you have made.

---

> > > ### Author Response · Authors · 2025-11-26
> > > **Official Comment by Authors**
> > >
> > > Thank you for your quick reply and for acknowledging our effort to compute uncertainty in a statistically principled way. Our conclusion is based on the observation that we see consistent behavior across all experiments and settings reported in the paper.
> > >
> > > While we would love to present results with additional seeds right now, we are currently constrained by compute and time during the rebuttal period. However, we commit to increasing the number of seeds and updating the corresponding confidence intervals in the final version of the paper, and we hope the reviewer is comfortable with this compromise.

---

### Official Review · Reviewer_J4j4 · 2025-11-05

**Soundness:** 2
**Presentation:** 2
**Contribution:** 2
**Rating:** 4
**Confidence:** 2

**Summary:**

The paper introduces In-Context Pure Exploration, a Transformer-based meta-RL framework that learns how to collect information adaptively and identify hypotheses without any parameter updates at test time. ICPE shows that in-context learning can implement adaptive experiment design—learning both data-acquisition and inference rules from experience.

**Strengths:**

(1) The theoretical formulation is principled and elegant. The authors derive an information-theoretic reward function directly from the posterior optimality conditions, avoiding ad hoc design choices.
(2) By showing that Transformers can meta-learn to explore—learning both when and how to query—ICPE opens a new research direction connecting in-context learning with active learning and sequential testing.

**Weaknesses:**

(1) It remains unclear how ICPE scales to unseen environment distributions or how its meta-training distribution affects performance.
(2) The experimental suite, though diverse, primarily focuses on low-dimensional or discrete problems. These settings validate proof-of-concept behavior but do not test ICPE’s limits in large or continuous hypothesis spaces.
(3) Many experiments rely on synthetic or well-defined priors over tasks, where the environment distribution and hypotheses are known. In more realistic scenarios, such priors are unavailable or nonstationary.
(4) While ablations such as I-DPT and I-IDS isolate the inference component, the paper lacks a deeper analysis of what the Transformer learns in-context.
(5) Meta-training Transformers on families of ASHT tasks could be expensive.

**Questions:**

(1) How sensitive is ICPE to the distribution of training environments?
(2) How would ICPE extend to continuous or combinatorial hypothesis spaces?
(3) The method involves several interacting components (Q-network, inference network, and λ updates). How sensitive is performance to the stability of these updates or to hyperparameters like λ’s learning rate, buffer size, and synchronization intervals (Tθ, Tϕ)?
(4) In the fixed-confidence setting, λ and the stop-action encode correctness constraints. How robust is this mechanism in practice?

---

> ### Author Response · Authors · 2025-11-20
> **Response to Reviewer J4j4 Part 1**
>
> We appreciate the reviewer’s feedback and the time they dedicated to evaluating our paper. In particular, we appreciate how the reviewer highlighted the theoretical strength and novelty of our work. Below, we address your concerns in detail.
>
> > - It remains unclear how ICPE scales to unseen environment distributions or how its meta-training distribution affects performance.
> > - How sensitive is ICPE to the distribution of training environments?
>
> We thank the reviewer for raising this point and for the opportunity to clarify how ICPE behaves under distribution shift. As an in-context learning method, ICPE is designed to be meta-trained on the same family of tasks on which it will be deployed, consistent with prior work in in-context RL. That said, we agree that understanding robustness to changes in the environment distribution is important, and we therefore ran additional experiments to probe this sensitivity.
>
> Concretely, we trained ICPE in the stochastic fixed-confidence bandit setting described in lines 304–306, where environments are sampled from a uniform distribution over Gaussian bandits with a minimum gap. At test time, we then evaluated the same trained model on bandit instances drawn from *shifted* environment distributions. We constructed these shifts by sampling reward means from a symmetric Dirichlet distribution with parameter $\alpha$ chosen so that
>
> $$
> \mathrm{KL}(\mathrm{Dir}(\alpha,\dots,\alpha)\,\|\,\mathrm{Dir}(1,\dots,1)) = \text{target KL},
> $$
>
> thereby controlling the divergence from the uniform training distribution. Intuitively, varying the target KL controls how concentrated generated samples are with respect to the simplex. In the table below, we report ICPE’s correctness and average stopping time across a range of KL values and number of actions. Across all experiments, we observe that both correctness and stopping time remain remarkably stable, with only minor fluctuations within the reported confidence intervals. This suggests that ICPE is not excessively sensitive to moderate shifts in the environment distribution around the training family. We are happy to add these experimental results to the appendix of the manuscript, yet we note that extending this type of analysis to more severe out-of-distribution shifts and to other structured settings is a direction for future work.
>
> | KL Divergence from Uniform | Correctness (K=4) | Avg. Stop Time (K=4) | Correctness (K=6) | Avg. Stop Time (K=6) | Correctness (K=8) | Avg. Stop Time (K=8) |
> | -------------------------- | ----------------- | --------------------- | ----------------- | --------------------- | ----------------- | --------------------- |
> | 0.00                       | 0.91 ± 0.01       | 7.76 ± 0.20           | 0.91 ± 0.01       | 9.78 ± 0.22           | 0.90 ± 0.01       | 11.37 ± 0.22          |
> | 0.25                       | 0.91 ± 0.01       | 7.59 ± 0.19           | 0.91 ± 0.01       | 9.97 ± 0.26           | 0.89 ± 0.01       | 11.45 ± 0.26          |
> | 0.50                       | 0.90 ± 0.01       | 7.65 ± 0.21           | 0.91 ± 0.01       | 9.79 ± 0.26           | 0.89 ± 0.01       | 11.54 ± 0.26          |
> | 1.00                       | 0.90 ± 0.01       | 7.68 ± 0.20           | 0.90 ± 0.01       | 9.89 ± 0.28           | 0.90 ± 0.01       | 11.33 ± 0.24          |
> | 2.00                       | 0.89 ± 0.01       | 7.63 ± 0.21           | 0.90 ± 0.01       | 9.86 ± 0.28           | 0.89 ± 0.01       | 11.41 ± 0.28          |
> | 4.00                       | 0.89 ± 0.01       | 7.73 ± 0.22           | 0.90 ± 0.01       | 10.07 ± 0.28          | 0.88 ± 0.01       | 11.47 ± 0.28          |
>
> | KL Divergence from Uniform | Correctness (K=10) | Avg. Stop Time (K=10) | Correctness (K=12) | Avg. Stop Time (K=12) | Correctness (K=14) | Avg. Stop Time (K=14) |
> | -------------------------- | ------------------ | ---------------------- | ------------------ | ---------------------- | ------------------ | ---------------------- |
> | 0.00                       | 0.91 ± 0.01        | 15.41 ± 0.37           | 0.91 ± 0.01        | 18.86 ± 0.51           | 0.91 ± 0.01        | 22.23 ± 0.72           |
> | 0.25                       | 0.92 ± 0.01        | 15.13 ± 0.37           | 0.91 ± 0.02        | 18.28 ± 0.52           | 0.92 ± 0.01        | 22.63 ± 0.71           |
> | 0.50                       | 0.91 ± 0.01        | 15.35 ± 0.40           | 0.91 ± 0.01        | 18.55 ± 0.53           | 0.91 ± 0.01        | 22.18 ± 0.75           |
> | 1.00                       | 0.90 ± 0.01        | 15.33 ± 0.42           | 0.90 ± 0.01        | 18.78 ± 0.52           | 0.91 ± 0.01        | 22.36 ± 0.72           |
> | 2.00                       | 0.91 ± 0.01        | 15.54 ± 0.41           | 0.91 ± 0.01        | 19.00 ± 0.60           | 0.91 ± 0.01        | 22.57 ± 0.75           |
> | 4.00                       | 0.91 ± 0.01        | 15.22 ± 0.40           | 0.91 ± 0.01        | 18.52 ± 0.54           | 0.91 ± 0.01        | 22.97 ± 0.75           |

---

> ### Author Response · Authors · 2025-11-20
> **Response to Reviewer J4j4 Part 2**
>
> > - The experimental suite [...] does not test ICPE’s limits in large or continuous hypothesis spaces.
> > - How would ICPE extend to continuous or combinatorial hypothesis spaces?
>
> You raised a good point regarding continuous settings. Indeed, transitioning ICPE to continuous hypothesis spaces is, in general, nontrivial (a simple way would be to discretize the hypothesis space being considered).
>
> In fact, how to deal with continuous hypothesis spaces is a current venue of research, even for classical theory papers that deal with pure exploration problems (see for example [A] below).
>
> - With a continuous hypothesis space, the goal is to identify an $\epsilon$-ball $B_\epsilon$ of correct hypotheses. However, there is an issue in ensuring $\delta$-correctness at the boundary of such a ball (it becomes harder to say if a hypothesis is true or not with confidence $1-\delta$); see also [A].
> - Another approach [B], proposes to explore as if the target radius of the ball is $\epsilon/2$ (instead of $\epsilon$). This forces a gap that can be exploited for guaranteeing $\delta$-correctness for the original $\epsilon$-ball (their algorithm uses $\epsilon/2$ in line 1 and 2, but the guarantees hold for the original $B_\epsilon$ ball). In their case, such an approach leads to a sample complexity that is nearly optimal (a multiplicative factor $4$ appears).
>
> We believe this last approach is an exciting venue for research and could potentially be applied to ICPE for extension to continuous hypothesis spaces. We will ensure to discuss this setting in detail in the final version of the manuscript; however, we view this extension as a strict direction of future work.
>
> [A] Garivier, Aurélien, and Emilie Kaufmann. “Nonasymptotic sequential tests for overlapping hypotheses applied to near-optimal arm identification in bandit models.” *Sequential Analysis* 40.1 (2021): 61–96.
>
> [B] Russo, Alessio, and Aldo Pacchiano. “Adaptive Exploration for Multi-Reward Multi-Policy Evaluation.” *Forty-second International Conference on Machine Learning*, 2025.
>
> > Many experiments rely on synthetic or well-defined priors [...] In more realistic scenarios, such priors are unavailable or nonstationary.
>
> While our experiments assume access to a well-specified family of environments, we note that this is a standard assumption shared by most Bayesian and meta-RL approaches. ICPE requires a meta-training distribution over tasks, but this prior need not be perfectly known in practice. In applied settings it can be induced by a simulator or generative model calibrated to historical data, and as long as the prior covers a sufficiently rich family of environments, the resulting policy can generalize effectively within that family.
>
> Regarding nonstationarity, we respectfully disagree with the statement that most realistic environments for BAI/ASHT-style problems are non-stationary. In classical BAI or ASHT, the goal is to localize a fixed ground-truth hypothesis within an episode, such as the best arm in a bandit, the hidden number in binary search, or, in more realistic cases, a patient’s diagnosis or whether a protein has a certain binding affinity. These problems do not naturally admit a non-stationary hypothesis within a single episode. Allowing the underlying hypothesis to drift over time would fundamentally change the problem (for example, even defining a "best arm" becomes ambiguous) and leads to a different class of nonstationary decision problems that fall outside the standard ASHT/BAI framework. Extending in-context exploration methods to explicitly nonstationary task families is an interesting direction for future work, but is orthogonal to the ASHT setting we study here.
>
> > While ablations such as I-DPT and I-IDS isolate the inference component, the paper lacks a deeper analysis of what the Transformer learns in-context.
>
> Understanding what a Transformer learns in-context is a difficult and broadly open question in interpretability. In our work, we provide behavioral rather than mechanistic evidence of what ICPE learns. Specifically in the magic-action and magic-chain bandits, ICPE reliably exploits the underlying structure by rapidly probing the “magic” arms and adapting its strategy as the number and position of magic actions change. Likewise, in the feedback-graph experiments located in Section D.4, ICPE’s sampling patterns respect the graph geometry, querying informative neighbors rather than treating arms as independent. It would be implausible to believe that ICPE could consistently demonstrate such structure-aware behavior without learning a representation of the task geometry, therefore demonstrating that the Transformer must be learning such structure in-context. A full mechanistic interpretability study of the Transformer’s internal representations would require substantial additional analysis and is orthogonal to our primary contribution of formulating and demonstrating ICPE as a meta-learned ASHT procedure.

---

> ### Author Response · Authors · 2025-11-20
> **Response to Reviewer J4j4 Part 3**
>
> > Meta-training Transformers on families of ASHT tasks could be expensive.
>
> We would like to emphasize that our aim is to introduce a novel meta-learning approach to the classical ASHT problem. We recognize that meta-training Transformers on families of ASHT tasks carries a nontrivial computational cost, and note this is standard for deep RL and in-context learning methods. However, this cost is incurred once at training time and then amortized over many downstream episodes, where ICPE runs with a single forward pass and no parameter updates.
>
> > The method involves several interacting components [...] How sensitive is performance to the stability of these updates or to hyperparameters like $\lambda$’s learning rate, buffer size, and synchronization intervals ($T_\theta$, $T_\phi$)?
>
> We appreciate this question and agree that, in principle, stability of the joint updates is important. In practice, we find that ICPE is most sensitive to the overall learning-rate scale, and much less sensitive to buffer size and synchronization intervals. Our updates form a two-timescale optimization procedure, where the dual variable $\lambda$ is updated more slowly than the $Q$-network and inference network. In all experiments we use a $\lambda$ learning rate that is smaller than those of the networks. Buffer size and target-network synchronization intervals ($T_\theta$, $T_\phi$) are chosen following standard DQN heuristics and kept fixed across experiments, and we found that moderate changes to these values did not qualitatively affect performance. More generally, we have not performed a dedicated ablation study, since exhaustively evaluating hyperparameter and architectural variations across every setup would be prohibitively extensive.
>
> > In the fixed-confidence setting, $\lambda$ and the stop-action encode correctness constraints. How robust is this mechanism in practice?
>
> Empirically, we find the mechanism to be highly robust. Across all fixed-confidence experiments, the observed correctness closely matches the desired level $1-\delta$. This is shown for stochastic bandits in Figure 2(c), for binary search in Table 1, for magic-action bandits in Figure 11(a), and for the magic-room MDPs in Table 2 and Figures 18–19, where the empirical coverage consistently lies near $1-\delta$ with only small fluctuations. This aligns with our dual formulation in Section 3.1, where $\lambda$ is updated to enforce the correctness constraint.
>
>
> We sincerely thank the reviewer again for their valuable feedback. We hope our clarifications
> have addressed all of your concerns, and we would greatly appreciate it if you could consider increasing
> your score accordingly.

---

### Author Response · Authors · 2025-11-30
**Global Response to Reviewers and ACs**

We sincerely thank the area chairs and all the reviewers for their thoughtful evaluation and positive feedback.

We are particularly glad that the reviewers have found the paper to be well written, theoretically rigorous and with a good breadth of experiments. We believe the reviews have been extremely helpful in clarifying our presentation and strengthening both the theoretical and empirical components of the paper.

We   have carefully addressed the points raised by the reviewers in detail. Below, we provide a **summary of central concerns and our revisions.**

- ``Clarifying the theory and its takeaways.`` Per feedback from Reviewers **fitX** and **g3VE**, we have added short paragraphs after each main result in Section 3.1 to explain the intuitive meaning and role of Proposition 3.1 (see lines 157-159) and Theorems 3.2–3.3 (see lines 178-182 and 210-214 respectively), and how they motivate ICPE’s architecture, stopping rule and reward design.

- ``Fixed-budget vs. fixed-confidence training.`` In response to Reviewer **g3VE**, we made it clearer  in the main text that ICPE uses separate models for the fixed-budget and fixed-confidence regimes, each trained with its own reward function and $Q$-learning objective derived from the corresponding theoretical formulation in Section 3.1 (see lines 242-244).

- ``Robustness to distribution shifts.`` In response to reviewer **J4j4**, we  ran additional experiments to probe the robustness of ICPE, and reported the results in the paper (see Appendix D.1.2 paragraph at line 3657; and in the main at line 355).

- ``Deterministic bandit experiment.`` Following Reviewer **Ncbv**’s feedback, we expanded the discussion of the deterministic bandit experiment in Figure 3 to clarify its purpose as a sanity check (see lines 382-386): ICPE reliably recovers the simple but optimal “enumerate all arms” exploration strategy in a setting where several baselines fail, highlighting the need for better learned exploration policies even in seemingly easy problems.

- ``Continuous hypotheses, real-world deployment and limitations.`` Addressing comments from Reviewers **J4j4** and **g3VE**, we added a more explicit discussion in Section 5 of both the challenges and potential approaches for extending ICPE to continuous  hypothesis spaces,  also promising directions for real-world deployment (see paragraph at line 510). In response to Reviewer **Ncbv**, we also commented on the limitation of having access to an online simulator (same paragraph at line 510). We also remind the readers that there is a "Limitations" section in the first page of the appendix.

- ``Hierarchical bootstrapping and uncertainty quantification.`` Per Reviewer **fitX**’s request, we added a concise description of hierarchical bootstrapping in the appendix, explaining what it is, how it respects the seed-environment-trajectory hierarchy in our data, and why it provides a statistically principled way to compute confidence intervals in our experiments (see appendix D, paragraph at line 3498). As promised to the reviewer, we will also increase the number of seeds used in the experiments.

- ``Finite-sample and meta-learning guarantees.`` In response to Reviewer  **g3VE**'s concern about finite-sample theoretical guarantees for the meta-training phase, we have added new theoretical results in Appendix Section B.2 that provide finite-sample complexity bounds for ICPE in the meta-training phase, and formal  convergence guarantees  to an $\epsilon$-optimal policy under appropriate assumptions. The new analysis culminates in two theorems (we refer to these results in the main text at lines 300-306):
    - **Theorem B.14** (line 1948) shows how single-step Bellman residuals propagate during training and bounds the suboptimality of the policy at training epoch $k$ in terms of these residuals. The result is controlled by concentrability coefficients that quantify how much errors can be amplified along the induced trajectory distribution.
    - **Theorem B.15** (line 1968) provides a finite-sample bound on these residuals, decomposing the error into an _approximation_ term (how well the function class can represent the exact Bellman update) and an _estimation_ term that depends on the dimension of the function space and the batch size.



We take the opportunity to thank again the area chairs and the reviewers for their careful and constructive feedback. We have worked hard during the rebuttal period to incorporate your suggestions, and we believe the resulting revisions significantly improve the clarity and impact of the paper.

---

### Meta-Review · Area_Chair_hCPa · 2025-12-24

**Summary:**

This paper studies in-context learning (ICL) as a mechanism for exploration, specifically for pure exploration, the goal of gathering information to identify a correct hypothesis. The proposed framework, In-Context Pure Exploration (ICPE), meta-trains Transformers to map observation histories to query actions and a predicted hypothesis, enabling the model to transfer in-context without parameter updates. The method is evaluated on various benchmarks, including Best-Arm Identification (BAI) and generalized search tasks, demonstrating competitive performance with adaptive baselines.

The paper is being praised for the elegant theoretical formulation and derivation of a principled information-theoretic reward function that avoids ad-hoc design.
Reviewers appreciated the fact that the proposed method unifies several exploration settings under a common framework, demonstrating versatility across different problem domains.
The presentation also include a literature review that has been described as thorough and providing an excellent summary connecting the work to other online learning problems.
Reviewers expressed positive opinions about the breadth of domains covered in terms of empirical evaluation, which also includes several widely used baselines from BAI.

The main outstanding concerns following the rebuttal relate to an extension to the continuous hypothesis space, and requests for more rigorous real-world experiments that have not been directly addressed in the rebuttal. However, these concerns are arguably beyond the scope of the current work.

**Reviewer Concerns:**

* Addressed in the rebuttal:
  - Questions about robustness to shift to unseen environments have been addressed with additional bandit experiments where bandit instances are drawn from shifted environment distributions at test time.
  - Fundamental questions about the mechanism enabling in-context learning in transformers have been superficially but satisfactorily addressed in a discussion paragraph in the rebuttals, since this is arguably a question beyond the scope of the current work.
  - Questions about the evaluation protocol, number of independent runs and statistical aggregation have been answered in the rebuttals, including with the promise of running more runs with more independent random seeds for the final version of the paper.

* Not addressed in the rebuttal:
  - Question about extension to continuous hypothesis spaces is only partially addressed in the rebuttal alluding to a couple of possible research venues. This is however arguably beyond the scope of the current work.
  - Requests for more rigorous real-world experiments have not been directly addressed in the rebuttal, with authors however acknowledging the importance of such an effort, but arguing that this would an subsequent complementary development following this initial contribution.

**Reviewer Scores:**

| Reviewer | initial score | predicted final score |
|---:|---:|---:|
| J4j4 | 4 | 6 |
| fitX | 6 | 6 |
| Ncbv | 6 | 6 |
| HFZw | 8 | 8 |
| g3VE | 4 | 6 |

---

### Decision · Program_Chairs · 2026-01-26

Accept (Poster)